# LS²MC-GDA: A Smoothed Algorithm for Federated Stochastic Multi-Level Compositional Minimax Optimization

**Xinwen Zhang** [1]  **Richard Souvenir** [1]  **Hongchang Gao** [1]

## Abstract

Federated stochastic multi-level compositional minimax optimization supports a growing number of machine learning applications. However, the interplay of multi-level compositional structure and minimax formulation in federated learning setting poses significant optimization challenges, resulting in slow convergence rates for existing algorithms. In this paper, we propose a novel federated learning algorithm, LS²MC-GDA, that leverages smoothing techniques and variance-reduced stochastic compositional gradients. To support our theoretical analysis, we introduce a stage-wise extension of LS²MC-GDA, which serves to bridge the gap between different stationarity measures. We establish that our algorithm achieves a sample complexity of $O(\kappa^{3/2}/N\epsilon^3)$ and a communication complexity of $O(\kappa/\epsilon^2)$, substantially improving existing theoretical results in terms of the condition number $\kappa$ and the solution accuracy $\epsilon$ and achieving a linear speedup with respect to the number of workers $N$. Finally, experimental results validate the effectiveness of our algorithm.

## 1. Introduction

Stochastic multi-level compositional optimization has garnered increasing interest due to its wide range applications, such as model-agnostic meta-learning (Finn et al., 2017), risk-averse portfolio optimization (Bruno et al., 2016; Shapiro et al., 2021), reinforcement learning (Dann et al., 2014; Zhang et al., 2020b). More recently, its minimax formulation has emerged in tasks that include deep AUC maximization (Yuan et al., 2021), multi-objective learning (Gu et al., 2022), and multi-instance learning (Zhu et al., 2023). With the rising demand for large-scale distributed

computation and increasing concerns on data privacy, recent works (Zhang et al., 2023; 2024) have begun to investigate federated stochastic compositional minimax optimization. However, existing methods typically rely on the nonconvex–strongly-concave (NC–SC) assumption, which represents a restrictive special case of the more general nonconvex–PL (NC–PL) condition.

Motivated by this gap in theoretical generality, we study federated stochastic compositional minimax optimization under the broader NC–PL framework. The PL condition (Polyak, 1963), weaker than strong concavity, has enabled convergence guarantees in various problems, including linear quadratic regulators (Fazel et al., 2018), overparameterized neural networks (Liu et al., 2022), robust phase retrieval (Sun et al., 2018), and deep AUC maximization (Guo et al., 2020). Moreover, NC–PL has recently been adopted in federated minimax optimization (Deng & Mahdavi, 2021; Sharma et al., 2022; Wu et al., 2023). Despite these advances, theoretical guarantees for federated multi-level compositional minimax optimization under NC–PL remain largely unexplored.

Addressing this problem is technically challenging due to the complex interplay among multi-level compositional structure, minimax formulation, and federated learning settings. First, the multi-level compositional structure introduces bias at each level when estimating function values and gradients (Yang et al., 2019; Balasubramanian et al., 2022). Second, the minimax formulation induces biased stochastic gradients for both primal and dual variables, complicating stability and convergence analysis. Third, the distributed setting introduces additional consensus errors between local and global models, which interact non-trivially with compositional and minimax optimization process (Gao & Huang, 2021; Gao et al., 2022; Yang & Li, 2024; Gao, 2024b). Recently, (Zhang et al., 2024) proposed LocalSMCGDAM to address these challenges; however, its convergence rate of $O(\kappa^4/N\epsilon^4)$ remains substantially slower than known results for standard federated minimax or compositional optimization, highlighting a gap between theory and practice. This naturally leads to the central question of this work:

**Can we design an algorithm with provable convergence guarantees that achieves an improved rate for federated**

---

[1]Department of Computer and Information Sciences, Temple University, Philadelphia, USA. Correspondence to: Hongchang Gao <hongchang.gao@temple.edu>.

*Proceedings of the 43ʳᵈ International Conference on Machine Learning*, Seoul, South Korea. PMLR 306, 2026. Copyright 2026 by the author(s).

**stochastic multi-level compositional minimax optimization problems?**

To answer this question, we propose the Local Stochastic Smoothed Multi-level Compositional Gradient Descent Ascent algorithm (LS²MC-GDA), which integrates both smoothing and variance reduction techniques in the federated learning setting. Smoothing was first introduced for single-machine minimax optimization (Zhang et al., 2020a), and recently extended to multi-level compositional minimax problems (Zhang & Gao, 2025b). However, these developments remain confined to the single-machine setting. FESS-GDA (Shen et al., 2024) subsequently introduced smoothing to federated learning, but applied it only to the global model, thereby requiring a very small local learning rate. This design often leads to degraded practical performance and introduces additional hyperparameter tuning overhead. To accelerate convergence, we propose applying the smoothing technique to local models and incorporating variance reduction techniques into our algorithm, leveraging methods such as STORM (Cutkosky & Orabona, 2019), which improves convergence via recursive momentum; these techniques have also been explored in multi-level compositional problems (Jiang et al., 2022; Gao, 2024b; Zhang & Gao, 2025b). However, establishing convergence guarantees of our algorithm presents unique technical challenges, particularly in simultaneously controlling gradient estimation errors and consensus errors across local and global workers under a multi-level compositional minimax structure, thereby necessitating new theoretical analysis.

Another challenge arises in the translation between different stationarity measures. As discussed in (Yang et al., 2022b), smoothing algorithms typically guarantee an $(\epsilon, \epsilon/\sqrt{\kappa})$-stationary point, which must then be further converted into an $\epsilon$-stationary point for fair comparison with baselines. Existing analyses of this translation are limited either to standard federated minimax problems (Shen et al., 2024) or to single-machine settings (Zhang & Gao, 2025b). Extending such results to the federated multi-level compositional case is highly nontrivial, as it requires carefully controlling the consensus errors of all variance-reduced variables across multiple levels and workers. To overcome this challenge, we develop a new theoretical analysis based on a stage-wise variant without smoothing updates. Finally, we validate our approach through extensive experiments.

In summary, our contributions are as follows:

- We propose LS²MC-GDA method under the federated NC-PL condition, which incorporates smoothing and variance reduction techniques. It achieves an $(\epsilon, \epsilon/\sqrt{\kappa})$-stationary point with a convergence rate of $O(\kappa^{3/2}/N\epsilon^3)$, demonstrating both level-independent convergence and linear speedup with respect to the number of workers $N$.

- We further develop a stage-wise variant with an $O(1/N\epsilon)$ convergence rate under the federated PL-PL condition, which enables the translation from an $(\epsilon, \epsilon/\sqrt{\kappa})$-stationary point of LS²MC-GDA to an $\epsilon$-stationary point. As a result, LS²MC-GDA achieves a sample complexity of $O(\kappa^{3/2}/N\epsilon^3)$ and a communication complexity of $O(\kappa/\epsilon^2)$, outperforming existing baselines. Detailed comparisons are provided in Table 1.

- Extensive experimental results under the federated NC–PL framework confirm the effectiveness of our proposed method.

## 2. Related Work

**Multi-Level Compositional Optimization.** (Yang et al., 2019) first introduced multi-level compositional optimization, achieving a sample complexity of $O(\epsilon^{-(7+K)/2})$, where the rate depends exponentially on the number of levels $K$. Subsequent works (Balasubramanian et al., 2022; Zhang & Xiao, 2021; Jiang et al., 2022; 2025) adopted moving-average and variance reductions to improve the rate to $O(\epsilon^{-4})$ and $O(\epsilon^{-3})$, achieving level-independent convergence, where the order of the convergence rate does not depend on $K$. Multi-level compositional optimization has also been investigated in distributed settings (Yang & Li, 2024; Gao, 2024a;b). In particular, (Yang & Li, 2024) proposed a federated learning algorithm for strongly-convex problems, and (Gao, 2024b) developed an algorithm for nonconvex problems that can achieve linear speedup with respect to the number of workers under heterogeneous settings. Recently, multi-level compositional minimax optimization has attracted increasing attention. In the single-machine setting, (Gao et al., 2021; Liu et al., 2024; Deng et al., 2025) studied two-level minimax compositional problems corresponding to the case $K = 1$, while (Zhang & Gao, 2025b) investigated the more general multi-level minimax compositional setting. In the federated learning setting, (Zhang et al., 2024) is among the few existing works. However, it suffers from a relatively slow convergence rate and relies on strong assumptions.

**Federated Minimax Optimization.** Early work by (Reisizadeh et al., 2020) formulated robust federated learning problems in the PL-PL and NC-PL minimax settings. (Deng & Mahdavi, 2021) introduced the LocalSGDA method, which performs local updates and periodic communication to solve federated minimax optimization problems. Subsequently, a series of algorithms have been developed under the federated NC-SC setting (Tarzanagh et al., 2022; Sun & Wei, 2022; Zhang et al., 2023; 2024; 2025) and the federated NC-PL setting (Sharma et al., 2022; Yang et al., 2022a; Wu et al., 2023; Shen et al., 2024). Notably, (Wu et al.,

*Table 1.* Comparison with state-of-the-art federated stochastic minimax algorithms, to achieve an $\epsilon$-stationary point. $\kappa$ denotes the condition number. **Sample complexity** refers to the number of stochastic samples required to finding an $\epsilon$-stationary point. **Communication complexity** refers to the number of communication rounds required among workers during training. Under the NC-SC/NC-PL condition, our LS$^2$MC-GDA consistently outperforms all baselines. Under the PL-PL condition, FESS-GDA achieves a better communication complexity because its stochastic gradient is an unbiased estimator, while our Stagewise-LSMC-GDA cannot due to the biased gradient estimator caused by the complex multi-level compositional structure.

| Algorithms | Compositional Level | Assumption | Sample Complexity | Communication Complexity |
|---|---|---|---|---|
| FedSGDA-M[a] (Wu et al., 2023) | ✗ | NC-SC/NC-PL | $O\left(\kappa^3/N\epsilon^3\right)$ | $O\left(\kappa^2/\epsilon^2\right)$ |
| FESS-GDA[b] (Shen et al., 2024) | ✗ | NC-SC/NC-PL | $O\left(\kappa^2/N\epsilon^4\right)$ | $O\left(\kappa/\epsilon^2\right)$ |
| | | PL-PL | $O\left(\log(1/\epsilon)/N\epsilon^2\right)$ | $O\left(\log(1/\epsilon)\right)$ |
| LocalSCGDAM[c] (Zhang et al., 2023) | $K=1$ | NC-SC | $O\left(\kappa^4/N\epsilon^4\right)$ | $O\left(\kappa^3/\epsilon^3\right)$ |
| LocalSMCGDAM[d] (Zhang et al., 2024) | $K>1$ | NC-SC | $O\left(\kappa^4/N\epsilon^4\right)$ | $O\left(\kappa^3/\epsilon^3\right)$ |
| LS$^2$MC-GDA (Corollary 3.7) | $K>1$ | NC-PL | $O\left(\kappa^{3/2}/N\epsilon^3\right)$ | $O\left(\kappa/\epsilon^2\right)$ |
| Stagewise-LSMC-GDA (Remark 3.9) | $K>1$ | PL-PL | $O\left(1/N\epsilon\right)$ | $O\left(1/\epsilon^{1/2}\right)$ |

[a] Using the STORM momentum-based variance-reduction.
[b] Using the standard gradient.
[c,d] Using the moving-average momentum.

2023) incorporated a STORM gradient estimator (Cutkosky & Orabona, 2019) to reduce variance, achieving an improved sample complexity of $O(\epsilon^{-3})$. Meanwhile, (Shen et al., 2024) applied smoothing (Yang et al., 2022b) to federated minimax optimization, enhancing the condition number dependence to $O(\kappa^2)$ under the NC-PL assumption, and further established convergence guarantees under the PL–PL condition. (Zhang et al., 2023; 2024) extended the standard federated minimax framework to multi-level compositional optimization, addressing the additional challenges introduced by nested objective structures. More recently, (Zhang & Gao, 2025a) explored the impact of heavy-tailed noise in federated stochastic minimax optimization.

# 3. Federated Stochastic Multi-level Compositional Minimax Optimization

## 3.1. Problem Setup

In this section, we define the federated stochastic multi-level compositional minimax optimization problem:

$$\min_{x\in\mathbb{R}^p}\max_{y\in\mathbb{R}^q}\frac{1}{N}\sum_{n=1}^{N}f^{(n)}\left(\frac{1}{N}\sum_{n'=1}^{N}G^{(n')}(x),y\right). \quad (1)$$

Here, $G^{(n')}(x) \triangleq g^{(n',K)}(\cdots(g^{(n',1)}(x))\cdots)$ denotes a $K$-level compositional function, where each $g^{(n',k)}(\cdot) = \mathbb{E}[g^{(n')}(\cdot\,;\xi^{(n',k)})]$ is the expectation of the $k$-th inner-level function on the $n'$-th worker, with respect to the random variable $\xi^{(n',k)}$. Similarly, $f^{(n)}(\cdot,\cdot) = \mathbb{E}[f^{(n)}(\cdot,\cdot;\zeta^{(n)})]$ represents the expected outer-level function on the $n$-th

worker, where $\zeta^{(n)}$ is a random variable. Throughout this paper, we assume that $f^{(n)}(\cdot,\cdot)$ defines a nonconvex-PL (NC-PL) minimax problem.

We introduce the assumptions adopted throughout this paper, which are commonly used in prior work (Yang et al., 2022b; Shen et al., 2024; Zhang et al., 2024; Zhang & Gao, 2025b).

**Assumption 3.1. [Lipschitz Smooth]** For any worker $n \in \{1,\ldots,N\}$ and any inner-level $k \in \{1,\ldots,K\}$, $g^{(n,k)}(\cdot;\xi)$ is $C_g$-Lipschitz continuous and $L_g$-smooth, where $C_g > 0$ and $L_g > 0$; $f^{(n)}(\cdot,\cdot;\zeta)$ is $C_f$-Lipschitz continuous and $L_f$-smooth, where $C_f > 0$ and $L_f > 0$.

**Assumption 3.2. [Bounded Variance]** For any worker $n \in \{1,\ldots,N\}$ and any inner-level $k \in \{1,\ldots,K\}$, the stochastic gradients $\nabla g^{(n,k)}(\cdot;\xi^{(k)})$ and $\nabla_x f^{(n)}(\cdot,\cdot;\zeta)$, $\nabla_y f^{(n)}(\cdot,\cdot;\zeta)$ have upper bounded variance $\sigma^2$, where $\sigma > 0$.

**Assumption 3.3. [PL condition]** For any fixed $x \in \mathbb{R}^p$, $\max_{y\in\mathbb{R}^q} f(x,y)$, has a nonempty solution set and a finite optimal value. There exists $\mu > 0$ such that $\|\nabla_y f(x,y)\|^2 \geq 2\mu(f(x,y^*(x)) - f(x,y))$, where $y^*(x) = \arg\max_{y\in\mathbb{R}^q} f(x,y)$.

Under Assumption 3.1 and 3.3, and based on the Lipschitz properties established in Lemma C.1, we define $\ell = \max\{L_f, C_g^{2K}L_f + C_f\sum_{k=0}^{K-1}L_f C_g^{K-1+k}\}$ and set the condition number as $\kappa = \ell/\mu$.

**Definition 3.4. [Stationarity Measures]** $(x,y)$ is an $(\epsilon_1,\epsilon_2)$-stationary point of a differentiable function $f(x,y)$,

if $\|\nabla_x f(x,y)\| \leq \epsilon_1$ and $\|\nabla_y f(x,y)\| \leq \epsilon_2$. $x$ is an $\epsilon$-stationary point of a differentiable function if $\|\nabla \mathcal{L}(x)\| \leq \epsilon$, where $\mathcal{L}(x) = f(x, y^*(x))$ and $y^*(x) = \arg\max_{y \in \mathbb{R}^q} f(x,y)$.

We further follow (Yang et al., 2020; Chen et al., 2022) and introduce the two-sided PL condition:

**Assumption 3.5. [Two-sided PL condition]** A differentiable function $f(x,y)$ satisfies two-sided PL condition, *i.e.*, there exists $\mu > 0$ such that $f(\cdot, y)$ is $\mu$-PL for any $y \in \mathbb{R}^q$, and $-f(x, \cdot)$ is $\mu$-PL for any $x \in \mathbb{R}^p$.

### 3.2. Revisiting Current Methods: Limitations and Motivations

As the first effort toward solving the federated stochastic multi-level compositional minimax problem in Eq. (1), (Zhang et al., 2024) proposed the LocalSMCGDAM algorithm. Following their terminology, the inner-level function on $n$-th worker can be denoted as:

$$G^{(n,k)}(x) = g^{(n,k)}\left(G^{(n,k-1)}(x)\right), \quad (2)$$

$$\nabla G^{(n,k)}(x) = \nabla G^{(n,k-1)}(x)\nabla g^{(n,k)}(G^{(n,k-1)}(x)),$$

where $k \in \{1, \ldots, K\}$.

Then, by denoting $G(x) = \frac{1}{N}\sum_{n'=1}^N G^{(n',K)}(x)$ and $\nabla G(x) = \frac{1}{N}\sum_{n'=1}^N \nabla G^{(n',K)}(x)$, the compositional gradients can be represented as:

$$\nabla_x f_g = \nabla G(x) \times \frac{1}{N}\sum_{n=1}^N \nabla_G f^{(n)}(G(x), y),$$

$$\nabla_y f_g = \frac{1}{N}\sum_{n=1}^N \nabla_y f^{(n)}(G(x), y). \quad (3)$$

A critical issue in Eq. (1) lies in the inherent bias in both the global $K$-level function and its corresponding gradient, which results in biased estimators for $\nabla_x f_g$ and $\nabla_y f_g$. To mitigate this issue, LocalSMCGDAM employs a STORM-like technique applied to the $k$-th inner level function on the $n$-th worker as:

$$h_{t+1}^{(n,k)} = (1 - \alpha\eta^2)(h_t^{(n,k)} - g^{(n,k)}(h_t^{(n,k-1)}; \xi_{t+1}^{(n,k)}))$$
$$+ g^{(n,k)}(h_{t+1}^{(n,k-1)}; \xi_{t+1}^{(n,k)}), \quad (4)$$

which effectively reduce the variance in the estimates.

However, a non-negligible limitation remains: the algorithm achieves only an $O(1/\epsilon^4)$ convergence rate, which is weaker than the $O(1/\epsilon^3)$ rate established in the single-machine setting (Liu et al., 2024; Deng et al., 2025; Zhang & Gao, 2025b), as well as for distributed multi-level compositional algorithms (Gao, 2024a;b). Moreover, in terms of dependence on the condition number $\kappa$, LocalSMCGDAM

exhibits a rate of $O(\kappa^4)$, which is worse than the $O(\kappa^2)$ dependence reported by the algorithm in (Shen et al., 2024) in the context of federated minimax optimization. Therefore, current algorithms for Eq. (1) remain far from satisfactory in terms of convergence rate, both in their dependence on $\epsilon$ and $\kappa$. Furthermore, the analysis of LocalSMCGDAM is based on NC-SC assumption, and it remains an open question how the convergence rate behaves under the weaker NC-PL assumption. These limitations motivates the development of new algorithms with improved convergence guarantees.

### 3.3. LS$^2$MC-GDA

To propose a more efficient algorithm, we first investigate which component of the existing method in (Zhang et al., 2024) primarily contributes to its suboptimal convergence rate. Specifically, we attribute the slow convergence with respect to $\epsilon$ to the term $O(\eta/N)$, which emerges when bounding the error between the moving-average momentum and the stochastic compositional gradients. This suggests that introducing STORM-like updates to the stochastic compositional gradients on $x$ and $y$ may improve the convergence rate. On the other hand, the worse dependence on $\kappa$ stems from the hyperparameter relation $\gamma_x = \gamma_y \cdot O(1/\kappa^2)$ with $\gamma_y = O(1/\kappa)$. Building on the insights from (Yang et al., 2022b; Shen et al., 2024), a natural approach is to incorporate the smoothing technique into Eq. (1) by introducing an auxiliary variable $z \in \mathbb{R}^p$:

$$F(G(x), y; z) = \frac{1}{N}\sum_{n=1}^N f^{(n)}(G(x), y) + \frac{\omega}{2}\|x - z\|^2, \quad (5)$$

where $\omega > 0$ is a hyperparameter controlling the smoothing term $\|x - z\|^2$, and $F(G(x), y; z)$ is strongly convex with respect to $x$ for an appropriate $\omega$, e.g., $\omega = 2\ell$. Based on the smoothed loss in Eq. (5), we now present our novel algorithm, Local Stochastic Smoothed Multi-level Compositional Gradient Descent Ascent method (LS$^2$MC-GDA), as detailed in Algorithm 1 (the complete version is provided in Appendix C).

For simplicity, we denote the stochastic estimated compositional gradients of the smoothed loss with respect to $x$ and $y$ on $n$-th worker as:

$$\nabla_x F_{h,t;t}^{(n)} := \nabla g^{(n,1)}(x_t^{(n)}; \xi_t^{(n,1)})\nabla g^{(n,2)}(h_t^{(n,1)}; \xi_t^{(n,2)})$$
$$\cdots \nabla g^{(n,K)}(h_t^{(n,K-1)}; \xi_t^{(n,K)})\nabla_1 f^{(n)}(h_t^{(n,K)}, y_t^{(n)}; \zeta_t^{(n)})$$
$$+ \omega(x_t^{(n)} - z_t^{(n)}),$$
$$\nabla_y F_{h,t;t}^{(n)} := \nabla_2 f^{(n)}(h_t^{(n,K)}, y_t^{(n)}; \zeta_t^{(n)}), \quad (6)$$

where $h_t^{(n,k)}$ is obtained from Eq. (4), and the gradients with respect to the primal variable $x$ are derived via the chain rule. Here, the first subscript $t$ refers to the gradient computation at the $t$-th iteration, while the second $t$ indexes the stochastic samples drawn at the $t$-th iteration.

**Algorithm 1** LS$^2$MC-GDA

**Require:** $x_0$, $y_0$, $z_0$, $\eta > 0$, $\beta_x > 0$, $\beta_y > 0$, $\gamma_x > 0$,
$\gamma_y > 0$, $\gamma_z > 0$, $\beta_x \eta^2 < 1$, $\beta_y \eta^2 < 1$, $\gamma_z \eta < 1$.

1: Initialize $p_0^{(n)}$, $q_0^{(n)}$, and $h_0^{(n,k)}$ for $k \in \{1, \cdots, K\}$ in Eq. (10).
2: **for** $t = \{0, \ldots, T-1\}$, each device $n$ **do**
3:     Update $(x_{t+1}^{(n)}, y_{t+1}^{(n)}, z_{t+1}^{(n)})$ in Eq. (8),
4:     **for** $k = \{0, \ldots, K\}$ **do**
5:         Update $k$-th inner-level variable $h_{t+1}^{(n,k)}$ in Eq. (4)
6:     **end for**
7:     Update stochastic smoothed compositional gradients $p_{t+1}^{(n)}$ and $q_{t+1}^{(n)}$ in Eq. (7)
8:     **if** $(t+1) \mod \tau = 0$ **then**
9:         PERIODCOMM$\big(\mathbf{s}_{t+1}^{(n)}\big)$ in Eq. (9), where $\mathbf{s}_{t+1}^{(n)} \triangleq (x, y, z, h^{(k)}, p, q)_{t+1}^{(n)}$.
10:     **end if**
11: **end for**

To reduce variance, we perform STORM-like updates on the stochastic estimated compositional gradients of the smoothed loss on $x$ and $y$ on $n$-th worker:

$$p_{t+1}^{(n)} = (1 - \beta_x \eta^2)(p_t^{(n)} - \nabla_x F_{h,t;t+1}^{(n)}) + \nabla_x F_{h,t+1;t+1}^{(n)},$$
$$q_{t+1}^{(n)} = (1 - \beta_y \eta^2)(q_t^{(n)} - \nabla_y F_{h,t;t+1}^{(n)}) + \nabla_y F_{h,t+1;t+1}^{(n)}. \quad (7)$$

We then perform local updates on the variables $x_{t+1}^{(n)}$, $y_{t+1}^{(n)}$ and $z_{t+1}^{(n)}$ simultaneously:

$$\begin{aligned}
x_{t+1}^{(n)} &= x_t^{(n)} - \gamma_x \eta p_t^{(n)}, \\
y_{t+1}^{(n)} &= y_t^{(n)} + \gamma_y \eta q_t^{(n)}, \quad\quad (8)\\
z_{t+1}^{(n)} &= z_t^{(n)} + \gamma_z \eta (x_{t+1}^{(n)} - z_t^{(n)}).
\end{aligned}$$

For periodic communication, we denote the synchronization operation by PERIODCOMM$\big(\mathbf{s}_{t+1}^{(n)}\big)$, where $\mathbf{s}_{t+1}^{(n)} \triangleq (x, y, z, h^{(k)}, p, q)_{t+1}^{(n)}$. Specifically, each local variable is replaced by the average across all workers once every $\tau$ iterations:

$$x_{t+1}^{(n)} \leftarrow \frac{1}{N} \sum_{n'=1}^{N} x_{t+1}^{(n')}, \quad y_{t+1}^{(n)} \leftarrow \frac{1}{N} \sum_{n'=1}^{N} y_{t+1}^{(n')},$$

$$z_{t+1}^{(n)} \leftarrow \frac{1}{N} \sum_{n'=1}^{N} z_{t+1}^{(n')}, \quad h_{t+1}^{(n,k)} \leftarrow \frac{1}{N} \sum_{n'=1}^{N} h_{t+1}^{(n',k)}, \quad (9)$$

$$p_{t+1}^{(n)} \leftarrow \frac{1}{N} \sum_{n'=1}^{N} p_{t+1}^{(n')}, \quad q_{t+1}^{(n)} \leftarrow \frac{1}{N} \sum_{n'=1}^{N} q_{t+1}^{(n')}.$$

The initialization of our algorithm is given by

$$p_0^{(n)} = \nabla g^{(n,1)}(x_0^{(n)}; \xi_0^{(n,1)}) \cdots \nabla g^{(n,K)}(h_0^{(n,K-1)}; \xi_0^{(n,K)})$$
$$\nabla_1 f^{(n)}(h_0^{(n,K)}, y_0^{(n)}; \zeta_0^{(n)}) + \omega(x_0^{(n)} - z_0^{(n)}),$$

$$q_0^{(n)} = \nabla_2 f^{(n)}(h_0^{(n,K)}, y_0^{(n)}; \zeta_0^{(n)}), \quad\quad (10)$$
$$h_0^{(n,k)} = g^{(n,k)}(h_0^{(n,k-1)}; \xi_0^{(n,k)}), \ k \in \{1, \cdots, K\}.$$

Next, we establish the convergence rate of our Algorithm 1. Here, we denotes $\bar{a} = \frac{1}{N} \sum_{n=1}^{N} a^{(n)}$ as the average of any variable $a$ over all workers.

**Theorem 3.6.** *Given Assumptions 3.1-3.3, when $\beta_x > 0$, $\beta_y > 0$, $\alpha > 0$, by setting $O(\beta) := O(\beta_x) = O(\beta_y) = O(\alpha)$, $\omega = O(\ell)$,*

$$\gamma_y = \gamma_x \underbrace{\frac{(\omega - \ell)^2}{64\ell^2}}_{\psi_{\gamma_y} = O(1)}, \quad \gamma_z = \gamma_x \underbrace{\frac{(\omega - \ell)^3 \mu}{1024 \times 96 \omega \ell^2}}_{\psi_{\gamma_z} = O(1/\kappa)},$$

*and the conditions in Eq. (44) hold, Algorithm 1 has the following convergence upper bound:*

$$\frac{1}{T} \sum_{t=0}^{T-1} \Big( \mathbb{E}[\|\nabla_x f(G(\bar{x}_t), \bar{y}_t)\|^2] + \kappa \mathbb{E}[\|\nabla_y f(G(\bar{x}_t), \bar{y}_t)\|^2] \Big)$$

$$\leq O\Big(\frac{\kappa \Psi_0}{\gamma_x \eta T}\Big) + O\Big(\frac{\kappa K}{\beta \eta^2 N S_0 T}\Big) + O\Big(\frac{\kappa \beta \eta^2 K}{N}\Big) \quad (11)$$

$$+ O\Big(\frac{\kappa K \tau^2 \eta^4 \beta \gamma_x^2}{N}\Big) + O\big(\kappa \tau^4 \eta^6 \beta^2 \gamma_x^2\big) + O\big(\kappa \tau^2 \eta^4 \beta^2\big).$$

*where $\Psi_0 = F(G(\bar{x}_0), \bar{y}_0; \bar{z}_0) - 2\psi_d(\bar{y}_0; \bar{z}_0) + 2\psi(\bar{z}_0)$, whose definitions can be found in Eq. (15).*

**Corollary 3.7.** *Given Assumptions 3.1-3.3, by setting $\gamma_x = O(1)$, $\gamma_y = O(1)$, $\gamma_z = O\left(\frac{1}{\kappa}\right)$, $\eta = O\left(\frac{N\epsilon}{\kappa^{1/2}}\right)$, $\tau = O\left(\frac{\kappa^{1/2}}{N\epsilon}\right)$, $\beta = O\left(\frac{1}{N}\right)$, $S_0 = O\left(\frac{\kappa^{1/2}}{\epsilon}\right)$, $T = O\left(\frac{\kappa^{3/2}}{N\epsilon^3}\right)$, Algorithm 1 can achieve an $(\epsilon, \epsilon/\sqrt{\kappa})$-stationary point. Here, $S_0$ is the batch size in the initial iteration $(t = 0)$. Notably, our algorithm maintains a constant mini-batch size throughout training, i.e., $S = O(1)$ when $t > 0$. The convergence rate of $O\left(\frac{\kappa^{3/2}}{N\epsilon^3}\right)$ indicates both level-independent convergence, where $K$ appears only in the constants and does not affect the order of the rate, and linear speedup with respect to the number of workers $N$. The communication complexity of our algorithm is $T/\tau = O(\frac{\kappa}{\epsilon^2})$.*

**Technical Challenges.** We highlight several key differences in our convergence analysis compared to the baseline method proposed in (Zhang et al., 2024), which contribute to an improved convergence rate:

1. First, while the baseline analysis bounds the error between the moving-average momentum and the true compositional gradients $\mathbb{E}[\|\bar{p}_{t+1} - \frac{1}{N} \sum_{n=1}^{N} \nabla_x f_{g,t+1}^{(n)}\|^2]$, our analysis instead focuses on the error between the **variance-reduced** momentum and the **estimated** compositional gradients of the **smoothed loss**

$\mathbb{E}[\|\bar{p}_{t+1} - \frac{1}{N}\sum_{n=1}^{N}\nabla_x F_{h,t+1}^{(n)}\|^2]$, as shown in Lemma C.7. In addition, bounding this term is more challenging than in the single-machine setting (Zhang & Gao, 2025b), as it is affected by the consensus error, especially that related to $z$.

2. Second, in addition to bounding the local estimation error $\sum_{k=1}^{K}\nu_k\mathbb{E}[\|g_{h,t+1}^{(n,k)} - h_{t+1}^{(n,k)}\|^2]$, we further include the global estimation error $\sum_{k=1}^{K}\theta_k\mathbb{E}[\|\frac{1}{N}\sum_{n=1}^{N}g_{h,t+1}^{(n,k)} - \frac{1}{N}\sum_{n=1}^{N}h_{t+1}^{(n,k)}\|^2]$ as a new recursive component of the potential function, as shown in Lemma C.9 and Eq. (30).

3. Third, in the baseline method, the consensus error $\frac{1}{N}\sum_{n=1}^{N}\mathbb{E}[\|p_{t+1}^{(n)} - \bar{p}_{t+1}\|^2]$ can be directly bounded due to the use of moving-average momentum. In contrast, our use of **variance-reduced** momentum introduces additional dependencies on the variables $y$, $h^{(n,k)}$, and the auxiliary variable $z$, making the consensus analysis more intricate, as shown in Lemma C.11. To address this challenge, we derive a unified upper bound on the overall consensus error in Lemma C.14.

4. Fourth, unlike the baseline, our convergence analysis is conducted under the **smoothed loss**, which introduces additional challenges. While (Shen et al., 2024) incorporate smoothing in a federated setting, their method applies it only at the global level, with basic SGD updates in a standard minimax framework, making it inadequate for our setting. In contrast, our algorithm combines a multi-level structure, variance-reduced momentum, and locally applied smoothing with periodic communication, resulting in a more technically challenging analysis. We address these difficulties by carefully analyzing both the optimization error $\mathbb{E}[\Psi_{t+1}] - \mathbb{E}[\Psi_t]$, and the consensus error $\frac{1}{N}\sum_{n=1}^{N}\mathbb{E}[\|z_t^{(n)} - \bar{z}_t\|^2]$, as shown in Lemma C.5 and Eq. (28).

5. Last, the baseline analysis assumes a NC-SC setting, whereas our analysis relies on the **NC-PL** assumption, which relaxes the strong concavity and covers a broader class of problems.

These differences underscore the novelty of our convergence analysis in solving Eq. (1), offering a more refined understanding of its complexity behavior. Notably, such analysis is unique to the federated learning setting, particularly in handling consensus errors, and does not arise in single-machine scenarios (Zhang & Gao, 2025b). Importantly, our theoretical results are of independent interest and can potentially be extended to other optimization settings, including traditional minimax problems (Shen et al., 2024) and constrained minimization problems (Huang et al., 2025). Therefore, our work contributes to the theoretical understanding of smoothing techniques in federated learning and may have broader implications for federated optimization.

**Proof Sketch.** We provide a high-level proof sketch outlining the convergence analysis of Algorithm 1. To begin, we divide the proof into two preparatory sections: smoothed properties and federated properties.

Smoothed properties (Section C.1) includes three fundamental descent rules:

- Descent of the optimal value $\psi(z)$, in Lemma C.2;
- Descent of the objective function $F(G(\bar{x}_t), \bar{y}_t; \bar{z}_t)$, in Lemma C.3;
- Descent of the dual function $\psi_d(\bar{y}_t; \bar{z}_t)$, in Lemma C.4.

Based on these results, we define a composite smoothed function:

$$\Psi_t := F(G(\bar{x}_t), \bar{y}_t; \bar{z}_t) - 2\psi_d(\bar{y}_t; \bar{z}_t) + 2\psi(\bar{z}_t),$$

and derive an upper bound of on $\Psi_{t+1} - \Psi_t$ in Lemma C.5.

Federated properties (Section C.2) includes two critical components:

- Gradient Estimation Errors, in Lemma C.7-C.10.
- Consensus Errors, in Lemma C.11-C.14.

Combining these properties, we construct a potential function in Eq. (30). Our goal is to select coefficients that eliminate each error term. The detailed derivation and coefficient selection process are provided in Section C.3, with a step-by-step illustration. The complete proof is given in Appendix C.

### 3.4. Stagewise Variant

Following (Yang et al., 2022b; Shen et al., 2024; Zhang & Gao, 2025b), we note that after obtaining an $(\epsilon, \epsilon/\sqrt{\kappa})$-stationary point $(\hat{x}, \hat{y})$, an $\epsilon$-stationary point of $\mathcal{L}(x)$ can be obtained by approximately solving:

$$\min_{x\in\mathbb{R}^p}\max_{y\in\mathbb{R}^q}\frac{1}{N}\sum_{n=1}^{N}f^{(n)}(G(x),y) + \frac{\omega}{2}\|x-\hat{x}\|^2, \quad (12)$$

where $\hat{x}$ is fixed. Note that this formulation is strongly convex in $x$ when $\omega > \ell$, and therefore it satisfies the two-sided PL condition in Assumption 3.5.

We then propose Algorithm 2 (the complete version is provided in Appendix D), a stage-wise extension designed to analyze the convergence rate of Eq. (12). At each stage, we apply LS$^2$MC-GDA to the objective function in Eq. (12) for $T_r$ iterations, while *omitting the update of $\hat{x}$* (i.e., $z$ in LS$^2$MC-GDA). Upon completing $T_r$ iterations, each local worker randomly selects a local state $\tilde{\mathbf{s}}_{r+1}^{(n),-\mathbf{z}}$ from $\mathbf{s}_{r,t}^{(n),-\mathbf{z}} \triangleq \{(x,y,h^{(k)},p,q)_{r,t}^{(n)}\}_{t=0}^{T_r-1}$, excluding the auxiliary variable $z$, to serve as the initialization of the subsequent stage.

Next, we establish the convergence rate of our Algorithm 2.

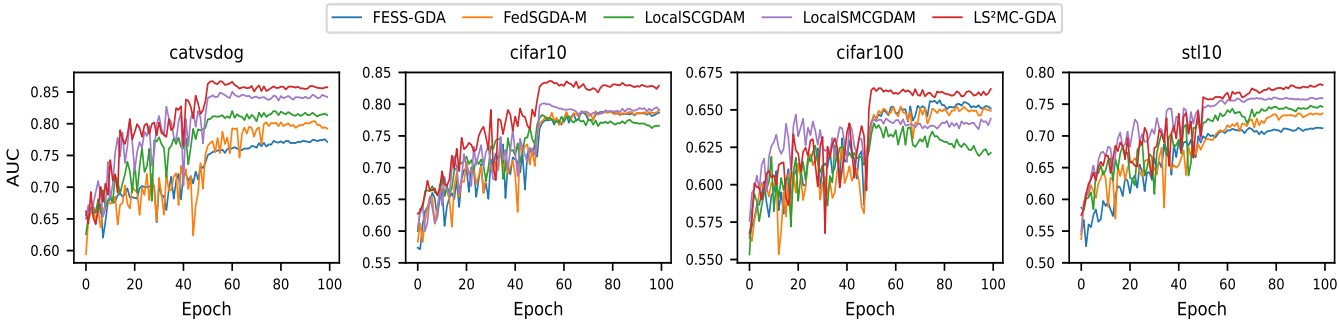

*Figure 1.* The test AUC score versus the number of epochs when communication period is 4.

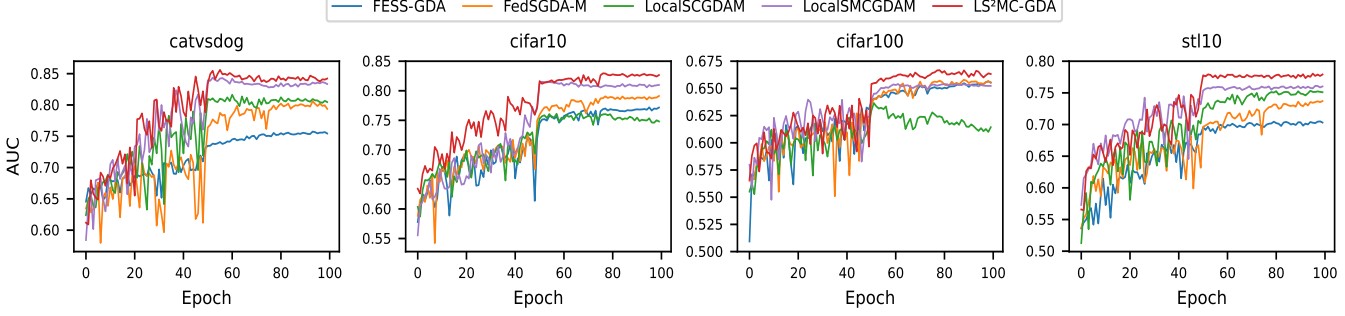

*Figure 2.* The test AUC score versus the number of epochs when communication period is 8.

---

**Algorithm 2** Stagewise-LSMC-GDA

**Require:** $\beta_x > 0, \beta_y > 0, \alpha > 0, \eta_{x,r} > 0, \eta_{y,r} > 0$.

1: **for** Stage $r = 0, \cdots, R-1$, each device $n$ **do**
2:      Initiate $\mathbf{s}_{r,0}^{(n),-\mathbf{z}} = \tilde{\mathbf{s}}_{r,0}^{(n),-\mathbf{z}}$.
3:      **for** $t = 0, \cdots, T_r - 1$, **do**
4:          Perform one iteration LS$^2$MC-GDA update on the objective function Eq. (12).
5:          Randomly select $\tilde{\mathbf{s}}_{r+1}^{(n),-\mathbf{z}}$ from $\{\mathbf{s}_{r,t}^{(n),-\mathbf{z}}\}_{t=0}^{T_r-1}$.
6:      **end for**
7: **end for**

---

**Theorem 3.8.** *Given Assumption 3.1-3.5, by setting $c_0 = \frac{32\ell^2}{\mu^2}$, $\beta_x = 51200\frac{c_0 L_v^2}{N}$, $\beta_y = 1280\frac{L_v^2}{N}$, $\alpha = 1280\frac{c_0 L_v^2}{N}$, $\eta_{y,0} = \frac{N}{40\tau L_v}$, $R_0 = \max\{300\tau^2, \frac{25\mathcal{V}_{0,0}}{NL_v\sigma^2}\}$, where $\mathcal{V}_{0,0}$ is defined in Eq. (48) and $L_v$ in Eq. (63). For any stage $r \geq 1$, $\eta_{x,r} = O(\frac{\mu^2 N}{\sqrt{2^{r-1}}L_v})$, $\eta_{y,r} = O(\frac{N}{\sqrt{2^{r-1}}L_v})$, $\tau = O(\frac{\sqrt{2^{r-1}}}{NL_v})$, $T_r = O(\frac{c_0}{\mu N} \times 2^{r-1})$. Consequently, Algorithm 2 requires a total of $O\left(\frac{1}{\mu^6 N\epsilon}\right)$ iterations to achieve an $\epsilon$-stationary point of $\mathcal{L}(x)$.*

*Remark* 3.9. Algorithm 2 achieves a convergence rate of $O(1/N\epsilon)$ and a communication complexity of $O(1/\epsilon^{1/2})$ for Eq. (12). Under the PL-PL condition, (Shen et al., 2024) establishes an $O(\log(1/\epsilon)/N\epsilon^2)$ convergence rate and an $O(\log(1/\epsilon))$ communication complexity for federated standard minimax optimization. Our method exhibits a little

worse communication complexity than (Shen et al., 2024) due to the biased gradient estimator caused by the more complex multi-level compositional structure. Nevertheless, it is the first work to provide convergence guarantees for federated multi-level compositional minimax optimization under the PL-PL condition.

The proof sketch, unique challenges, and the complete proof of Algorithm 2 are provided in Appendix D.

After establishing the convergence rate of Algorithm 2, we enable to bridge the gap between two stationarity measures and compared our Algorithm 1 with existing baselines:

**Corollary 3.10.** *The convergence rate of Algorithm 2 is $O(\frac{1}{N\epsilon})$, which is significantly faster than that of Algorithm 1. Therefore, the overall iteration complexity for achieving an $O(\epsilon)$-stationary point of $\mathcal{L}(x)$ in Eq. (1) is dominated by Algorithm 1, resulting in a total complexity of $O\left(\frac{\kappa^{3/2}}{N\epsilon^3}\right)$. Similarly, the dominated communication complexity for achieving an $O(\epsilon)$-stationary point is $O\left(\kappa/\epsilon^2\right)$.*

Since PL condition is weaker than strong concavity, the convergence results of Algorithm 1 under the NC-PL setting also hold for NC-SC problems. This allows direct comparison with existing methods designed for both the NC-PL and NC-SC setting.

*Remark* 3.11. The iteration complexity of Algorithm 1 is $O(1/N\epsilon^3)$, which improves upon the $O(1/N\epsilon^4)$ complexity (Zhang et al., 2024) for Eq. (1). It matches the correspond-

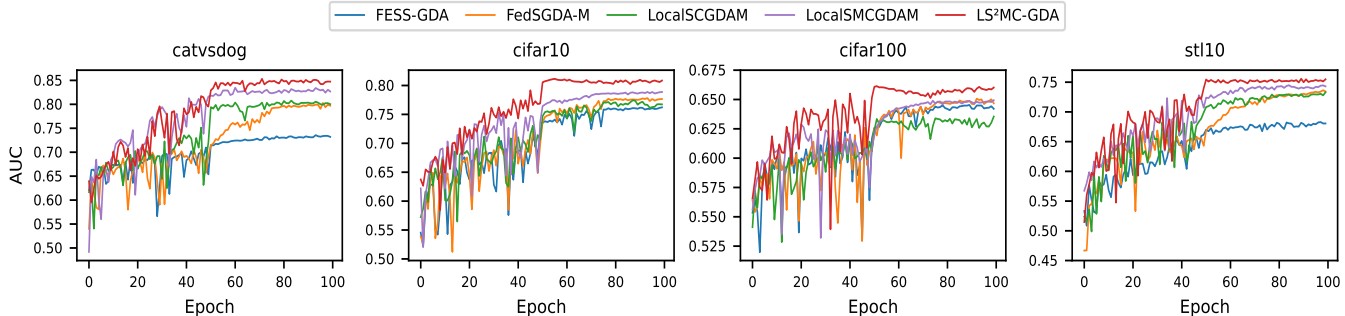

*Figure 3.* The test AUC score versus the number of epochs when communication period is 16.

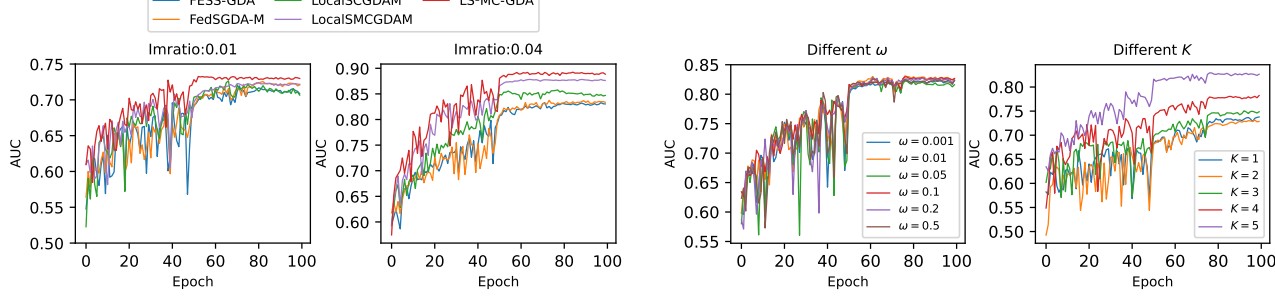

*Figure 4.* Test AUC scores with different hyperparameters on CIFAR10.

ing single-machine results ($N = 1$) (Liu et al., 2024; Deng et al., 2025; Zhang & Gao, 2025b), and aligns with the complexity established for distributed multi-level compositional minimization problems (Gao, 2024a;b).

*Remark* 3.12. The convergence rate of Algorithm 1 exhibits a dependence on the condition number of $O(\kappa^{3/2})$, improving upon the $O(\kappa^4)$ dependence in (Zhang et al., 2024) for Eq. (1). It also outperforms the $O(\kappa^3)$ in (Wu et al., 2023) and $O(\kappa^2)$ in (Shen et al., 2024) under the standard federated minimax setting.

*Remark* 3.13. The communication complexity of Algorithm 1 is $O(\kappa/\epsilon^2)$, which improves upon the $O(\kappa^3/\epsilon^3)$ in (Zhang et al., 2024) for Eq. (1), and matches that of (Shen et al., 2024) under the standard federated minimax setting.

## 4. Experiments

We conduct experiments on federated deep AUC maximization. Prior work (Zhang et al., 2024; Zhang & Gao, 2025c) has shown that deep AUC maximization is an NC-SC problem, and therefore also satisfies the NC-PL condition. All experiments are conducted on four NVIDIA V100 GPUs, each equipped with two compute nodes, resulting in a total of $N = 8$ devices.

We adopt ResNet20 (He et al., 2016) for binary image classification, and evaluate our method on four benchmark datasets: CATvsDOG[1], CIFAR10, CIFAR100, and

STL10 (Krizhevsky et al., 2009; Coates et al., 2011). For the latter three datasets, the first half of the classes are grouped as positive and the remainder as negative. To create imbalance, a fraction of positive samples is randomly removed from the training set, while the test set remains balanced. The imbalance ratio, defined as the proportion of positive samples relative to the total dataset size, is fixed at 0.02 for CatvsDog and CIFAR10, and at 0.04 for CIFAR100, and STL10. The mini-batch size is set to 16 for CatvsDog, CIFAR10, and CIFAR100, and to 8 for STL10.

We compare our algorithm, LS²MC-GDA, with four state-of-the-art federated minimax baselines: FedSGDA-M (Wu et al., 2023), FESS-GDA (Shen et al., 2024), LocalSCG-DAM (Zhang et al., 2023), and LocalSMCGDAM (Zhang et al., 2024). A detailed comparison of these baselines is reported in Table 1. The learning rate $\gamma\eta$ is selected from the set $\{0.02, 0.05, 0.1, 0.3, 0.5\}$, with the best-performing value used in the experiments, and is decayed by a factor of 10 at the 50-th and 75-th epochs for all methods to avoid overfitting. For multi-level compositional methods, the learning rate for each inner-level is further reduced by a factor of 100. For the momentum hyperparameter $\beta$, we fix the product $\beta\eta^2 = 0.9$ and apply this setting for all momentum based baselines. Moreover, we set the smoothing hyperparameter $\omega = 0.1$ for all smoothed based baselines, and fix the inner-level depth $K = 5$ for multi-level compositional methods.

We illustrate the effectiveness of our proposed method by plotting the test AUC results for communication periods of

---

[1] https://www.kaggle.com/c/dogs-vs-cats

4, 8, and 16, as shown in Figure 1, Figure 2, and Figure 3, respectively. Specifically, the large stochastic fluctuations observed at the early stage of training are typical in stochastic minimax optimization. After the learning rate decay at epoch 50, the training process becomes more stable, resulting in the smoother behavior observed thereafter.

For the experimental results, our algorithm consistently achieves superior performance compared to the baseline methods across all settings, attaining the highest testing AUC scores on different datasets and communication periods. In particular, compared to LocalSMCGDAM, our method outperforms this baseline by a large margin in terms of testing AUC, demonstrating the effectiveness of smoothing techniques and variance reduction for stochastic compositional gradients in federated stochastic multi-level compositional minimax optimization. These results further highlight the efficiency of our approach in highly imbalanced federated learning scenarios.

In addition, we conduct a comprehensive hyperparameter study on the CIFAR10 dataset under a communication period of 8, as shown in Figure 4. First, we study the impact of different imbalance ratios (Imratio) by evaluating the test AUC at Imratio values of 0.01 and 0.04. The results show that our method consistently achieves the highest testing AUC across varying imbalance levels, demonstrating its robustness to skewed data distributions. Even under the extreme imbalance ratio of 0.01, our method still achieves the best performance. Next, we tune the smoothing term $\omega$ over $\{0.001, 0.01, 0.05, 0.1, 0.2, 0.5\}$. The results show that our method maintains stable performance across different choices of $\omega$, indicating low sensitivity to the smoothing parameter. Finally, we examine the influence of the number of inner compositional levels $K$ by varying it from 1 to 5. The results show that increasing $K$ generally leads to improved performance, suggesting that our algorithm is capable of effectively leveraging deeper compositional structures.

To further support our theoretical findings, we additionally report the results evaluated using the ergodic iterate on the Catvs-Dog dataset with communication period 8 and imbalance ratio 0.02, as shown in Figure 5. The results show that our proposed method still consistently outperforms the baseline methods. More experiments on federated risk-averse portfolio optimization are provided in the Appendix A.

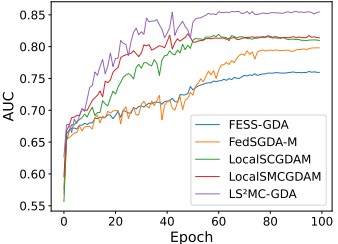

*Figure 5.* Test AUC scores evaluated using the ergodic iterate on CATvsDOG.

# 5. Conclusion

In this work, we propose **LS$^2$MC-GDA**, a novel algorithm that integrates smoothing techniques and variance-reduced gradients to efficiently address the challenges of federated multi-level compositional minimax optimization problems. By further developing a stage-wise variant to bridge the theoretical analysis gap, our algorithm achieves an improved sample complexity of $O(\kappa^{3/2}/N\epsilon^3)$ and a communication complexity of $O(\kappa/\epsilon^2)$, compared with existing baselines. To the best of our knowledge, this is the first work to establish such favorable rate in the federated multi-level compositional minimax optimization problem. Extensive experiments validate the practical effectiveness of our approach.

# Acknowledgements

We thank anonymous reviewers for constructive comments. X. Zhang and H. Gao was partially supported by U.S. NSF CAREER 2339545, NSF IIS 2416607, NSF CNS 2107014.

# Impact Statement

This paper presents work whose goal is to advance the field of Machine Learning. There are many potential societal consequences of our work, none which we feel must be specifically highlighted here.

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

## A. Federated Risk-Averse Portfolio Optimization

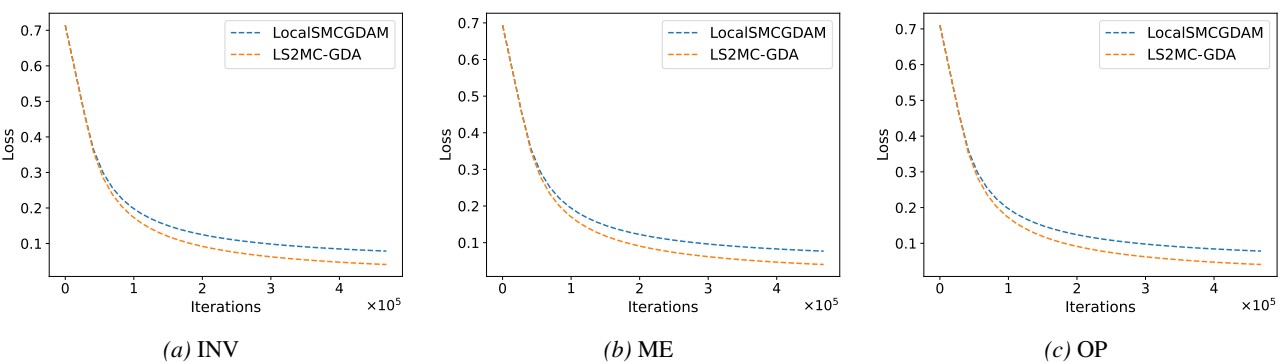

*Figure 6.* Loss over iterations when the communication period is 4.

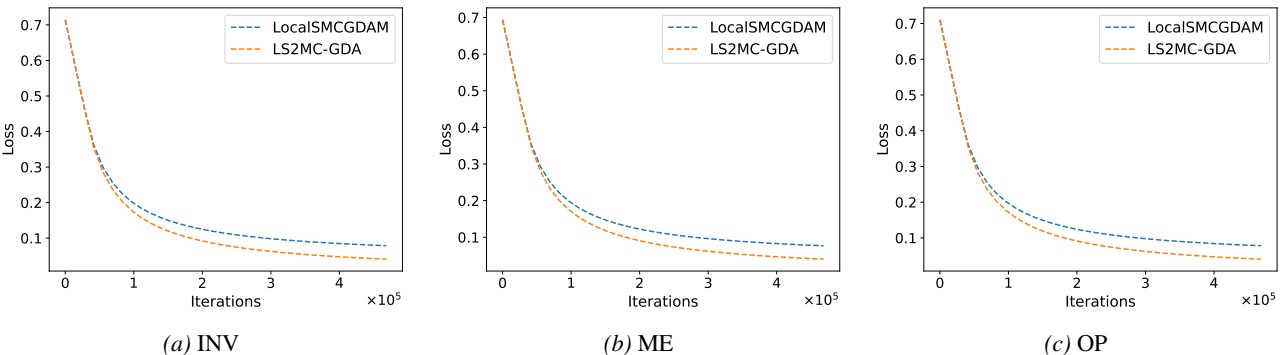

*Figure 7.* Loss over iterations when the communication period is 12.

In the additional experiments, we investigate the risk-averse portfolio optimization problem, a widely studied application of multi-level compositional optimization algorithms (Zhang & Xiao, 2021; Jiang et al., 2022), which has more recently been extended to federated learning (Gao, 2024b) and minimax formulations (Liu et al., 2024). Suppose there are $D$ assets traded over $T$ time steps among $N$ workers. Let $r_t \in \mathbb{R}^D$ denote the payoff vector of these assets at time $t \in \{1, \cdots, T\}$. We reformulate this task as a multi-level compositional minimax optimization problem in the federated setting:

$$\min_x \max_\theta \frac{1}{N} \sum_{n=1}^N \frac{1}{D} \sum_{d=1}^D \theta_d^{(n)} \left( -\frac{1}{T} \sum_{t=1}^T \langle r_t^{(n)}, x^{(n)} \rangle + \sqrt{\frac{1}{T} \sum_{t=1}^T (\langle r_t^{(n)}, x^{(n)} \rangle - \langle \bar{r}^{(n)}, x^{(n)} \rangle} - \|\theta_d^{(n)} - \frac{1}{D}\|^2) \right), \quad (13)$$

where $\bar{r}^{(n)} = \sum_{t=1}^T r_t^{(n)}$, variable $x$ denotes the investment quantity across $d$ assets, variable $\theta$ satisfies $\{\theta = [\theta_d] \in \mathbb{R}^D | \sum_{d=1}^D \theta_d = 1, \theta_d \geq 0, \forall d\}$. This problem is a three-level stochastic compositional minimax optimization problem that satisfies the NC-PL condition, where each level regarding primal variable $x$ can be represented as:

$$f^{(1)}(x) = \left( \frac{1}{T} \sum_{t=1}^T \langle r_t, x \rangle, x \right),$$

$$f^{(2)}(y, x) = \left( y, \frac{1}{T} \sum_{t=1}^T (\langle r_t, x \rangle - y)^2 \right), \quad (14)$$

$$f^{(3)}(z_1, z_2) = -z_1 + \sqrt{z_2}.$$

For evaluation, we compare our approach against LocalSMCGDAM (Zhang et al., 2024) on three datasets: INV, ME, and OP, from Keneth R. French Data Library[2]. Each dataset consists of 13,781 time steps and 100 assets. The learning rate

---

[2] https://mba.tuck.dartmouth.edu/pages/faculty/ken.french/

is set to $\eta = 1e - 6$. We conduct experiments with communication periods of 4 and 12, and report the loss values over iterations in Figures 6 and 7. The results show that our method achieves lower loss values compared to LocalSMCGDAM, demonstrating the effectiveness of the smoothing and variance-reduction techniques.

## B. Notations

To initiate the proof, and following (Zhang et al., 2020a; Yang et al., 2022b; Zhang & Gao, 2025b), we introduce the following auxiliary functions, which will be instrumental in the convergence analysis:

$$\text{dual function:} \quad \psi_d(y; z) = \min_{x \in \mathbb{R}^p} F(G(x), y; z); \quad x^*(y, z) = \arg\min_{x \in \mathbb{R}^p} F(G(x), y; z);$$

$$\text{primal function:} \quad \psi_p(x; z) = \max_{y \in \mathbb{R}^q} F(G(x), y; z); \quad x^*(z) = \arg\min_{x \in \mathbb{R}^p} \psi_p(x; z);$$

$$\text{optimal value:} \quad \psi(z) = \min_{x \in \mathbb{R}^p} \max_{y \in \mathbb{R}^q} F(G(x), y; z); \quad y^*(z) = \arg\max_{y \in \mathbb{R}^q} \psi_d(y; z). \tag{15}$$

Additionally, we introduce the following terminology to simplify the complex expressions. For each $n$-th worker, the true gradients of the original loss function are defined as:

$$\nabla_x f_{g,t}^{(n)} := \nabla g^{(n,1)}(x_t^{(n)}) \nabla g^{(n,2)}(G^{(n,1)}(x_t^{(n)})) \cdots \nabla g^{(n,K)}(G^{(n,K-1)}(x_t^{(n)})) \nabla_1 f^{(n)}(G^{(n,K)}(x_t^{(n)}), y_t^{(n)}),$$

$$\nabla_y f_{g,t}^{(n)} := \nabla_2 f^{(n)}(G^{(n,K)}(x_t^{(n)}), y_t^{(n)}), \tag{16}$$

The corresponding stochastic true gradients of the original loss function on each $n$-th worker at the $t$-th iteration are expressed as:

$$\nabla_x f_{g,t;t}^{(n)} := \nabla g^{(n,1)}(x_t^{(n)}; \xi_t^{(n,1)}) \cdots \nabla g^{(n,K)}(G^{(n,K-1)}(x_t^{(n)}); \xi_t^{(n,K)}) \nabla_1 f^{(n)}(G^{(n,K)}(x_t^{(n)}), y_t^{(n)}; \zeta_t^{(n)}),$$

$$\nabla_y f_{g,t;t}^{(n)} := \nabla_2 f^{(n)}(G^{(n,K)}(x_t^{(n)}), y_t^{(n)}; \zeta_t^{(n)}). \tag{17}$$

Similarly, for the original loss function with estimator $h^{(k)}$, the estimated gradients are defined as:

$$\nabla_x f_{h,t}^{(n)} := \nabla g^{(n,1)}(x_t^{(n)}) \nabla g^{(n,2)}(h_t^{(n,1)}) \cdots \nabla g^{(n,K-1)}(h_t^{(n,K-2)}) \nabla g^{(n,K)}(h_t^{(n,K-1)}) \nabla_1 f^{(n)}(h_t^{(n,K)}, y_t^{(n)}),$$

$$\nabla_y f_{h,t}^{(n)} := \nabla_2 f^{(n)}(h_t^{(n,K)}, y_t^{(n)}), \tag{18}$$

and the corresponding stochastic estimated gradients with estimator $h^{(k)}$ at iteration $t$ are denoted by $\nabla_x f_{h,t;t}^{(n)}$ and $\nabla_y f_{h,t;t}^{(n)}$. For the smoothed loss function, we use the uppercase notation $F$ in the same manner.

At each $k$-th inner level and each $n$-th worker, we denote $g_{h,t}^{(n,k)} := g^{(n,k)}(h_t^{(n,k-1)})$.

## C. Appendix: LS²MC-GDA

### C.1. Smoothed Properties

**Lemma C.1.** *(Zhang & Gao, 2025b) Under Assumptions 3.1–3.3, the following basic **Lipschitz continuity properties** hold:*

*(1).* $\|G^{(k)}(x_1) - G^{(k)}(x_2)\| \le C_g^k \|x_1 - x_2\|$, *for* $k \in \{1, \ldots, K\}$ *and* $C_G = C_g^K$;

*(2).* $\|\nabla G(x_1) - \nabla G(x_2)\| \le L_G \|x_1 - x_2\|$, *where* $L_G = \sum_{k=0}^{K-1} L_g C_g^{K-1+k}$;

*(3).* $\|\nabla_x f(G(x_1), y_1) - \nabla_x f(G(x_2), y_2)\| \le L_F \|(x_1, y_1) - (x_2, y_2)\|$, *where* $L_F = C_G^2 L_f + C_f L_G$;

*(4).* $\|\nabla \mathcal{L}(x_1) - \nabla \mathcal{L}(x_2)\| \le L_{\mathcal{L}} \|x_1 - x_2\|$, *where* $\mathcal{L}(x) = \max_{y \in \mathbb{R}^q} f(G(x), y)$ *and* $L_{\mathcal{L}} = \dfrac{2C_G^2 L_f^2}{\mu} + C_f L_G$;

*By setting* $\ell = \max\{L_F, L_f\}$, *and* $\omega > \ell$, *the following properties hold:*

*(5).* $\|x^*(y_1, z) - x^*(y_2, z)\| \le C_{x_{yz}^1} \|y_1 - y_2\|$, *where* $C_{x_{yz}^1} = \dfrac{\omega + \ell}{\omega - \ell}$;

---

**Algorithm 1** Local Stochastic Smoothed Multi-Level Compositional Gradient Descent Ascent (LS$^2$MC-GDA)

---

**Require:** $x_0, y_0, z_0, \eta > 0, \beta_x > 0, \beta_y > 0, \gamma_x > 0, \gamma_y > 0, \gamma_z > 0, \beta_x \eta^2 < 1, \beta_y \eta^2 < 1, \gamma_z \eta < 1.$

1: Initialize $p_0^{(n)} = \nabla g^{(n,1)}(x_0^{(n)}; \xi_0^{(n,1)}) \cdots \nabla g^{(n,K)}(h_0^{(n,K-1)}; \xi_0^{(n,K)}) \nabla_1 f^{(n)}(h_0^{(n,K)}, y_0^{(n)}; \zeta_0^{(n)}) + \omega(x_0^{(n)} - z_0^{(n)})$ ,
$q_0^{(n)} = \nabla_2 f^{(n)}(h_0^{(n,K)}, y_0^{(n)}; \zeta_0^{(n)})$ , $h_0^{(n,k)} = g^{(n,k)}(h_0^{(n,k-1)}; \xi_0^{(n,k)})$, for $k \in \{1, \cdots, K\}$.

2: **for** $t = 0, \cdots, T-1$, each device $n$ **do**

3:     Update $x$: $x_{t+1}^{(n)} = x_t^{(n)} - \gamma_x \eta p_t^{(n)}$ ,

4:     Update $y$: $y_{t+1}^{(n)} = y_t^{(n)} + \gamma_y \eta q_t^{(n)}$ ,

5:     Update $z$: $z_{t+1}^{(n)} = z_t^{(n)} + \gamma_z \eta(x_{t+1}^{(n)} - z_t^{(n)})$ ,

6:     $h_{t+1}^{(n,0)} = x_{t+1}^{(n)}$ ,

7:     **for** $k = 1, \cdots, K$ **do**

8:         Compute $k$-th inner-level function:
        $h_{t+1}^{(n,k)} = (1 - \alpha\eta^2)(h_t^{(n,k)} - g^{(n,k)}(h_t^{(n,k-1)}; \xi_{t+1}^{(n,k)})) + g^{(n,k)}(h_{t+1}^{(n,k-1)}; \xi_{t+1}^{(n,k)})$

9:     **end for**

10:     Update stochastic smoothed compositional gradients $p_{t+1}^{(n)}$ and $q_{t+1}^{(n)}$:
     $\nabla_x F_{h,t+1;t+1}^{(n)} = \nabla g^{(n,1)}(x_{t+1}^{(n)}; \xi_{t+1}^{(n,1)}) \cdots \nabla g^{(n,K-1)}(h_{t+1}^{(n,K-2)}; \xi_{t+1}^{(n,K-1)}) \nabla g^{(n,K)}(h_{t+1}^{(n,K-1)}; \xi_{t+1}^{(n,K)}) \times$
     $\nabla_1 f^{(n)}(h_{t+1}^{(n,K)}, y_{t+1}^{(n)}; \zeta_{t+1}^{(n)}) + \omega(x_{t+1}^{(n)} - z_{t+1}^{(n)})$ ,
     $\nabla_y F_{h,t+1;t+1}^{(n)} = \nabla_2 f^{(n)}(h_{t+1}^{(n,K)}, y_{t+1}^{(n)}; \zeta_t^{(n)})$ ,
     $p_{t+1}^{(n)} = (1 - \beta_x \eta^2)(p_t^{(n)} - \nabla_x F_{h,t;t+1}^{(n)}) + \nabla_x F_{h,t+1;t+1}^{(n)}$ ,
     $q_{t+1}^{(n)} = (1 - \beta_y \eta^2)(q_t^{(n)} - \nabla_y F_{h,t;t+1}^{(n)}) + \nabla_y F_{h,t+1;t+1}^{(n)}$ ,

11:     **if** $\mathrm{mod}(t+1, \tau) == 0$ **then**

12:       $x_{t+1}^{(n)} = \frac{1}{N}\sum_{n'=1}^N x_{t+1}^{(n')}, \quad y_{t+1}^{(n)} = \frac{1}{N}\sum_{n'=1}^N y_{t+1}^{(n')}, \quad z_{t+1}^{(n)} = \frac{1}{N}\sum_{n'=1}^N z_{t+1}^{(n')},$
      $p_{t+1}^{(n)} = \frac{1}{N}\sum_{n'=1}^N p_{t+1}^{(n')}, \quad q_{t+1}^{(n)} = \frac{1}{N}\sum_{n'=1}^N q_{t+1}^{(n')},$

13:       **for** $k = 1, \cdots, K$ **do**

14:         $h_{t+1}^{(n,k)} = \frac{1}{N}\sum_{n'=1}^N h_{t+1}^{(n',k)},$

15:       **end for**

16:     **end if**

17: **end for**

---

$(6). \quad \|x^*(y, z_1) - x^*(y, z_2)\| \leq C_{x_{yz}^2} \|z_1 - z_2\|, \text{ where } C_{x_{yz}^2} = \frac{\omega}{\omega - \ell};$

$(7). \quad \|x^*(z_1) - x^*(z_2)\| \leq C_{x_z} \|z_1 - z_2\|, \text{ where } C_{x_z} = \frac{\omega}{\omega - \ell}.$

**Lemma C.2.** *Given Assumptions 3.1-3.3, the following inequalities related to **auxiliary functions** hold:*

1. *The function $\psi(z_t)$ satisfies:*

$$\psi(\bar{z}_{t+1}) - \psi(\bar{z}_t) \leq \frac{\omega}{2} \langle \bar{z}_{t+1} - \bar{z}_t, \bar{z}_{t+1} + \bar{z}_t - 2x^*(y^*(\bar{z}_{t+1}); \bar{z}_t) \rangle .$$

2. *By introducing $y^+(\bar{z}_t) = \bar{y}_t + \gamma_y \eta \nabla_y F(x^*(\bar{y}_t, \bar{z}_t), \bar{y}_t; \bar{z}_t)$, the function $x^*(z_t)$ satisfies:*

$$\|x^*(\bar{z}_t) - x^*(y^+(\bar{z}_t); \bar{z}_t)\|^2 \leq \frac{2(1 + \gamma_y^2 \eta^2 \ell^2 C_{x_{yz}^1}^2 + \gamma_y^2 \eta^2 \ell^2)}{(\omega - \ell)\mu} \|\nabla_y F(x^*(\bar{y}_t, \bar{z}_t), \bar{y}_t; \bar{z}_t)\|^2 .$$

3. *The function $x^*(y_t, z_t)$ satisfies:*

$$\|x^*(\bar{y}_{t+1}; \bar{z}_{t+1}) - x^*(y^*(\bar{z}_{t+1}); \bar{z}_{t+1})\|^2 \leq 4(C_{x_z}^2 + C_{x_{yz}^2}^2)\|\bar{z}_{t+1} - \bar{z}_t\|^2 + 4\|x^*(\bar{z}_t) - x^*(y^+(\bar{z}_t); \bar{z}_t)\|^2$$

$$+ \frac{8\gamma_y^2 \eta^2 \ell^2 C_{x_{yz}^1}^2}{(\omega - \ell)^2} \|\nabla_x F(G(\bar{x}_t), \bar{y}_t; \bar{z}_t)\|^2 + 8\gamma_y^2 \eta^2 C_{x_{yz}^1}^2 \|\nabla_y F(G(\bar{x}_t), \bar{y}_t; \bar{z}_t) - \bar{q}_t\|^2 .$$

*Proof.* The first property follows directly from Lemmas B.7 in (Zhang et al., 2020a), and the second property follows directly from Lemma C.2 in (Yang et al., 2022b).

As for the last property, from $y^+(\bar{z}_t) = \bar{y}_t + \gamma_y \eta \nabla_y F(x^*(\bar{y}_t, \bar{z}_t), \bar{y}_t; \bar{z}_t)$, we first obtain

$$
\begin{aligned}
\|y^+(\bar{z}_t) - \bar{y}_{t+1}\|^2 &= \|\bar{y}_t + \gamma_y \eta \nabla_y F(x^*(\bar{y}_t, \bar{z}_t), \bar{y}_t; \bar{z}_t) - \bar{y}_{t+1}\|^2 \leq \gamma_y^2 \eta^2 \|\nabla_y F(x^*(\bar{y}_t, \bar{z}_t), \bar{y}_t; \bar{z}_t) - \bar{q}_t\|^2 \\
&\leq 2\gamma_y^2 \eta^2 \|\nabla_y F(x^*(\bar{y}_t, \bar{z}_t), \bar{y}_t; \bar{z}_t) - \nabla_y F(G(\bar{x}_t), \bar{y}_t; \bar{z}_t)\|^2 + \gamma_y^2 \eta^2 \|\nabla_y F(G(\bar{x}_t), \bar{y}_t; \bar{z}_t) - \bar{q}_t\|^2 \\
&\leq 2\gamma_y^2 \eta^2 \ell \|x^*(\bar{y}_t, \bar{z}_t) - \bar{x}_t\|^2 + \gamma_y^2 \eta^2 \|\nabla_y F(G(\bar{x}_t), \bar{y}_t; \bar{z}_t) - \bar{q}_t\|^2 \,.
\end{aligned}
$$

Then, we derive

$$
\begin{aligned}
\|x^*(\bar{y}_{t+1}; \bar{z}_{t+1}) &- x^*(y^*(\bar{z}_{t+1}); \bar{z}_{t+1})\|^2 = \|x^*(\bar{y}_{t+1}; \bar{z}_{t+1}) - x^*(\bar{z}_{t+1})\|^2 \\
&\leq 4\|x^*(\bar{z}_{t+1}) - x^*(\bar{z}_t)\|^2 + 4\|x^*(\bar{z}_t) - x^*(y^+(\bar{z}_t); \bar{z}_t)\|^2 + 4\|x^*(y^+(\bar{z}_t); \bar{z}_t) - x^*(\bar{y}_{t+1}; \bar{z}_t)\|^2 \\
&\quad + 4\|x^*(\bar{y}_{t+1}; \bar{z}_t) - x^*(\bar{y}_{t+1}; \bar{z}_{t+1})\|^2 \\
&\leq 4C_{x_z}^2 \|\bar{z}_{t+1} - \bar{z}_t\|^2 + 4\|x^*(\bar{z}_t) - x^*(y^+(\bar{z}_t); \bar{z}_t)\|^2 + 4C_{x_{yz}^1}^2 \|y^+(\bar{z}_t) - \bar{y}_{t+1}\|^2 + 4C_{x_{yz}^2}^2 \|\bar{z}_t - \bar{z}_{t+1}\|^2 \\
&\leq 4(C_{x_z}^2 + C_{x_{yz}^2}^2) \|\bar{z}_{t+1} - \bar{z}_t\|^2 + 4\|x^*(\bar{z}_t) - x^*(y^+(\bar{z}_t); \bar{z}_t)\|^2 + 8\gamma_y^2 \eta^2 \ell^2 C_{x_{yz}^1}^2 \|x^*(\bar{y}_t, \bar{z}_t) - \bar{x}_t\|^2 \\
&\quad + 8\gamma_y^2 \eta^2 C_{x_{yz}^1}^2 \|\nabla_y F(G(\bar{x}_t), \bar{y}_t; \bar{z}_t) - \bar{q}_t\|^2 \,.
\end{aligned}
$$

Moreover, due to the strong convexity of $F(G(x), y; z)$ with respect to $x$, we have

$$
\|x^*(\bar{y}_t, \bar{z}_t) - \bar{x}_t\|^2 \leq \frac{1}{(\omega - \ell)^2} \|\nabla_x F(G(\bar{x}_t), \bar{y}_t; \bar{z}_t)\|^2 \,. \tag{19}
$$

By combining the above inequalities, the proof is complete. $\qquad \square$

**Lemma C.3.** *(Zhang & Gao, 2025b) Given Assumptions 3.1-3.3, when $\eta \leq \frac{1}{2\gamma_x(\omega+\ell)}$, and $\gamma_z \eta \leq 1$, the following inequalities hold:*

$$
\begin{aligned}
\mathbb{E}[F(G(\bar{x}_{t+1}), \bar{y}_{t+1}; \bar{z}_{t+1})] &\leq \mathbb{E}[F(G(\bar{x}_t), \bar{y}_t; \bar{z}_t)] - \frac{\gamma_x \eta}{2} \mathbb{E}[\|\nabla_x F(G(\bar{x}_t), \bar{y}_t; \bar{z}_t)\|^2] + \frac{\gamma_y \eta}{2} \mathbb{E}[\|\nabla_y F(G(\bar{x}_t), \bar{y}_t; \bar{z}_t)\|^2] \\
&\quad - \frac{\omega}{2\gamma_z \eta} \mathbb{E}[\|\bar{z}_{t+1} - \bar{z}_t\|^2] + \frac{\gamma_x \eta}{2} \mathbb{E}[\|\nabla_x F(G(\bar{x}_t), \bar{y}_t; \bar{z}_t) - \bar{p}_t\|^2] + \left(\gamma_y \eta \gamma_x^2 \eta^2 \ell^2 - \frac{\gamma_x \eta}{4}\right) \mathbb{E}[\|\bar{p}_t\|^2] \\
&\quad + \left(\frac{3\gamma_y \eta}{4} + \frac{\omega + \ell}{2} \gamma_y^2 \eta^2\right) \mathbb{E}[\|\bar{q}_t\|^2] \,.
\end{aligned}
$$

*Proof.* This lemma follows directly from Lemma B.4 in (Zhang & Gao, 2025b). In that work, the authors explicitly decompose $\mathbb{E}[\|\nabla_x f_\omega(G(x_t), y_t; z_t) - p_t\|^2]$ into two terms: $\mathbb{E}[\|\nabla_x f_\omega(H(x_t), y_t; z_t) - p_t\|^2]$ and $\mathbb{E}[\|g^{(k)}(h_t^{(k-1)}) - h_t^{(k)}\|^2]$, under a single-machine setting. We note that $\mathbb{E}[\|\nabla_x f_\omega(G(x_t), y_t; z_t) - p_t\|^2]$ is used in their lemma, which we believe to be a typo.

In contrast, our algorithm operates in a federated learning setting, where variables are distributed across multiple workers. As a result, this decomposition cannot be directly applied. Therefore, we do not expand this term here and instead defer its treatment to a later stage of the analysis. $\qquad \square$

**Lemma C.4.** *Given Assumptions 3.1–3.3, the dual function $\mathbb{E}[\psi_d(y_t; z_t)]$ is $L_d$-smooth and satisfies:*

$$
\begin{aligned}
\mathbb{E}[\psi_d(\bar{y}_{t+1}; \bar{z}_{t+1})] - \mathbb{E}[\psi_d(\bar{y}_t; \bar{z}_t)] &\geq -\frac{4\gamma_y \eta \ell^2}{(\omega - \ell)^2} \mathbb{E}[\|\nabla_x F(G(\bar{x}_t), \bar{y}_t; \bar{z}_t)\|^2] + \frac{\gamma_y \eta}{2} \mathbb{E}[\|\nabla_y F(G(\bar{x}_t), \bar{y}_t; \bar{z}_t)\|^2] \\
&\quad - \frac{\gamma_y \eta}{2} \mathbb{E}[\|\nabla_y F(G(\bar{x}_t), \bar{y}_t; \bar{z}_t) - \bar{q}_t\|^2] - \left(\frac{\gamma_y^2 \eta^2 L_d}{2} - \frac{7\gamma_y \eta}{16}\right) \mathbb{E}[\|\bar{q}_t\|^2] + \frac{\omega}{2} \mathbb{E}[\langle \bar{z}_{t+1} - \bar{z}_t, \bar{z}_{t+1} + \bar{z}_t - 2x^*(\bar{y}_{t+1}; \bar{z}_{t+1}) \rangle] \,.
\end{aligned}
$$

*Proof.* By Eq. (24) in (Yang et al., 2022b), the dual function $\mathbb{E}[\psi_d(\bar{y}_t; \bar{z}_t)]$ satisfies:

$$
\mathbb{E}[\psi_d(\bar{y}_{t+1}; \bar{z}_{t+1})] - \mathbb{E}[\psi_d(\bar{y}_t; \bar{z}_t)] \geq \gamma_y \eta \mathbb{E}[\langle \nabla_y F(G(x^*(\bar{y}_t, \bar{z}_t)), \bar{y}_t; \bar{z}_t), \bar{q}_t \rangle]
$$

$$- \frac{\gamma_y^2 \eta^2 L_d}{2} \mathbb{E}[\|\bar{q}_t\|^2] + \frac{\omega}{2} \mathbb{E}[\langle \bar{z}_{t+1} - \bar{z}_t, \bar{z}_{t+1} + \bar{z}_t - 2x^*(\bar{y}_{t+1}; \bar{z}_{t+1}) \rangle] .$$

Additionally, we obtain

$$
\begin{aligned}
&\mathbb{E}[\langle \nabla_y F(G(x^*(\bar{y}_t, \bar{z}_t)), \bar{y}_t; \bar{z}_t), \bar{q}_t \rangle] \\
&= \mathbb{E}[\langle \nabla_y F(G(x^*(\bar{y}_t, \bar{z}_t)), \bar{y}_t; \bar{z}_t) - \nabla_y F(G(\bar{x}_t), \bar{y}_t; \bar{z}_t), \bar{q}_t \rangle] + \mathbb{E}[\langle \nabla_y F(G(\bar{x}_t), \bar{y}_t; \bar{z}_t), \bar{q}_t \rangle] \\
&\geq -\frac{1}{2a} \mathbb{E}[\|\nabla_y F(G(x^*(\bar{y}_t), \bar{z}_t), \bar{y}_t; \bar{z}_t) - \nabla_y F(G(\bar{x}_t), \bar{y}_t; \bar{z}_t)\|^2] - \frac{a}{2} \mathbb{E}[\|\bar{q}_t\|^2] \\
&\quad + \frac{1}{2} \mathbb{E}[\|\nabla_y F(G(\bar{x}_t), \bar{y}_t; \bar{z}_t)\|^2] + \frac{1}{2} \mathbb{E}[\|\bar{q}_t\|^2] - \frac{1}{2} \mathbb{E}[\|\nabla_y F(G(\bar{x}_t), \bar{y}_t; \bar{z}_t) - \bar{q}_t\|^2] \\
&= -4\ell^2 \mathbb{E}[\|x^*(\bar{y}_t, \bar{z}_t) - \bar{x}_t\|^2] - \frac{1}{16} \mathbb{E}[\|\bar{q}_t\|^2] + \frac{1}{2} \mathbb{E}[\|\nabla_y F(G(\bar{x}_t), \bar{y}_t; \bar{z}_t)\|^2] + \frac{1}{2} \mathbb{E}[\|\bar{q}_t\|^2] \\
&\quad - \frac{1}{2} \mathbb{E}[\|\nabla_y F(G(\bar{x}_t), \bar{y}_t; \bar{z}_t) - \bar{q}_t\|^2] \\
&\geq -\frac{4\ell^2}{(\omega - \ell)^2} \mathbb{E}[\|\nabla_x F(G(\bar{x}_t), \bar{y}_t; \bar{z}_t)\|^2] + \frac{1}{2} \mathbb{E}[\|\nabla_y F(G(\bar{x}_t), \bar{y}_t; \bar{z}_t)\|^2] + \frac{7}{16} \mathbb{E}[\|\bar{q}_t\|^2] \\
&\quad - \frac{1}{2} \mathbb{E}[\|\nabla_y F(G(\bar{x}_t), \bar{y}_t; \bar{z}_t) - \bar{q}_t\|^2] ,
\end{aligned}
$$

where the second step holds due to Young's inequality, the third step follows from $a = \frac{1}{8}$, and the last step follows from Eq. (19). By combining the above inequalities, the proof is complete. $\qquad\square$

**Lemma C.5.** *Given Assumptions 3.1-3.3, define:*

$$\Psi_t := F(G(\bar{x}_t), \bar{y}_t; \bar{z}_t) - 2\psi_d(\bar{y}_t; \bar{z}_t) + 2\psi(\bar{z}_t) .$$

*If $\gamma_z \eta \leq 1$ and $\gamma_x \leq \min\left\{ \frac{\ell^2}{6\omega(\omega+\ell)^2} , \frac{64\ell}{(\omega-\ell)^2 \sqrt{C_{x_{yz}^1}^2 + 1}} \right\}$, then the following inequality holds:*

$$
\begin{aligned}
\mathbb{E}[\Psi_{t+1}] - \mathbb{E}[\Psi_t] &\leq -\frac{\gamma_x \eta}{4} \mathbb{E}[\|\nabla_x F(G(\bar{x}_t), \bar{y}_t; \bar{z}_t)\|^2] - \frac{\gamma_y \eta}{4} \mathbb{E}[\|\nabla_y F(G(\bar{x}_t), \bar{y}_t; \bar{z}_t)\|^2] \\
&\quad + \left( 2C_{x_{yz}^2} + 24\gamma_z \eta (C_{x_z}^2 + C_{x_{yz}^2}^2) - \frac{1}{3\gamma_z \eta} \right) \omega \mathbb{E}[\|\bar{z}_{t+1} - \bar{z}_t\|^2] + \frac{\gamma_x \eta}{2} \mathbb{E}[\|\nabla_x F(G(\bar{x}_t), \bar{y}_t; \bar{z}_t) - \bar{p}_t\|^2] \\
&\quad + \frac{5\gamma_y \eta}{4} \mathbb{E}[\|\nabla_y F(G(\bar{x}_t), \bar{y}_t; \bar{z}_t) - \bar{q}_t\|^2] + \left( \gamma_y \eta \gamma_x^2 \eta^2 \ell^2 - \frac{\gamma_x \eta}{4} \right) \mathbb{E}[\|\bar{p}_t\|^2] \\
&\quad + \left( \frac{\omega + \ell}{2} \gamma_y^2 \eta^2 + \gamma_y^2 \eta^2 L_d - \frac{\gamma_y \eta}{8} \right) \mathbb{E}[\|\bar{q}_t\|^2] ,
\end{aligned}
$$

*where*

$$\gamma_y = \gamma_x \underbrace{\frac{(\omega - \ell)^2}{64\ell^2}}_{\psi_{\gamma_y} = O(1)} , \quad \gamma_z = \gamma_x \underbrace{\frac{(\omega - \ell)^3 \mu}{1024 \times 96 \omega \ell^2}}_{\psi_{\gamma_z} = O(1/\kappa)} .$$

*Proof.* Based on Lemmas C.3 and C.4, together with the first property in Lemma C.2, we have

$$
\begin{aligned}
\mathbb{E}[\Psi_{t+1}] - \mathbb{E}[\Psi_t] &\leq \left( \frac{8\gamma_y \eta \ell^2}{(\omega - \ell)^2} - \frac{\gamma_x \eta}{2} \right) \mathbb{E}[\|\nabla_x F(G(\bar{x}_t), \bar{y}_t; \bar{z}_t)\|^2] - \frac{\gamma_y \eta}{2} \mathbb{E}[\|\nabla_y F(G(\bar{x}_t), \bar{y}_t; \bar{z}_t)\|^2] \\
&\quad - \frac{\omega}{2\gamma_z \eta} \mathbb{E}[\|\bar{z}_{t+1} - \bar{z}_t\|^2] + \frac{\gamma_x \eta}{2} \mathbb{E}[\|\nabla_x F(G(\bar{x}_t), \bar{y}_t; \bar{z}_t) - \bar{p}_t\|^2] + \gamma_y \eta \mathbb{E}[\|\nabla_y F(G(\bar{x}_t), \bar{y}_t; \bar{z}_t) - \bar{q}_t\|^2] \\
&\quad + \left( \gamma_y \eta \gamma_x^2 \eta^2 \ell^2 - \frac{\gamma_x \eta}{4} \right) \mathbb{E}[\|\bar{p}_t\|^2] + \left( \frac{\omega + \ell}{2} \gamma_y^2 \eta^2 + \gamma_y^2 \eta^2 L_d - \frac{\gamma_y \eta}{8} \right) \mathbb{E}[\|\bar{q}_t\|^2] \\
&\quad + 2\omega \mathbb{E}[\langle \bar{z}_{t+1} - \bar{z}_t, x^*(\bar{y}_{t+1}; \bar{z}_{t+1}) - x^*(y^*(\bar{z}_{t+1}); \bar{z}_t) \rangle] . \tag{20}
\end{aligned}
$$

Moreover, from Eq. (29) in (Yang et al., 2022b), we obtain

$$
2\omega\langle \bar{z}_{t+1} - \bar{z}_t, x^*(\bar{y}_{t+1}; \bar{z}_{t+1}) - x^*(y^*(\bar{z}_{t+1}); \bar{z}_t)\rangle
$$
$$
\leq (\frac{1}{6\gamma_z\eta} + 2C_{x_{yz}^2})\omega\mathbb{E}[\|\bar{z}_{t+1} - \bar{z}_t\|^2] + 6\omega\gamma_z\eta\mathbb{E}[\|x^*(\bar{y}_{t+1}; \bar{z}_{t+1}) - x^*(y^*(\bar{z}_{t+1}); \bar{z}_{t+1})\|^2] . \tag{21}
$$

From the second property in Lemma C.2, we further deduce

$$
\|\nabla_y F(x^*(\bar{y}_t, \bar{z}_t), \bar{y}_t; \bar{z}_t)\|^2 \leq 2\|\nabla_y F(x^*(\bar{y}_t, \bar{z}_t), \bar{y}_t; \bar{z}_t) - \nabla_y F(G(\bar{x}_t), \bar{y}_t; \bar{z}_t)\|^2 + 2\|\nabla_y F(G(\bar{x}_t), \bar{y}_t; \bar{z}_t)\|^2
$$
$$
\leq 2\ell^2\|x^*(\bar{y}_t, \bar{z}_t) - \bar{x}_t\|^2 + 2\|\nabla_y F(G(\bar{x}_t), \bar{y}_t; \bar{z}_t)\|^2
$$
$$
\leq \frac{2\ell^2}{(\omega - \ell)^2}\|\nabla_x F(G(\bar{x}_t), \bar{y}_t; \bar{z}_t)\|^2 + 2\|\nabla_y F(G(\bar{x}_t), \bar{y}_t; \bar{z}_t)\|^2 ,
$$

where the last step follows from the strong convexity of $F(G(x), y; z)$ with respect to $x$ with Eq. (19). Combining the above inequalities with the second and third properties in Lemma C.2 yields:

$$
\mathbb{E}[\|x^*(\bar{y}_{t+1}; \bar{z}_{t+1}) - x^*(y^*(\bar{z}_{t+1}); \bar{z}_{t+1})\|^2] \leq 4(C_{x_z}^2 + C_{x_{yz}^2}^2)\mathbb{E}[\|\bar{z}_{t+1} - \bar{z}_t\|^2]
$$
$$
+ \left(\frac{8\gamma_y^2\eta^2\ell^2 C_{x_{yz}^1}^2}{(\omega - \ell)^2} + \frac{16(1 + \gamma_y^2\eta^2\ell^2 C_{x_{yz}^1}^2 + \gamma_y^2\eta^2\ell^2)\ell^2}{(\omega - \ell)^3\mu}\right)\mathbb{E}[\|\nabla_x F(G(\bar{x}_t), \bar{y}_t; \bar{z}_t)\|^2]
$$
$$
+ \frac{16(1 + \gamma_y^2\eta^2\ell^2 C_{x_{yz}^1}^2 + \gamma_y^2\eta^2\ell^2)}{(\omega - \ell)\mu}\mathbb{E}[\|\nabla_y F(G(\bar{x}_t), \bar{y}_t; \bar{z}_t)\|^2] + 8\gamma_y^2\eta^2 C_{x_{yz}^1}^2\mathbb{E}[\|\nabla_y F(G(\bar{x}_t), \bar{y}_t; \bar{z}_t) - \bar{q}_t\|^2] . \tag{22}
$$

Combining Eq. (20), (21) and (22), we obtain:

$$
\mathbb{E}[\Psi_{t+1}] - \mathbb{E}[\Psi_t] \leq \left(\frac{8\gamma_y\eta\ell^2}{(\omega - \ell)^2} - \frac{\gamma_x\eta}{2}\right)\mathbb{E}[\|\nabla_x F(G(\bar{x}_t), \bar{y}_t; \bar{z}_t)\|^2] - \frac{\gamma_y\eta}{2}\mathbb{E}[\|\nabla_y F(G(\bar{x}_t), \bar{y}_t; \bar{z}_t)\|^2]
$$
$$
+ \left(2C_{x_{yz}^2} + 24\gamma_z\eta(C_{x_z}^2 + C_{x_{yz}^2}^2) - \frac{1}{3\gamma_z\eta}\right)\omega\mathbb{E}[\|\bar{z}_{t+1} - \bar{z}_t\|^2] + \frac{\gamma_x\eta}{2}\mathbb{E}[\|\nabla_x F(G(\bar{x}_t), \bar{y}_t; \bar{z}_t) - \bar{p}_t\|^2]
$$
$$
+ \left(\gamma_y\eta + 48\omega\gamma_z\eta\gamma_y^2\eta^2 C_{x_{yz}^1}^2\right)\mathbb{E}[\|\nabla_y F(G(\bar{x}_t), \bar{y}_t; \bar{z}_t) - \bar{q}_t\|^2]
$$
$$
+ \left(\gamma_y\eta\gamma_x^2\eta^2\ell^2 - \frac{\gamma_x\eta}{4}\right)\mathbb{E}[\|\bar{p}_t\|^2] + \left(\frac{\omega + \ell}{2}\gamma_y^2\eta^2 + \gamma_y^2\eta^2 L_d - \frac{\gamma_y\eta}{8}\right)\mathbb{E}[\|\bar{q}_t\|^2]
$$
$$
+ \left(\frac{48\omega\gamma_z\eta\gamma_y^2\eta^2\ell^2 C_{x_{yz}^1}^2}{(\omega - \ell)^2} + \frac{96\omega\gamma_z\eta(1 + \gamma_y^2\eta^2\ell^2 C_{x_{yz}^1}^2 + \gamma_y^2\eta^2\ell^2)\ell^2}{(\omega - \ell)^3\mu}\right)\mathbb{E}[\|\nabla_x F(G(\bar{x}_t), \bar{y}_t; \bar{z}_t)\|^2]
$$
$$
+ \frac{96\omega\gamma_z\eta(1 + \gamma_y^2\eta^2\ell^2 C_{x_{yz}^1}^2 + \gamma_y^2\eta^2\ell^2)}{(\omega - \ell)\mu}\mathbb{E}[\|\nabla_y F(G(\bar{x}_t), \bar{y}_t; \bar{z}_t)\|^2] .
$$

Since $\gamma_z\eta \leq 1$ and $C_{x_{yz}^1} = \frac{\omega + \ell}{\omega - \ell}$, by setting

$$
\gamma_y = \gamma_x \underbrace{\frac{(\omega - \ell)^2}{64\ell^2}}_{\psi_{\gamma_y} = O(1)} , \quad \gamma_z = \gamma_x \underbrace{\frac{(\omega - \ell)^3\mu}{1024 \times 96\omega\ell^2}}_{\psi_{\gamma_z} = O(1/\kappa)} , \quad \gamma_x \leq \min\left\{\frac{\ell^2}{6\omega(\omega + \ell)^2} , \frac{64\ell}{(\omega - \ell)^2\sqrt{C_{x_{yz}^1}^2 + 1}}\right\} ,
$$

we obtain

$$
\frac{8\gamma_y\eta\ell^2}{(\omega - \ell)^2} \leq \frac{\gamma_x\eta}{8} , \quad \frac{48\omega\gamma_z\eta\gamma_y^2\eta^2 C_{x_{yz}^1}^2\ell^2}{(\omega - \ell)^2} \leq \frac{\gamma_x\eta}{32 \times 16} ,
$$
$$
\frac{96\omega\gamma_z\eta(1 + \gamma_y^2\eta^2\ell^2 C_{x_{yz}^1}^2 + \gamma_y^2\eta^2\ell^2)\ell^2}{(\omega - \ell)^3\mu} \leq \frac{\gamma_x\eta}{32 \times 16} ,
$$
$$
\frac{96\omega\gamma_z\eta(1 + \gamma_y^2\eta^2 L_f^2 C_{x_{yz}^1}^2 + \gamma_y^2\eta^2\ell^2)}{(\omega - \ell)\mu} \leq \frac{\gamma_y\eta}{8} .
$$

This choice ensures the coefficients of $\mathbb{E}[\|\nabla_x F(G(\bar{x}_t), \bar{y}_t; \bar{z}_t)\|^2]$ and $\mathbb{E}[\|\nabla_y F(G(\bar{x}_t), \bar{y}_t; \bar{z}_t)\|^2]$ are $-\frac{\gamma_x \eta}{4}$ and $-\frac{\gamma_y \eta}{4}$, respectively. Moreover, for the coefficient of $\mathbb{E}[\|\nabla_y F(G(\bar{x}_t), \bar{y}_t; \bar{z}_t) - \bar{q}_t\|^2]$, since the second inequality holds above, we derive

$$\gamma_y \eta + 48\omega\gamma_z \eta\gamma_y^2 \eta^2 C_{x_{yz}^1}^2 \leq \gamma_y \eta + \frac{\gamma_x \eta(\omega - \ell)^2}{32 \times 16 L_f^2} \leq \frac{5\gamma_y \eta}{4} .$$

Hence, the proof is complete. $\qquad \square$

### C.2. Federated Properties

We further introduce $C_K^2$ as follows:

$$C_K^2 = \max\left\{ (K+1)C_g^{2(K-1)}(KC_f^2 + C_g^2), (K+1)\ell^2 \right\} .$$

**Lemma C.6.** *Given Assumptions 3.1-3.3, the following inequalities for compositional multi-level functions hold:*

1. *The bounded error between **stochastic estimation gradient** $\nabla_x F_{h,t;t}^{(n)}$ and **estimation gradient** $\nabla_x F_{h,t}^{(n)}$ satisfies:*

$$\mathbb{E}[\|\nabla_x F_{h,t;t}^{(n)} - \nabla_x F_{h,t}^{(n)}\|^2] \leq C_K^2 \sigma^2 .$$

2. *The estimation error between **global true gradient** $\nabla_x F_{g,t}$ and **global estimation gradient** $\nabla_x F_{h,t}$ satisfies:*

$$\mathbb{E}[\|\nabla_x F_{g,t} - \nabla_x F_{h,t}\|^2] \leq K\sum_{k=1}^{K} A_k \mathbb{E}[\|\frac{1}{N}\sum_{n=1}^{N} g_{h,t}^{(n,k)} - \frac{1}{N}\sum_{n=1}^{N} h_t^{(n,k)}\|^2] ,$$

*where $A_k = \left( C_g^{2(K-1)} C_f^2 L_g^2 \left( \sum_{j=k}^{K-1} C_g^{j-k} \right)^2 + C_g^{2K} L_f^2 C_g^{2(K-k)} \right)$.*

3. *For any $k \in \{1, \cdots, K\}$, the estimation error between **true value** $G^{(n,k)}(x_t^{(n)})$ and **estimation value** $h_t^{(n,k)}$ satisfies:*

$$\mathbb{E}[\|G^{(n,k)}(x_t^{(n)}) - h_t^{(n,k)}\|^2] \leq \sum_{j=1}^{k} C_g^{2(k-j)}\mathbb{E}[\|g_{h,t}^{(n,j)} - h_t^{(n,j)}\|^2] .$$

4. *For any $k \in \{1, \cdots, K\}$, the descent error between $h_{t+1}^{(n,k)}$ and $h_t^{(n,k)}$ satisfies*

$$\mathbb{E}[\|h_{t+1}^{(n,k)} - h_t^{(n,k)}\|^2] \leq 2\alpha^2\eta^4 \sum_{j=1}^{k} (2C_g^2)^{k-j}\mathbb{E}[\|g_{h,t}^{(n,j)} - h_t^{(n,j)}\|^2]$$

$$+ (2C_g^2)^k \gamma_x^2 \eta^2 \mathbb{E}[\|p_t^{(n)}\|^2] + 2\alpha^2\eta^4\sigma^2 \sum_{j=1}^{k}(2C_g^2)^{j-1} .$$

*Proof.* First, we have

$$\mathbb{E}[\|\nabla_x F_{h,t;t}^{(n)} - \nabla_x F_{h,t}^{(n)}\|^2] = \mathbb{E}[\|\nabla_x f_{h,t;t}^{(n)} + \omega(x_t^{(n)} - z_t^{(n)}) - \nabla_x f_{h,t}^{(n)} + \omega(x_t^{(n)} - z_t^{(n)})\|^2]$$
$$= \mathbb{E}[\|\nabla_x f_{h,t;t}^{(n)} - \nabla_x f_{h,t}^{(n)}\|^2] \leq (K+1)C_g^{2(K-1)}(KC_f^2 + C_g^2)\sigma^2 .$$

where the last step follows from Lemma B.2, Eq. (28) in (Zhang et al., 2024). From the definition of $C_K^2$, the proof is complete.

Similarly,

$$\mathbb{E}[\|\nabla_x F_{g,t} - \nabla_x F_{h,t}\|^2] = \mathbb{E}[\|\nabla_x f_{g,t} - \nabla_x f_{h,t}\|^2] \leq K\sum_{k=1}^{K} A_k \mathbb{E}[\|\frac{1}{N}\sum_{n=1}^{N} g_{h,t}^{(n,k)} - \frac{1}{N}\sum_{n=1}^{N} h_t^{(n,k)}\|^2] .$$

where the last step follows from Lemma B.2, Eq. (25) in (Zhang et al., 2024).

Lastly, the third property follows directly from Eq. (26), and the final property from Lemma B.5, both in (Zhang et al., 2024). $\qquad \square$

### C.2.1. GRADIENT ESTIMATION ERRORS

**Lemma C.7.** *Given Assumption 3.1-3.3 , we derive **gradient estimation error** regarding $x$:*

$$\mathbb{E}[\|\bar{p}_{t+1} - \frac{1}{N}\sum_{n=1}^{N}\nabla_x F_{h,t+1}^{(n)}\|^2] \leq (1 - \beta_x\eta^2)\mathbb{E}[\|\bar{p}_t - \frac{1}{N}\sum_{n=1}^{N}\nabla_x F_{h,t}^{(n)}\|^2]$$

$$+ 4C_K^2\alpha^2\eta^4\sum_{k=1}^{K}\Big(\sum_{j=k}^{K}(2C_g^2)^{j-k}\Big)\frac{1}{N^2}\sum_{n=1}^{N}\mathbb{E}[\|g_{h,t}^{(n,k)} - h_t^{(n,k)}\|^2] + 4C_K^2\alpha^2\eta^4\sum_{k=1}^{K}\sum_{j=1}^{k}(2C_g^2)^{j-1}\frac{\sigma^2}{N}$$

$$+ 4C_K^2\sum_{k=0}^{K}(2C_g^2)^k\gamma_x^2\eta^2\frac{1}{N}\mathbb{E}[\|\bar{p}_t\|^2] + 4C_K^2\sum_{k=0}^{K}(2C_g^2)^k\gamma_x^2\eta^2\frac{1}{N^2}\sum_{n=1}^{N}\mathbb{E}[\|\bar{p}_t - p_t^{(n)}\|^2]$$

$$+ 4C_K^2\gamma_y^2\eta^2\frac{1}{N}\mathbb{E}[\|\bar{q}_t\|^2] + 4C_K^2\gamma_y^2\eta^2\frac{1}{N^2}\sum_{n=1}^{N}\mathbb{E}[\|\bar{q}_t - q_t^{(n)}\|^2] + 2\beta_x^2\eta^4 C_K^2\frac{\sigma^2}{N}\,.$$

*Proof.* For the $t + 1$-th iteration, the following relation holds:

$$\mathbb{E}[\|\bar{p}_{t+1} - \frac{1}{N}\sum_{n=1}^{N}\nabla_x F_{h,t+1}^{(n)}\|^2]$$

$$= \mathbb{E}\Big[\Big\|\frac{1}{N}\sum_{n=1}^{N}(1 - \beta_x\eta^2)(p_t^{(n)} - \nabla_x F_{h,t;t+1}^{(n)}) + \frac{1}{N}\sum_{n=1}^{N}\nabla_x F_{h,t+1;t+1}^{(n)} - \frac{1}{N}\sum_{n=1}^{N}\nabla_x F_{h,t+1}^{(n)}\Big\|^2\Big]$$

$$= \mathbb{E}\Big[\Big\|\frac{1}{N}\sum_{n=1}^{N}(1 - \beta_x\eta^2)\Big(p_t^{(n)} - \nabla_x F_{h,t}^{(n)}\Big) + \frac{1}{N}\sum_{n=1}^{N}\Big(\nabla_x F_{h,t+1;t+1}^{(n)} - \nabla_x F_{h,t;t+1}^{(n)} + \nabla_x F_{h,t}^{(n)} - \nabla_x F_{h,t+1}^{(n)}\Big)$$

$$+ \beta_x\eta^2\Big(\frac{1}{N}\sum_{n=1}^{N}\nabla_x F_{h,t;t+1}^{(n)} - \frac{1}{N}\sum_{n=1}^{N}\nabla_x F_{h,t}^{(n)}\Big)\Big\|^2\Big]$$

$$\leq (1 - \beta_x\eta^2)^2\mathbb{E}[\|\bar{p}_t - \frac{1}{N}\sum_{n=1}^{N}\nabla_x F_{h,t}^{(n)}\|^2] + 2\frac{1}{N^2}\sum_{n=1}^{N}\mathbb{E}[\|\nabla_x f_{h,t+1;t+1}^{(n)} - \nabla_x f_{h,t;t+1}^{(n)}\|^2]$$

$$+ 2\beta_x^2\eta^4\frac{1}{N^2}\sum_{n=1}^{N}\mathbb{E}[\|\nabla_x F_{h,t;t+1}^{(n)} - \nabla_x F_{h,t}^{(n)}\|^2]\,.$$

where the third step holds due to the following inequality:

$$\mathbb{E}[\|\nabla_x F_{h,t+1;t+1}^{(n)} - \nabla_x F_{h,t;t+1}^{(n)} + \nabla_x F_{h,t}^{(n)} - \nabla_x F_{h,t+1}^{(n)}\|^2]$$

$$= \mathbb{E}[\|\nabla_x f_{h,t+1;t+1}^{(n)} + \omega(x_{t+1}^{(n)} - z_{t+1}^{(n)}) - \nabla_x f_{h,t;t+1}^{(n)} - \omega(x_t^{(n)} - z_t^{(n)}) + \nabla_x f_{h,t}^{(n)} + \omega(x_t^{(n)} - z_t^{(n)})$$

$$- \nabla_x f_{h,t+1}^{(n)} - \omega(x_{t+1}^{(n)} - z_{t+1}^{(n)})\|^2]$$

$$\leq \mathbb{E}[\|\nabla_x f_{h,t+1;t+1}^{(n)} - \nabla_x f_{h,t;t+1}^{(n)}\|^2]\,.$$

We further bound $\mathbb{E}[\|\nabla_x f_{h,t+1;t+1}^{(n)} - \nabla_x f_{h,t;t+1}^{(n)}\|^2]$ as follows:

$$\mathbb{E}[\|\nabla_x f_{h,t+1;t+1}^{(n)} - \nabla_x f_{h,t;t+1}^{(n)}\|^2]$$

$$= \mathbb{E}[\|\nabla g^{(n,1)}(h_{t+1}^{(n,0)};\xi_{t+1}^{(n,1)})\cdots\nabla g^{(n,K)}(h_{t+1}^{(n,K-1)};\xi_{t+1}^{(n,K)})\nabla_x f^{(n)}(h_{t+1}^{(n,K)},y_{t+1}^{(n)};\zeta_{t+1}^{(n)})$$

$$- \nabla g^{(n,1)}(h_t^{(n,0)};\xi_{t+1}^{(n,1)})\cdots\nabla g^{(n,K)}(h_t^{(n,K-1)};\xi_{t+1}^{(n,K)})\nabla_x f^{(n)}(h_t^{(n,K)},y_t^{(n)};\zeta_{t+1}^{(n)})\|^2]$$

$$\leq (K+1)C_g^{2K}L_f^2\mathbb{E}[\|h_{t+1}^{(n,K)} - h_t^{(n,K)}\|^2] + (K+1)C_g^{2K}L_f^2\mathbb{E}[\|y_{t+1}^{(n)} - y_t^{(n)}\|^2]$$

$$+ (K+1)C_g^{2(K-1)}C_f^2L_g^2\mathbb{E}[\|h_{t+1}^{(n,K-1)} - h_t^{(n,K-1)}\|^2] + \cdots + (K+1)C_g^{2(K-1)}C_f^2L_g^2\mathbb{E}[\|h_{t+1}^{(n,1)} - h_t^{(n,1)}\|^2]$$

$$+ C_g^{2(K-1)}C_f^2L_g^2\mathbb{E}[\|x_{t+1}^{(n)} - x_t^{(n)}\|^2]$$

$$\leq C_K^2 \sum_{k=1}^{K} \mathbb{E}[\|h_{t+1}^{(n,k)} - h_t^{(n,k)}\|^2] + C_K^2 \mathbb{E}[\|x_{t+1}^{(n)} - x_t^{(n)}\|^2] + C_K^2 \mathbb{E}[\|y_{t+1}^{(n)} - y_t^{(n)}\|^2] . \tag{23}$$

In addition, applying the update rule on $x$ and $y$ yields:

$$\mathbb{E}[\|x_{t+1}^{(n)} - x_t^{(n)}\|^2] \leq \gamma_x^2 \eta^2 \mathbb{E}[\|p_t^{(n)}\|^2] \leq 2\gamma_x^2 \eta^2 \mathbb{E}[\|\bar{p}_t - p_t^{(n)}\|^2] + 2\gamma_x^2 \eta^2 \mathbb{E}[\|\bar{p}_t\|^2] .$$

$$\mathbb{E}[\|y_{t+1}^{(n)} - y_t^{(n)}\|^2] \leq \gamma_y^2 \eta^2 \mathbb{E}[\|q_t^{(n)}\|^2] \leq 2\gamma_y^2 \eta^2 \mathbb{E}[\|\bar{q}_t - q_t^{(n)}\|^2] + 2\gamma_y^2 \eta^2 \mathbb{E}[\|\bar{q}_t\|^2] .$$

Combining the above inequalities with the first and last properties in Lemma C.6 completes the proof. $\qquad\square$

**Lemma C.8.** *Given Assumption 3.1-3.3, we derive **gradient estimation error** regarding y:*

$$\mathbb{E}[\|\bar{q}_{t+1} - \frac{1}{N}\sum_{n=1}^{N}\nabla_y F_{h,t+1}^{(n)}\|^2] \leq (1 - \beta_y\eta^2)\mathbb{E}[\|\bar{q}_t - \frac{1}{N}\sum_{n=1}^{N}\nabla_y F_{h,t}^{(n)}\|^2]$$

$$+ 4\alpha^2\eta^4 L_f^2 \sum_{k=1}^{K}(2C_g^2)^{K-k}\frac{1}{N^2}\sum_{n=1}^{N}\mathbb{E}[\|g_{h,t}^{(n,k)} - h_t^{(n,k)}\|^2] + 4\alpha^2\eta^4 L_f^2 \sum_{k=1}^{K}(2C_g^2)^{k-1}\frac{\sigma^2}{N}$$

$$+ 4L_f^2(2C_g^2)^K\gamma_x^2\eta^2\frac{1}{N}\mathbb{E}[\|\bar{p}_t\|^2] + 4L_f^2(2C_g^2)^K\gamma_x^2\eta^2\frac{1}{N^2}\sum_{n=1}^{N}\mathbb{E}[\|\bar{p}_t - p_t^{(n)}\|^2]$$

$$+ 4L_f^2\gamma_y^2\eta^2\frac{1}{N}\mathbb{E}[\|\bar{q}_t\|^2] + 4L_f^2\gamma_y^2\eta^2\frac{1}{N^2}\sum_{n=1}^{N}\mathbb{E}[\|\bar{q}_t - q_t^{(n)}\|^2] + 2\beta_y^2\eta^4\frac{\sigma^2}{N} .$$

*Proof.* The proof of this lemma follows the same arguments as Lemma C.7 and is therefore omitted. $\qquad\square$

**Lemma C.9.** *Given Assumptions 3.1-3.3 , we derive **global estimation error** regarding h:*

$$\sum_{k=1}^{K}\theta_k\mathbb{E}[\|\frac{1}{N}\sum_{n=1}^{N}g_{h,t+1}^{(n,k)} - \frac{1}{N}\sum_{n=1}^{N}h_{t+1}^{(n,k)}\|^2] \leq (1 - \alpha\eta^2)\sum_{k=1}^{K}\theta_k\mathbb{E}[\|\frac{1}{N}\sum_{n=1}^{N}g_{h,t}^{(n,k)} - \frac{1}{N}\sum_{n=1}^{N}h_t^{(n,k)}\|^2]$$

$$+ 2\alpha^2\eta^4 \sum_{k=1}^{K}\Big(\sum_{j=k+1}^{K}\theta_j(2C_g^2)^{j-k}\Big)\mathbb{E}[\|\frac{1}{N}\sum_{n=1}^{N}g_{h,t}^{(n,k)} - \frac{1}{N}\sum_{n=1}^{N}h_t^{(n,k)}\|^2] + 2\alpha^2\eta^4 \sum_{k=1}^{K}\theta_k \sum_{j=0}^{k-1}(2C_g^2)^j\frac{\sigma^2}{N}$$

$$+ 2\gamma_x^2\eta^2 \sum_{k=1}^{K}\theta_k(2C_g^2)^k\frac{1}{N^2}\sum_{n=1}^{N}\mathbb{E}[\|\bar{p}_t - p_t^{(n)}\|^2] + 2\gamma_x^2\eta^2 \sum_{k=1}^{K}\theta_k(2C_g^2)^k\frac{1}{N}\mathbb{E}[\|\bar{p}_t\|^2] .$$

*Proof.* This lemma can be directly obtained from Lemma B.4 in (Zhang et al., 2024). $\qquad\square$

**Lemma C.10.** *Given Assumptions 3.1-3.3 , we derive **local estimation error** regarding h:*

$$\sum_{k=1}^{K}\nu_k\mathbb{E}[\|g_{h,t+1}^{(n,k)} - h_{t+1}^{(n,k)}\|^2] \leq (1 - \alpha\eta^2)\sum_{k=1}^{K}\nu_k\mathbb{E}[\|g_{h,t}^{(n,k)} - h_t^{(n,k)}\|^2]$$

$$+ 2\alpha^2\eta^4 \sum_{k=1}^{K}\Big(\sum_{j=k+1}^{K}\nu_j(2C_g^2)^{j-k}\Big)\mathbb{E}[\|g_{h,t}^{(n,k)} - h_t^{(n,k)}\|^2] + 2\gamma_x^2\eta^2 \sum_{k=1}^{K}\nu_k(2C_g^2)^k\mathbb{E}[\|\bar{p}_t - p_t^{(n)}\|^2]$$

$$+ 2\gamma_x^2\eta^2 \sum_{k=1}^{K}\nu_k(2C_g^2)^k\mathbb{E}[\|\bar{p}_t\|^2] + 2\alpha^2\eta^4\sigma^2 \sum_{k=1}^{K}\nu_k \sum_{j=0}^{k-1}(2C_g^2)^j .$$

*Proof.* This lemma can be directly obtained from Lemma B.4 in (Zhang et al., 2024). $\qquad\square$

### C.2.2. CONSENSUS ERRORS

**Lemma C.11.** *Given Assumption 3.1-3.3, we derive* **consensus error** *regarding $x$:*

$$\frac{1}{N}\sum_{n=1}^{N}\mathbb{E}[\|p_{t+1}^{(n)} - \bar{p}_{t+1}\|^2] \leq (1 + \frac{1}{\tau})\frac{1}{N}\sum_{n=1}^{N}\mathbb{E}[\|p_t^{(n)} - \bar{p}_t\|^2] + 24\tau\alpha^2\eta^4 C_K^2 \sum_{k=1}^{K}\sum_{j=1}^{k}(2C_g^2)^{j-1}\sigma^2 + 24\tau\beta_x^2\eta^4 C_K^2\sigma^2$$

$$+ 24\tau\alpha^2\eta^4 C_K^2 \sum_{k=1}^{K}\Big(\sum_{j=k}^{K}(2C_g^2)^{j-k}\Big)\frac{1}{N}\sum_{n=1}^{N}\mathbb{E}[\|g_{h,t}^{(n,k)} - h_t^{(n,k)}\|^2] + 24\tau\gamma_x^2\eta^2\Big(C_K^2\sum_{k=0}^{K}(2C_g^2)^k + \omega^2\Big)\mathbb{E}[\|\bar{p}_t\|^2]$$

$$+ 24\tau\gamma_x^2\eta^2\Big(C_K^2\sum_{k=0}^{K}(2C_g^2)^k + \omega^2\Big)\frac{1}{N}\sum_{n=1}^{N}\mathbb{E}[\|\bar{p}_t - p_t^{(n)}\|^2] + 24\tau\gamma_y^2\eta^2 C_K^2\mathbb{E}[\|\bar{q}_t\|^2]$$

$$+ 24\tau\gamma_y^2\eta^2 C_K^2 \frac{1}{N}\sum_{n=1}^{N}\mathbb{E}[\|q_t^{(n)} - \bar{q}_t\|^2] + 36\tau\omega^2\mathbb{E}[\|\bar{z}_{t+1} - \bar{z}_t\|^2] + 36\tau^2\omega^2\gamma_z^2\gamma_x^2\eta^4 \frac{1}{N}\sum_{n=1}^{N}\sum_{\hat{t}=s_t\tau}^{t}\mathbb{E}[\|p_{\hat{t}}^{(n)} - \bar{p}_{\hat{t}}\|^2]$$

$$+ 48\tau\beta_x^2\eta^4 C_K^2 \sum_{k=1}^{K}\frac{1}{N}\sum_{n=1}^{N}\mathbb{E}[\|h_t^{(n,k)} - \bar{h}_t\|^2] + 12\tau\beta_x^2\eta^4(\omega^2 + 4C_K^2)\frac{1}{N}\sum_{n=1}^{N}\mathbb{E}[\|x_t^{(n)} - \bar{x}_t\|^2]$$

$$+ 48\tau\beta_x^2\eta^4 C_K^2 \frac{1}{N}\sum_{n=1}^{N}\mathbb{E}[\|y_t^{(n)} - \bar{y}_t\|^2] + 12\tau\omega^2(\beta_x^2\eta^4 + 3\gamma_z^2\eta^2)\frac{1}{N}\sum_{n=1}^{N}\mathbb{E}[\|z_t^{(n)} - \bar{z}_t\|^2].$$

*Proof.* For the $t + 1$-th iteration, the following relation holds:

$$\frac{1}{N}\sum_{n=1}^{N}\mathbb{E}[\|p_{t+1}^{(n)} - \bar{p}_{t+1}\|^2] = \frac{1}{N}\sum_{n=1}^{N}\mathbb{E}\Big[\Big\|\nabla_x F_{h,t+1;t+1}^{(n)} + (1 - \beta_x\eta^2)(p_t^{(n)} - \nabla_x F_{h,t;t+1}^{(n)}) - \frac{1}{N}\sum_{n'=1}^{N}\nabla_x F_{h,t+1;t+1}^{(n')}$$

$$+ (1 - \beta_x\eta^2)(\bar{p}_t - \frac{1}{N}\sum_{n'=1}^{N}\nabla_x F_{h,t;t+1}^{(n')})\Big\|^2\Big]$$

$$\leq (1 - \beta_x\eta^2)^2(1 + \frac{1}{\tau})\frac{1}{N}\sum_{n=1}^{N}\mathbb{E}[\|p_t^{(n)} - \bar{p}_t\|^2] + 2(1 + \tau)\frac{1}{N}\sum_{n=1}^{N}\mathbb{E}[\|\nabla_x F_{h,t+1;t+1}^{(n)} - \nabla_x F_{h,t;t+1}^{(n)}\|^2]$$

$$+ 2\beta_x^2\eta^4(1 + \tau)\frac{1}{N}\sum_{n=1}^{N}\mathbb{E}[\|\nabla_x F_{h,t;t+1}^{(n)} - \frac{1}{N}\sum_{n'=1}^{N}\nabla_x F_{h,t;t+1}^{(n')}\|^2]. \tag{24}$$

For the second term in Eq. (24), we can further establish:

$$\frac{1}{N}\sum_{n=1}^{N}\mathbb{E}[\|\nabla_x F_{h,t+1;t+1}^{(n)} - \nabla_x F_{h,t;t+1}^{(n)}\|^2]$$

$$= \frac{1}{N}\sum_{n=1}^{N}\mathbb{E}[\|\nabla_x f_{h,t+1;t+1}^{(n)} + \omega(x_{t+1}^{(n)} - z_{t+1}^{(n)}) - \nabla_x f_{h,t;t+1}^{(n)} - \omega(x_t^{(n)} - z_t^{(n)})\|^2]$$

$$\leq 3\frac{1}{N}\sum_{n=1}^{N}\mathbb{E}[\|\nabla_x f_{h,t+1;t+1}^{(n)} - \nabla_x f_{h,t;t+1}^{(n)}\|^2] + 3\omega^2\frac{1}{N}\sum_{n=1}^{N}\mathbb{E}[\|x_{t+1}^{(n)} - x_t^{(n)}\|^2] + 3\omega^2\frac{1}{N}\sum_{n=1}^{N}\mathbb{E}[\|z_{t+1}^{(n)} - z_t^{(n)}\|^2]$$

$$\leq 3C_K^2 \sum_{k=1}^{K}\frac{1}{N}\sum_{n=1}^{N}\mathbb{E}[\|h_{t+1}^{(n,k)} - h_t^{(n,k)}\|^2] + 3(C_K^2 + \omega^2)\frac{1}{N}\sum_{n=1}^{N}\mathbb{E}[\|x_{t+1}^{(n)} - x_t^{(n)}\|^2] + 3C_K^2\frac{1}{N}\sum_{n=1}^{N}\mathbb{E}[\|y_{t+1}^{(n)} - y_t^{(n)}\|^2]$$

$$+ 3\omega^2\frac{1}{N}\sum_{n=1}^{N}\mathbb{E}[\|z_{t+1}^{(n)} - z_t^{(n)}\|^2],$$

where the last step follows directly from Eq. (23). Moreover, regarding the auxiliary variable $z$, we derive the relation:

$$\frac{1}{N} \sum_{n=1}^{N} \mathbb{E}[\|z_{t+1}^{(n)} - z_t^{(n)}\|^2] = \gamma_z^2 \eta^2 \frac{1}{N} \sum_{n=1}^{N} \mathbb{E}[\|x_{t+1}^{(n)} - z_t^{(n)}\|^2]$$

$$\leq 3\gamma_z^2 \eta^2 \frac{1}{N} \sum_{n=1}^{N} \mathbb{E}[\|x_{t+1}^{(n)} - \bar{x}_{t+1}\|^2] + 3\gamma_z^2 \eta^2 \mathbb{E}[\|\bar{x}_{t+1} - \bar{z}_t\|^2] + 3\gamma_z^2 \eta^2 \frac{1}{N} \sum_{n=1}^{N} \mathbb{E}[\|\bar{z}_t - z_t^{(n)}\|^2]$$

$$= 3\gamma_z^2 \eta^2 \frac{1}{N} \sum_{n=1}^{N} \mathbb{E}[\|x_{t+1}^{(n)} - \bar{x}_{t+1}\|^2] + 3\mathbb{E}[\|\bar{z}_{t+1} - \bar{z}_t\|^2] + 3\gamma_z^2 \eta^2 \frac{1}{N} \sum_{n=1}^{N} \mathbb{E}[\|\bar{z}_t - z_t^{(n)}\|^2] , \tag{25}$$

and

$$\frac{1}{N} \sum_{n=1}^{N} \mathbb{E}[\|x_{t+1}^{(n)} - \bar{x}_{t+1}\|^2] = \frac{1}{N} \sum_{n=1}^{N} \mathbb{E}[\|x_{s_t\tau}^{(n)} - \gamma_x \eta \sum_{t'=s_t\tau}^{t} p_{t'}^{(n)} - \bar{x}_{s_t\tau} + \gamma_x \eta \sum_{t'=s_t\tau}^{t} \bar{p}_{t'}\|^2]$$

$$\leq \tau \gamma_x^2 \eta^2 \frac{1}{N} \sum_{n=1}^{N} \sum_{\hat{t}=s_t\tau}^{t} \mathbb{E}[\|p_{\hat{t}}^{(n)} - \bar{p}_{\hat{t}}\|^2] ,$$

where $s_t$ denotes the index of the latest communication round before iteration $t$. Moreover, the consensus error at iteration $s_t\tau$ is zero, i.e., $\mathbb{E}[\|p_{s_t\tau}^{(n)} - \bar{p}_{s_t\tau}\|^2] = 0$.

For the last term in Eq. (24), it follows that

$$\frac{1}{N} \sum_{n=1}^{N} \mathbb{E}[\|\nabla_x F_{h,t;t+1}^{(n)} - \frac{1}{N} \sum_{n'=1}^{N} \nabla_x F_{h,t;t+1}^{(n')}\|^2]$$

$$= \frac{1}{N} \sum_{n=1}^{N} \mathbb{E}\left[\left\|\nabla_x f_{h,t;t+1}^{(n)} + \omega(x_t^{(n)} - z_t^{(n)}) - \frac{1}{N} \sum_{n'=1}^{N} \left(\nabla_x f_{h,t;t+1}^{(n')} + \omega(x_t^{(n')} - z_t^{(n')})\right)\right\|^2\right]$$

$$\leq 3\frac{1}{N} \sum_{n=1}^{N} \mathbb{E}[\|\nabla_x f_{h,t;t+1}^{(n)} - \frac{1}{N} \sum_{n'=1}^{N} \nabla_x f_{h,t;t+1}^{(n')}\|^2] + 3\omega^2 \frac{1}{N} \sum_{n=1}^{N} \mathbb{E}[\|x_t^{(n)} - \bar{x}_t\|^2] + 3\omega^2 \frac{1}{N} \sum_{n=1}^{N} \mathbb{E}[\|z_t^{(n)} - \bar{z}_t\|^2]$$

$$\leq 6C_K^2\sigma^2 + 12C_K^2 \sum_{k=1}^{K} \frac{1}{N} \sum_{n=1}^{N} \mathbb{E}[\|h_t^{(n,k)} - \bar{h}_t\|^2] + 3(\omega^2 + 4C_K^2)\frac{1}{N} \sum_{n=1}^{N} \mathbb{E}[\|x_t^{(n)} - \bar{x}_t\|^2] + 12C_K^2 \frac{1}{N} \sum_{n=1}^{N} \mathbb{E}[\|y_t^{(n)} - \bar{y}_t\|^2]$$

$$+ 3\omega^2 \frac{1}{N} \sum_{n=1}^{N} \mathbb{E}[\|z_t^{(n)} - \bar{z}_t\|^2] ,$$

where the last step holds due to the following inequality:

$$\frac{1}{N} \sum_{n=1}^{N} \mathbb{E}[\|\nabla_x f_{h,t;t+1}^{(n)} - \frac{1}{N} \sum_{n'=1}^{N} \nabla_x f_{h,t;t+1}^{(n')}\|^2]$$

$$= \frac{1}{N} \sum_{n=1}^{N} \mathbb{E}[\|\nabla_x f_{h,t;t+1}^{(n)} - \nabla_x f_{h,t}^{(n)} + \nabla_x f_{h,t}^{(n)} - \frac{1}{N} \sum_{n'=1}^{N} \nabla_x f_{h,t}^{(n')} + \frac{1}{N} \sum_{n'=1}^{N} \nabla_x f_{h,t}^{(n')} - \frac{1}{N} \sum_{n'=1}^{N} \nabla_x f_{h,t;t+1}^{(n')}\|^2]$$

$$\leq 2C_K^2\sigma^2 + 4C_K^2 \sum_{k=1}^{K} \frac{1}{N} \sum_{n=1}^{N} \mathbb{E}[\|h_t^{(n,k)} - \bar{h}_t^{(k)}\|^2] + 4C_K^2 \frac{1}{N} \sum_{n=1}^{N} \mathbb{E}[\|x_t^{(n)} - \bar{x}_t\|^2] + 4C_K^2 \frac{1}{N} \sum_{n=1}^{N} \mathbb{E}[\|y_t^{(n)} - \bar{y}_t\|^2] .$$

Finally, by using the inequality $1 + \tau < 2\tau$ together with the update rules of $x$ and $y$, and applying the last property in Lemma C.6, the proof is complete. □

**Lemma C.12.** *Given Assumption 3.1-3.3, we derive* **consensus error** *regarding $y$:*

$$\frac{1}{N} \sum_{n=1}^{N} \mathbb{E}[\|q_{t+1}^{(n)} - \bar{q}_{t+1}\|^2] \leq (1 + \frac{1}{\tau})\frac{1}{N} \sum_{n=1}^{N} \mathbb{E}[\|q_t^{(n)} - \bar{q}_t\|^2] + 8\tau\alpha^2\eta^4 L_f^2 \sum_{k=1}^{K} (2C_g^2)^{K-k} \frac{1}{N} \sum_{n=1}^{N} \mathbb{E}[\|g_{h,t}^{(n,k)} - h_t^{(n,k)}\|^2]$$

$$+ 8\tau\gamma_x^2\eta^2 L_f^2 (2C_g^2)^K \mathbb{E}[\|\bar{p}_t\|^2] + 8\tau\gamma_x^2\eta^2 L_f^2 (2C_g^2)^K \frac{1}{N}\sum_{n=1}^N \mathbb{E}[\|\bar{p}_t - p_t^{(n)}\|^2] + 8\tau\alpha^2\eta^4 L_f^2 \sum_{k=1}^K (2C_g^2)^{k-1}\sigma^2$$

$$+ 8\tau\gamma_y^2\eta^2 L_f^2 \frac{1}{N}\sum_{n=1}^N \mathbb{E}[\|q_t^{(n)} - \bar{q}_t\|^2] + 8\tau\gamma_y^2\eta^2 L_f^2 \mathbb{E}[\|\bar{q}_t\|^2] + 16\tau\beta_y^2\eta^4 L_f^2 \frac{1}{N}\sum_{n=1}^N \mathbb{E}[\|y_t^{(n)} - \bar{y}_t\|^2]$$

$$+ 16\tau\beta_y^2\eta^4 L_f^2 \frac{1}{N}\sum_{n=1}^N \mathbb{E}[\|h_t^{(n,K)} - \bar{h}_t^{(K)}\|^2] + 8\tau\beta_y^2\eta^4\sigma^2 .$$

*Proof.* The proof of this lemma follows the same arguments as Lemma C.11 and is therefore omitted. □

**Lemma C.13.** *Given Assumption 3.1-3.3, we derive* **consensus error** *regarding $h$:*

$$\sum_{k=1}^K \frac{1}{N}\sum_{n=1}^N \mathbb{E}[\|h_{t+1}^{(n,k)} - \bar{h}_{t+1}^{(k)}\|^2] \le \left(1 + \frac{1}{\tau}\right)\sum_{k=1}^K \frac{1}{N}\sum_{n=1}^N \mathbb{E}[\|h_t^{(n,k)} - \bar{h}_t^{(k)}\|^2]$$

$$+ 16\tau\alpha^2\eta^4 C_g^2 \sum_{k=1}^K \frac{1}{N}\sum_{n=1}^N \mathbb{E}[\|h_t^{(n,k)} - \bar{h}_t^{(k)}\|^2] + 16\tau\alpha^2\eta^4 C_g^2 \frac{1}{N}\sum_{n=1}^N \mathbb{E}[\|x_t^{(n)} - \bar{x}_t\|^2]$$

$$+ 4\tau\alpha^2\eta^4 \sum_{k=1}^K \left(\sum_{j=k+1}^K (2C_g^2)^{j-k}\right)\frac{1}{N}\sum_{n=1}^N \mathbb{E}[\|g_{h,t}^{(n,k)} - h_t^{(n,k)}\|^2] + 4\tau\gamma_x^2\eta^2 \sum_{k=1}^K (2C_g^2)^k \mathbb{E}[\|\bar{p}_t\|^2]$$

$$+ 4\tau\gamma_x^2\eta^2 \sum_{k=1}^K (2C_g^2)^k \frac{1}{N}\sum_{n=1}^N \mathbb{E}[\|\bar{p}_t - p_t^{(n)}\|^2] + 4\tau\alpha^2\eta^4 \sum_{k=1}^K \sum_{j=1}^k (2C_g^2)^{j-1}\sigma^2 + 8\tau\alpha^2\eta^4\sigma^2 K .$$

*Proof.* The proof of this lemma follows the same arguments as Lemma C.11 and is therefore omitted. □

**Lemma C.14.** *Given Assumption 3.1-3.3, by setting*

$$C_{K,\omega}^2 = (C_K^2 + 1)\sum_{k=0}^K (2C_g^2)^k + \omega^2 , \quad \eta \le \frac{1}{400\tau C_{K,\omega}} , \quad \{\gamma_x, \gamma_y\} \le 10 ,$$

$$\beta_x = \frac{C_{K,\omega}^2}{N} , \quad \beta_y = \frac{C_{K,\omega}^2}{N} , \quad \alpha = \frac{C_{K,\omega}^2}{N} , \tag{26}$$

*we derive* **overall consensus error***:*

$$\frac{1}{T}\sum_{t=0}^{T-1}\frac{1}{N}\sum_{n=1}^N \left(\mathbb{E}[\|p_t^{(n)} - \bar{p}_t\|^2] + \mathbb{E}[\|q_t^{(n)} - \bar{q}_t\|^2] + \sum_{k=1}^K \mathbb{E}[\|h_t^{(n,k)} - \bar{h}_t^{(k)}\|^2]\right)$$

$$\le 6\tau E_p \frac{1}{T}\sum_{t=0}^{T-1}\mathbb{E}[\|\bar{p}_t\|^2] + 6\tau E_q \frac{1}{T}\sum_{t=0}^{T-1}\mathbb{E}[\|\bar{q}_t\|^2] + 216\tau^2\omega^2 \frac{1}{T}\sum_{t=0}^{T-1}\mathbb{E}[\|\bar{z}_{t+1} - \bar{z}_t\|^2]$$

$$+ 6\tau E_g \frac{1}{T}\sum_{t=0}^{T-1}\frac{1}{N}\sum_{n=1}^N \mathbb{E}[\|g_{h,t}^{(n,k)} - h_t^{(n,k)}\|^2] + 6\tau E , \tag{27}$$

*where the coefficients of each term are defined as Eq. (29),*

*Proof.* To begin with, by combining Lemmas C.11-C.13, and the following inequality

$$\frac{1}{N}\sum_{n=1}^N \mathbb{E}[\|z_t^{(n)} - \bar{z}_t\|^2] = \frac{1}{N}\sum_{n=1}^N \mathbb{E}[\|z_{t-1}^{(n)} + \gamma_z\eta(x_t^{(n)} - z_{t-1}^{(n)}) - \bar{z}_{t-1} - \gamma_z\eta(\bar{x}_t - \bar{z}_{t-1})\|^2]$$

$$= \frac{1}{N}\sum_{n=1}^N \mathbb{E}[\|(1 - \gamma_z\eta)(z_{t-1}^{(n)} - \bar{z}_{t-1}) + \gamma_z\eta(x_t^{(n)} - \bar{x}_t)\|^2]$$

$$\leq (1 + \frac{\gamma_z \eta}{1 - \gamma_z \eta})(1 - \gamma_z \eta)^2 \frac{1}{N} \sum_{n=1}^{N} \mathbb{E}[\|z_{t-1}^{(n)} - \bar{z}_{t-1}\|^2] + (1 + \frac{1 - \gamma_z \eta}{\gamma_z \eta}) \gamma_z^2 \eta^2 \frac{1}{N} \sum_{n=1}^{N} \mathbb{E}[\|x_t^{(n)} - \bar{x}_t\|^2]$$

$$= (1 - \gamma_z \eta) \frac{1}{N} \sum_{n=1}^{N} \mathbb{E}[\|z_{t-1}^{(n)} - \bar{z}_{t-1}\|^2] + \gamma_z \eta \frac{1}{N} \sum_{n=1}^{N} \mathbb{E}[\|x_t^{(n)} - \bar{x}_t\|^2]$$

$$\leq \sum_{\hat{t}=s_t\tau}^{t} (1 - \gamma_z \eta)^{t-\hat{t}} \gamma_z \eta \frac{1}{N} \sum_{n=1}^{N} \mathbb{E}[\|x_{\hat{t}}^{(n)} - \bar{x}_{\hat{t}}\|^2]$$

$$\leq \gamma_z \eta \sum_{\hat{t}=s_t\tau}^{t} \frac{1}{N} \sum_{n=1}^{N} \mathbb{E}[\|x_{\hat{t}}^{(n)} - \bar{x}_{\hat{t}}\|^2] \,, \tag{28}$$

we obtain

$$\frac{1}{N} \sum_{n=1}^{N} \mathbb{E}[\|p_{t+1}^{(n)} - \bar{p}_{t+1}\|^2] + \frac{1}{N} \sum_{n=1}^{N} \mathbb{E}[\|q_{t+1}^{(n)} - \bar{q}_{t+1}\|^2] + \frac{1}{N} \sum_{n=1}^{N} \sum_{k=1}^{K} \mathbb{E}[\|h_{t+1}^{(n,k)} - \bar{h}_{t+1}^{(k)}\|^2]$$

$$\leq (1 + \frac{1}{\tau} + E_p) \frac{1}{N} \sum_{n=1}^{N} \mathbb{E}[\|p_t^{(n)} - \bar{p}_t\|^2] + 36\tau^2 \omega^2 \gamma_z^2 \gamma_x^2 \eta^4 \frac{1}{N} \sum_{n=1}^{N} \sum_{\hat{t}=s_t\tau}^{t} \mathbb{E}[\|p_{\hat{t}}^{(n)} - \bar{p}_{\hat{t}}\|^2] + E_p \mathbb{E}[\|\bar{p}_t\|^2]$$

$$+ (1 + \frac{1}{\tau} + E_q) \frac{1}{N} \sum_{n=1}^{N} \mathbb{E}[\|q_t^{(n)} - \bar{q}_t\|^2] + E_q \mathbb{E}[\|\bar{q}_t\|^2] + (1 + \frac{1}{\tau} + E_h) \sum_{k=1}^{K} \frac{1}{N} \sum_{n=1}^{N} \mathbb{E}[\|h_t^{(n,k)} - \bar{h}_t^{(k)}\|^2]$$

$$+ 36\tau\omega^2 \mathbb{E}[\|\bar{z}_{t+1} - \bar{z}_t\|^2] + 12\tau\omega^2 \gamma_z \eta (\beta_x^2 \eta^4 + 3\gamma_z^2 \eta^2) \sum_{\hat{t}=s_t\tau}^{t} \frac{1}{N} \sum_{n=1}^{N} \mathbb{E}[\|x_{\hat{t}}^{(n)} - \bar{x}_{\hat{t}}\|^2]$$

$$+ \left(12\tau\beta_x^2 \eta^4 (\omega^2 + 4C_K^2) + 16\tau\alpha^2 \eta^4 C_g^2\right) \frac{1}{N} \sum_{n=1}^{N} \mathbb{E}[\|x_t^{(n)} - \bar{x}_t\|^2]$$

$$+ \left(48\tau\beta_x^2 \eta^4 C_K^2 + 16\tau\beta_y^2 \eta^4 L_f^2\right) \frac{1}{N} \sum_{n=1}^{N} \mathbb{E}[\|y_t^{(n)} - \bar{y}_t\|^2] + E_g \frac{1}{N} \sum_{n=1}^{N} \mathbb{E}[\|g_{h,t}^{(n,k)} - h_t^{(n,k)}\|^2] + E \,.$$

where, for readability, we use the following notation for the coefficients of each term and the constant term:

$$E_p = 24\tau\gamma_x^2 \eta^2 \left(C_K^2 \sum_{k=0}^{K} (2C_g^2)^k + \omega^2\right) + 8\tau\gamma_x^2 \eta^2 L_f^2 (2C_g^2)^K + 4\tau\gamma_x^2 \eta^2 \sum_{k=1}^{K} (2C_g^2)^k \,,$$

$$E_q = 24\tau\gamma_y^2 \eta^2 C_K^2 + 8\tau\gamma_y^2 \eta^2 L_f^2 \,,$$

$$E_h = 48\tau\beta_x^2 \eta^4 C_K^2 + 16\tau\beta_y^2 \eta^4 L_f^2 + 16\tau\alpha^2 \eta^4 C_g^2 \,,$$

$$E_g = \sum_{k=1}^{K} \alpha^2 \eta^4 \left(24\tau C_K^2 \sum_{j=k}^{K} (2C_g^2)^{j-k} + 4\tau \sum_{j=k}^{K} (2C_g^2)^{j-k} + 8\tau L_f^2 (2C_g^2)^{K-k}\right) \,,$$

$$E = 4\tau\alpha^2 \eta^4 (8C_K^2 + 1) \sum_{k=1}^{K} \sum_{j=1}^{k} (2C_g^2)^{j-1} \sigma^2 + 24\tau\beta_x^2 \eta^4 C_K^2 \sigma^2 + 8\tau\beta_y^2 \eta^4 \sigma^2 + 8\tau\alpha^2 \eta^4 \sigma^2 K \,. \tag{29}$$

By setting $\eta \leq \frac{1}{400\tau C_{K,\omega}}$ , $\gamma_x \leq 10$, we obtain

$$E_p \leq 24\tau\gamma_x^2 \eta^2 \left(C_K^2 \sum_{k=0}^{K} (2C_g^2)^k + \omega^2\right) + 8\tau\gamma_x^2 \eta^2 L_f^2 (2C_g^2)^K + 4\tau\gamma_x^2 \eta^2 \sum_{k=1}^{K} (2C_g^2)^k$$

$$\leq \frac{2400\tau\left(C_K^2 \sum_{k=0}^{K} (2C_g^2)^k + \omega^2\right)}{400^2 \tau^2 \left((C_K^2 + 1) \sum_{k=0}^{K} (2C_g^2)^k + \omega^2\right)} + \frac{800\tau L_f^2 (2C_g^2)^K + 400\tau \sum_{k=1}^{K} (2C_g^2)^k}{400^2 \tau^2 \left((C_K^2 + 1) \sum_{k=0}^{K} (2C_g^2)^k + \omega^2\right)}$$

$$\leq \frac{6}{400\tau} + \frac{3}{400\tau} \leq \frac{9}{400\tau} \leq \frac{2}{25\tau} \ .$$

Similarly, by setting $\gamma_y \leq 10$, we obtain

$$E_q \leq 24\tau\gamma_y^2\eta^2 C_K^2 + 8\tau\gamma_y^2\eta^2 L_f^2 \leq \frac{2400\tau C_K^2 + 800\tau L_f^2}{400^2\tau^2 C_{K,\omega}^2} \leq \frac{8}{400\tau} \leq \frac{2}{25\tau} \ .$$

By setting $\{\beta_x, \beta_y, \alpha\} \leq 10$ with $\beta_x\eta^2 \leq 1$, $\beta_y\eta^2 \leq 1$, $\alpha\eta^2 \leq 1$, we obtain

$$E_h \leq 48\tau\beta_x^2\eta^4 C_K^2 + 16\tau\beta_y^2\eta^4 L_f^2 + 16\tau\alpha^2\eta^4 C_g^2 \leq \frac{(48\tau C_K^2 + 16\tau L_f^2 + 16\tau C_g^2) \times 10}{400^2\tau^2 C_{K,\omega}^2} \leq \frac{2}{400\tau} \leq \frac{2}{25\tau} \ .$$

Therefore, we obtain:

$$\frac{1}{N}\sum_{n=1}^{N}\mathbb{E}[\|p_{t+1}^{(n)} - \bar{p}_{t+1}\|^2] + \frac{1}{N}\sum_{n=1}^{N}\mathbb{E}[\|q_{t+1}^{(n)} - \bar{q}_{t+1}\|^2] + \frac{1}{N}\sum_{n=1}^{N}\sum_{k=1}^{K}\mathbb{E}[\|h_{t+1}^{(n,k)} - \bar{h}_{t+1}^{(k)}\|^2]$$

$$\leq 36\tau^3\omega^2\gamma_z^2\gamma_x^2\eta^4 \sum_{t'=s_t\tau}^{t-1}(1+\frac{27}{25\tau})^{t-1-t'}\frac{1}{N}\sum_{n=1}^{N}\mathbb{E}[\|p_{t'}^{(n)} - \bar{p}_{t'}\|^2] + E_p\sum_{t'=s_t\tau}^{t-1}(1+\frac{27}{25\tau})^{t-1-t'}\mathbb{E}[\|\bar{p}_{t'}\|^2]$$

$$+ E_q\sum_{t'=s_t\tau}^{t-1}(1+\frac{27}{25\tau})^{t-1-t'}\mathbb{E}[\|\bar{q}_{t'}\|^2] + 36\tau\omega^2\sum_{t'=s_t\tau}^{t-1}(1+\frac{27}{25\tau})^{t-1-t'}\mathbb{E}[\|\bar{z}_{t'+1} - \bar{z}_{t'}\|^2]$$

$$+ E_x\sum_{t'=s_t\tau}^{t-1}(1+\frac{27}{25\tau})^{t-1-t'}\frac{1}{N}\sum_{n=1}^{N}\mathbb{E}[\|x_{t'}^{(n)} - \bar{x}_{t'}\|^2] + E_y\sum_{t'=s_t\tau}^{t-1}(1+\frac{27}{25\tau})^{t-1-t'}\frac{1}{N}\sum_{n=1}^{N}\mathbb{E}[\|y_{t'}^{(n)} - \bar{y}_{t'}\|^2]$$

$$+ E_g\sum_{t'=s_t\tau}^{t-1}(1+\frac{27}{25\tau})^{t-1-t'}\frac{1}{N}\sum_{n=1}^{N}\mathbb{E}[\|g_{h,t'}^{(n,k)} - h_{t'}^{(n,k)}\|^2] + E\sum_{t'=s_t\tau}^{t-1}(1+\frac{27}{25\tau})^{t-1-t'} \ ,$$

where we use the following notation for the coefficients of according terms:

$$E_x = 12\tau^2\omega^2\gamma_z\eta(\beta_x^2\eta^4 + 3\gamma_z^2\eta^2) + 12\tau\beta_x^2\eta^4(\omega^2 + 4C_K^2) + 16\tau\alpha^2\eta^4 C_g^2 \ ,$$
$$E_y = 48\tau\beta_x^2\eta^4 C_K^2 + 16\tau\beta_y^2\eta^4 L_f^2 \ ,$$

where $s_t = \lfloor 27(t+1)/25\tau \rfloor$. Then it follows directly that $(1+\frac{27}{25\tau})^{t-1-t'} \leq (1+\frac{27}{25\tau})^\tau < 3$.

Then, by summing up $t'$ from $s_t\tau$ to $(s_t+1)\tau - 1$, we can get

$$\sum_{t'=s_t\tau}^{(s_t+1)\tau-1}\frac{1}{N}\sum_{n=1}^{N}\left(\mathbb{E}[\|p_{t+1}^{(n)} - \bar{p}_{t+1}\|^2] + \mathbb{E}[\|q_{t+1}^{(n)} - \bar{q}_{t+1}\|^2] + \sum_{k=1}^{K}\mathbb{E}[\|h_{t+1}^{(n,k)} - \bar{h}_{t+1}^{(k)}\|^2]\right)$$

$$\leq 3\tau E_p\sum_{t'=s_t\tau}^{(s_t+1)\tau-1}\mathbb{E}[\|\bar{p}_{t'}\|^2] + 3\tau E_q\sum_{t'=s_t\tau}^{(s_t+1)\tau-1}\mathbb{E}[\|\bar{q}_{t'}\|^2] + 108\tau^2\omega^2\sum_{t'=s_t\tau}^{(s_t+1)\tau-1}\mathbb{E}[\|\bar{z}_{t'+1} - \bar{z}_{t'}\|^2]$$

$$+ \left(3\tau^3\gamma_x^2\eta^2 E_x + 108\tau^4\omega^2\gamma_z^2\gamma_x^2\eta^4\right)\sum_{t'=s_t\tau}^{(s_t+1)\tau-1}\frac{1}{N}\sum_{n=1}^{N}\mathbb{E}[\|p_{t'}^{(n)} - \bar{p}_{t'}\|^2] + 3\tau^3\gamma_y^2\eta^2 E_y\sum_{t'=s_t\tau}^{(s_t+1)\tau-1}\frac{1}{N}\sum_{n=1}^{N}\mathbb{E}[\|q_{t'}^{(n)} - \bar{q}_{t'}\|^2]$$

$$+ 3\tau E_g\sum_{t'=s_t\tau}^{(s_t+1)\tau-1}\frac{1}{N}\sum_{n=1}^{N}\mathbb{E}[\|g_{h,t'}^{(n,k)} - h_{t'}^{(n,k)}\|^2] + 3\tau E \ .$$

At last, by enforcing

$$3\tau^3\gamma_x^2\eta^2 E_x + 108\tau^4\omega^2\gamma_z^2\gamma_x^2\eta^4 \leq \frac{1}{2} , \quad 3\tau^3\gamma_y^2\eta^2 E_y \leq \frac{1}{2} \ ,$$

the proof is complete, and Eq. (27) holds.

To make sure the above inequalities holds, we decompose the first one as follows:

$$
3\tau^3\gamma_x^2\eta^2 E_x + 108\tau^4\omega^2\gamma_z^2\gamma_x^2\eta^4
$$
$$
= 3\tau^4\gamma_x^2\eta^2\Big(\underbrace{36\tau\omega^2\gamma_z^3\eta^3}_{E_{x,1}} + \underbrace{12\tau\omega^2\gamma_z\eta\beta_x^2\eta^4}_{E_{x,2}} + \underbrace{12\beta_x^2\eta^4(\omega^2 + 4C_K^2) + 16\alpha^2\eta^4 C_g^2}_{E_{x,3}} + \underbrace{36\omega^2\gamma_z^2\eta^2}_{E_{x,4}}\Big).
$$

By imposing $\gamma_z \le 10\omega$, with $\gamma_x \le 10$, $\eta \le \frac{1}{400\tau C_{K,\omega}}$, we have

$$
3\tau^4\gamma_x^2\eta^2 E_{x,1} = 3\tau^4\gamma_x^2\eta^2 \times 36\tau\omega^2\gamma_z^3\eta^3 \le \frac{108 \times 100\tau^5\omega^2 \times 1000\omega^3}{400^5\tau^5 C_{K,\omega}^5} \le \frac{1}{12}.
$$

Similarly, by setting $\beta_x = \frac{C_{K,\omega}^2}{N}, \beta_y = \frac{C_{K,\omega}^2}{N}, \alpha = \frac{C_{K,\omega}^2}{N}$, we further deduce

$$
3\tau^4\gamma_x^2\eta^2 E_{x,2} = 36\tau^5\omega^2\gamma_x^2\eta^2\gamma_z\eta\beta_x^2\eta^4 \le \frac{36 \times 100\tau^5\omega^2 \times 10\omega C_{K,\omega}^4}{400^7\tau^7 C_{K,\omega}^7 N^2} \le \frac{1}{12},
$$

$$
3\tau^4\gamma_x^2\eta^2 E_{x,3} = 3\tau^4\gamma_x^2\eta^2(12\beta_x^2\eta^4(\omega^2 + 4C_K^2) + 16\alpha^2\eta^4 C_g^2) \le \frac{36 \times 100\tau^4 C_{K,\omega}^4(\omega^2 + 4C_K^2 + C_g^2)}{400^6\tau^6 C_{K,\omega}^6 N^2} \le \frac{1}{12},
$$

$$
3\tau^4\gamma_x^2\eta^2 E_{x,4} = 3\tau^4\gamma_x^2\eta^2 \times 36\omega^2\gamma_z^2\eta^2 \le \frac{108 \times 100\tau^4\omega^2 \times 100\omega^2}{400^4\tau^4 C_{K,\omega}^4} \le \frac{1}{12}.
$$

We can make sure $3\tau^3\gamma_y^2\eta^2 E_y \le \frac{1}{2}$ in a similar manner. Hence, the proof is complete. $\qquad\square$

### C.3. Proof of Theorem

*Proof.* To establish the convergence rate of Algorithm 1, we introduce the following novel potential function:

$$
\mathcal{P}_t = \Psi_t + c_a\mathbb{E}[\|\bar{p}_t - \frac{1}{N}\sum_{n=1}^N \nabla_x F_{h,t}^{(n)}\|^2] + c_b\mathbb{E}[\|\bar{q}_t - \frac{1}{N}\sum_{n=1}^N \nabla_y F_{h,t}^{(n)}\|^2]
$$
$$
+ \sum_{k=1}^K \theta_k\mathbb{E}[\|\frac{1}{N}\sum_{n=1}^N g_{h,t}^{(n,k)} - \frac{1}{N}\sum_{n=1}^N h_t^{(n,k)}\|^2] + \sum_{k=1}^K \nu_k\frac{1}{N}\sum_{n=1}^N \mathbb{E}[\|g_{h,t}^{(n,k)} - h_t^{(n,k)}\|^2], \tag{30}
$$

where $\Psi_t := F(G(\bar{x}_t), \bar{y}_t; \bar{z}_t) - 2\psi_d(\bar{y}_t; \bar{z}_t) + 2\psi(\bar{z}_t)$, the coefficient $c_a$, $c_b$ and $\{\theta_k\}_{k=1}^K$, $\{\nu_k\}_{k=1}^K$ are positive.

By Lemmas C.5, C.7, C.8, C.9 and C.10 to potential function Eq.(30), we have

$$
\mathcal{P}_{t+1} - \mathcal{P}_t \le -\frac{\gamma_x\eta}{4}\mathbb{E}[\|\nabla_x F(G(\bar{x}_t), \bar{y}_t; \bar{z}_t)\|^2] - \frac{\gamma_y\eta}{4}\mathbb{E}[\|\nabla_y F(G(\bar{x}_t), \bar{y}_t; \bar{z}_t)\|^2]
$$
$$
+ \Big(2C_{x_{yz}^2} + 24\gamma_z\eta(C_{x_z}^2 + C_{x_{yz}^2}^2) - \frac{1}{3\gamma_z\eta}\Big)\omega\mathbb{E}[\|\bar{z}_{t+1} - \bar{z}_t\|^2] + \frac{\gamma_x\eta}{2}\mathbb{E}[\|\nabla_x F(G(\bar{x}_t), \bar{y}_t; \bar{z}_t) - \bar{p}_t\|^2]
$$
$$
+ \frac{5\gamma_y\eta}{4}\mathbb{E}[\|\nabla_y F(G(\bar{x}_t), \bar{y}_t; \bar{z}_t) - \bar{q}_t\|^2] - \beta_x\eta^2 c_a\mathbb{E}[\|\bar{p}_t - \frac{1}{N}\sum_{n=1}^N \nabla_x F_{h,t}^{(n)}\|^2] - \beta_y\eta^2 c_b\mathbb{E}[\|\bar{q}_t - \frac{1}{N}\sum_{n=1}^N \nabla_y F_{h,t}^{(n)}\|^2]
$$
$$
+ M_p\mathbb{E}[\|\bar{p}_t\|^2] + M_q\mathbb{E}[\|\bar{q}_t\|^2] + M_{g,l}\frac{1}{N}\sum_{n=1}^N \mathbb{E}[\|g_{h,t}^{(n,k)} - h_t^{(n,k)}\|^2] + M_{g,g}\mathbb{E}[\|\frac{1}{N}\sum_{n=1}^N g_{h,t}^{(n,k)} - \frac{1}{N}\sum_{n=1}^N h_t^{(n,k)}\|^2]
$$
$$
+ M_{p,c}\frac{1}{N}\sum_{n=1}^N \mathbb{E}[\|\bar{p}_t - p_t^{(n)}\|^2] + M_{q,c}\frac{1}{N}\sum_{n=1}^N \mathbb{E}[\|\bar{q}_t - q_t^{(n)}\|^2] + M_0,
$$

where, for readability, we use the following notation for the coefficients of each term and the constant term:

$$
M_{p,c} = \gamma_x^2\eta^2\Big(4C_K^2\sum_{k=0}^K (2C_g^2)^k\frac{c_a}{N} + 4L_f^2(2C_g^2)^K\frac{c_b}{N} + 2\sum_{k=1}^K (2C_g^2)^k(\frac{\theta_k}{N} + \nu_k)\Big),
$$

$$M_{q,c} = \gamma_y^2\eta^2\Big(4C_K^2\frac{c_a}{N} + 4L_f^2\frac{c_b}{N}\Big),$$

$$M_p = M_{p,c} + \gamma_x^2\eta^2\Big(\gamma_y\eta\ell^2 - \frac{1}{4\gamma_x\eta}\Big),$$

$$M_q = M_{q,c} + \gamma_y^2\eta^2\Big(\frac{\omega+\ell}{2} + L_d - \frac{1}{8\gamma_y\eta}\Big),$$

$$M_{g,l} = \alpha^2\eta^4\sum_{k=1}^{K}\Big(4C_K^2\frac{c_a}{N}\sum_{j=k}^{K}(2C_g^2)^{j-k} + 4L_f^2\frac{c_b}{N}(2C_g^2)^{K-k} + 2\sum_{j=k}^{K}\nu_j(2C_g^2)^{j-k} - \frac{\nu_k}{\alpha\eta^2}\Big),$$

$$M_{g,g} = 2\alpha^2\eta^4\sum_{k=1}^{K}\Big(\sum_{j=k+1}^{K}\theta_j(2C_g^2)^{j-k}\Big) - \alpha\eta^2\sum_{k=1}^{K}\theta_k,$$

$$M_0 = \alpha^2\eta^4\sigma^2\sum_{k=1}^{K}\Big(2(\frac{2c_aC_K^2}{N} + \frac{\theta_k}{N} + \nu_k)\sum_{j=1}^{k}(2C_g^2)^{j-1} + 4L_f^2(2C_g^2)^{k-1}\frac{c_b}{N}\Big)$$

$$+ 2\beta_x^2\eta^4\sigma^2\frac{c_aC_K^2}{N} + 2\beta_y^2\eta^4\sigma^2\frac{c_b}{N}. \tag{31}$$

### C.3.1. STEP I: ELIMINATE THE GLOBAL GRADIENT ESTIMATION ERROR TERM.

There are three global gradient estimation error terms we need to handle:

(i)  Global gradient estimation error regarding $x$: $\mathbb{E}[\|\bar{p}_t - \frac{1}{N}\sum_{n=1}^{N}\nabla_x F_{h,t}^{(n)}\|^2]$;

(ii)  Global gradient estimation error regarding $y$: $\mathbb{E}[\|\bar{q}_t - \frac{1}{N}\sum_{n=1}^{N}\nabla_y F_{h,t}^{(n)}\|^2]$;

(iii)  Global estimation error regarding $h$: $\mathbb{E}[\|\frac{1}{N}\sum_{n=1}^{N} g_{h,t}^{(n,k)} - \frac{1}{N}\sum_{n=1}^{N} h_t^{(n,k)}\|^2]$.

(i) To eliminate the term $\mathbb{E}[\|\bar{p}_t - \frac{1}{N}\sum_{n=1}^{N}\nabla_x F_{h,t}^{(n)}\|^2]$, we begin by expanding $\mathbb{E}[\|\nabla_x F(G(\bar{x}_t), \bar{y}_t; \bar{z}_t) - \bar{p}_t\|^2]$ as follows:

$$\frac{\gamma_x\eta}{2}\mathbb{E}[\|\nabla_x F(G(\bar{x}_t), \bar{y}_t; \bar{z}_t) - \bar{p}_t\|^2]$$

$$\leq 2\gamma_x\eta\mathbb{E}[\|\nabla_x F(G(\bar{x}_t), \bar{y}_t; \bar{z}_t) - \frac{1}{N}\sum_{n=1}^{N}\nabla_x F^{(n)}(\frac{1}{N}\sum_{n'=1}^{N} G^{(n')}(x_t^{(n')}), y_t^{(n)}; z_t^{(n)})\|^2]$$

$$+ 2\gamma_x\eta\mathbb{E}[\|\frac{1}{N}\sum_{n=1}^{N}\nabla_x F^{(n)}(\frac{1}{N}\sum_{n'=1}^{N} G^{(n')}(x_t^{(n')}), y_t^{(n)}; z_t^{(n)}) - \frac{1}{N}\sum_{n=1}^{N}\nabla_x F^{(n)}(\frac{1}{N}\sum_{n'=1}^{N} H^{(n')}(x_t^{(n')}), y_t^{(n)}; z_t^{(n)})\|^2]$$

$$+ 2\gamma_x\eta\mathbb{E}[\|\frac{1}{N}\sum_{n=1}^{N}\nabla_x F^{(n)}(\frac{1}{N}\sum_{n'=1}^{N} H^{(n')}(x_t^{(n')}), y_t^{(n)}; z_t^{(n)}) - \frac{1}{N}\sum_{n=1}^{N}\nabla_x F_{h,t}^{(n)}\|^2] + 2\gamma_x\eta\mathbb{E}[\|\frac{1}{N}\sum_{n=1}^{N}\nabla_x F_{h,t}^{(n)} - \bar{p}_t\|^2]$$

$$\triangleq 2\gamma_x\eta\Big(T_1 + T_2 + T_3 + \mathbb{E}[\|\frac{1}{N}\sum_{n=1}^{N}\nabla_x F_{h,t}^{(n)} - \bar{p}_t\|^2]\Big). \tag{32}$$

Considering $T_1$ in Eq. (32), we derive

$$T_1 = \mathbb{E}\Big[\Big\|\nabla_x f(G(\bar{x}_t), \bar{y}_t) + \omega(\bar{x}_t - \bar{z}_t) - \frac{1}{N}\sum_{n=1}^{N}\Big(\nabla_x f^{(n)}(\frac{1}{N}\sum_{n'=1}^{N} G^{(n')}(x_t^{(n')}), y_t^{(n)}) + \omega(x_t^{(n)} - z_t^{(n)})\Big)\Big\|^2\Big]$$

$$= \mathbb{E}\Big[\Big\|\nabla_x f(G(\bar{x}_t), \bar{y}_t) - \frac{1}{N}\sum_{n=1}^{N}\nabla_x f^{(n)}(\frac{1}{N}\sum_{n'=1}^{N} G^{(n')}(x_t^{(n')}), y_t^{(n)})\Big\|^2\Big]$$

$$\leq (K+1)\Big(C_G^4 L_f^2 + \sum_{k=0}^{K-1} C_g^{2(K-1+k)}C_f^2 L_g^2\Big)\frac{1}{N}\sum_{n=1}^{N}\mathbb{E}[\|\bar{x}_t - x_t^{(n)}\|^2] + (K+1)C_G^2 L_f^2\frac{1}{N}\sum_{n=1}^{N}\mathbb{E}[\|\bar{y}_t - y_t^{(n)}\|^2]$$

$$\leq C_K^2 \frac{1}{N} \sum_{n=1}^{N} \mathbb{E}[\|\bar{x}_t - x_t^{(n)}\|^2] + C_K^2 \frac{1}{N} \sum_{n=1}^{N} \mathbb{E}[\|\bar{y}_t - y_t^{(n)}\|^2] \,,$$

where the second-to-last step follows from Eq. (37) in (Zhang et al., 2024), and the last step follows from the definition of $C_K^2$.

Similarly, for $T_2$ and $T_3$ in Eq. (32), we obtain

$$T_2 \leq K \sum_{k=1}^{K} A_k \mathbb{E}[\|\frac{1}{N} \sum_{n=1}^{N} g_{h,t}^{(n,k)} - \frac{1}{N} \sum_{n=1}^{N} h_t^{(n,k)}\|^2] \,,$$

$$T_3 \leq C_K^2 \frac{1}{N} \sum_{n=1}^{N} \sum_{k=1}^{K} \mathbb{E}[\|\bar{h}_t^{(k)} - h_t^{(n,k)}\|^2] \,,$$

where $T_2$ follows directly from the second property in Lemma C.6, and $T_3$ is derived in the same manner as $T_1$.

Hence, Eq. (32) can be bounded as follows:

$$\frac{\gamma_x \eta}{2} \mathbb{E}[\|\nabla_x F(G(\bar{x}_t), \bar{y}_t; \bar{z}_t) - \bar{p}_t\|^2]$$

$$\leq 2\gamma_x \eta C_K^2 \frac{1}{N} \sum_{n=1}^{N} \mathbb{E}[\|\bar{x}_t - x_t^{(n)}\|^2] + 2\gamma_x \eta C_K^2 \frac{1}{N} \sum_{n=1}^{N} \mathbb{E}[\|\bar{y}_t - y_t^{(n)}\|^2] + 2\gamma_x \eta C_K^2 \frac{1}{N} \sum_{n=1}^{N} \sum_{k=1}^{K} \mathbb{E}[\|\bar{h}_t^{(k)} - h_t^{(n,k)}\|^2]$$

$$+ 2\gamma_x \eta K \sum_{k=1}^{K} A_k \mathbb{E}[\|\frac{1}{N} \sum_{n=1}^{N} g_{h,t}^{(n,k)} - \frac{1}{N} \sum_{n=1}^{N} h_t^{(n,k)}\|^2] + 2\gamma_x \eta \mathbb{E}[\|\frac{1}{N} \sum_{n=1}^{N} \nabla_x F_{h,t}^{(n)} - \bar{p}_t\|^2] \,. \tag{33}$$

By setting $c_a = \frac{3\gamma_x}{\beta_x \eta}$, the coefficient of $\mathbb{E}[\|\bar{p}_t - \frac{1}{N} \sum_{n=1}^{N} \nabla_x F_{h,t}^{(n)}\|^2]$ can be negative: $2\gamma_x \eta - \beta_x \eta^2 c_a \leq 0$.

(ii) Similarly, to eliminate the term $\mathbb{E}[\|\bar{q}_t - \frac{1}{N} \sum_{n=1}^{N} \nabla_y F_{h,t}^{(n)}\|^2]$, $\mathbb{E}[\|\nabla_y F(G(\bar{x}_t), \bar{y}_t; \bar{z}_t) - \bar{q}_t\|^2]$ can be bounded as follows:

$$\frac{5\gamma_y \eta}{4} \mathbb{E}[\|\nabla_y F(G(\bar{x}_t), \bar{y}_t; \bar{z}_t) - \bar{q}_t\|^2]$$

$$\leq 5\gamma_y \eta L_f^2 C_G^2 \frac{1}{N} \sum_{n=1}^{N} \mathbb{E}[\|\bar{x}_t - x_t^{(n)}\|^2] + 5\gamma_y \eta L_f^2 \frac{1}{N} \sum_{n=1}^{N} \mathbb{E}[\|\bar{y}_t - y_t^{(n)}\|^2] + 5\gamma_y \eta L_f^2 \frac{1}{N} \sum_{n=1}^{N} \mathbb{E}[\|\bar{h}_t^{(K)} - h_t^{(n,K)}\|^2]$$

$$+ 5\gamma_y \eta L_f^2 K \sum_{k=1}^{K} C_g^{2(K-k)} \mathbb{E}[\|\frac{1}{N} \sum_{n=1}^{N} g_{h,t}^{(n,k)} - \frac{1}{N} \sum_{n=1}^{N} h_t^{(n,k)}\|^2] + 5\gamma_y \eta \mathbb{E}[\|\frac{1}{N} \sum_{n=1}^{N} \nabla_y F_{h,t}^{(n)} - \bar{q}_t\|^2] \,. \tag{34}$$

Therefore, by setting $c_b = \frac{6\gamma_y}{\beta_y \eta}$, the coefficient of $\mathbb{E}[\|\bar{q}_t - \frac{1}{N} \sum_{n=1}^{N} \nabla_y F_{h,t}^{(n)}\|^2]$ can be negative: $5\gamma_y \eta - \beta_y \eta^2 c_b \leq 0$.

(iii) Then, we proceed to eliminate the term $\mathbb{E}[\|\frac{1}{N} \sum_{n=1}^{N} g_{h,t}^{(n,k)} - \frac{1}{N} \sum_{n=1}^{N} h_t^{(n,k)}\|^2]$. After incorporating the newly introduced coefficient term from the previous steps with $M_{g,g}$, the coefficient of $\mathbb{E}[\|\frac{1}{N} \sum_{n=1}^{N} g_{h,t}^{(n,k)} - \frac{1}{N} \sum_{n=1}^{N} h_t^{(n,k)}\|^2]$ becomes:

$$2\gamma_x \eta K \sum_{k=1}^{K} A_k + 5\gamma_y \eta L_f^2 K \sum_{k=1}^{K} C_g^{2(K-k)} + 2\alpha^2 \eta^4 \sum_{k=1}^{K} \Big( \sum_{j=k+1}^{K} \theta_j (2C_g^2)^{j-k} \Big) - \alpha \eta^2 \sum_{k=1}^{K} \theta_k \leq 0 \,. \tag{35}$$

For any $k \in \{1, \cdots, K\}$, it follows that

$$2\gamma_x \eta K A_k + 5\gamma_y \eta L_f^2 K C_g^{2(K-k)} - \frac{1}{2} \alpha \eta^2 \theta_k = \alpha \eta^2 \Big[ \frac{1}{\alpha \eta} 2\gamma_x K A_k + \frac{1}{\alpha \eta} 5\gamma_x \psi_{\gamma_y} L_f^2 K C_g^{2(K-k)} - \frac{1}{2} \theta_k \Big] \,.$$

Since $\alpha \eta^2 \leq 1$, we require this upper bound to be non-positive, which yields

$$\theta_k = \frac{K \gamma_x}{\alpha \eta} \Big[ 4 A_k + 10 \psi_{\gamma_y} L_f^2 C_g^{2(K-k)} \Big] \triangleq \frac{K \gamma_x}{\alpha \eta} \hat{\theta}_k \,.$$

In a similar manner, for any $k \in 1, \ldots, K$, we impose

$$2\alpha^2\eta^4\Big( \sum_{j=k+1}^{K} \theta_j (2C_g^2)^{j-k} \Big) - \alpha\eta^2\theta_k \leq -\frac{1}{2}\alpha\eta^2\theta_k \,,$$

and consequently obtain

$$\eta \leq \frac{1}{2}\hat{\theta}_k^{1/2} \cdot \Big( \sum_{j=k}^{K} \alpha\hat{\theta}_j (2C_g^2)^{j-k} \Big)^{-1/2} \,.$$

### C.3.2. STEP II. SUMMATION OVER $T$ AND ELIMINATION OF REMAINING TERMS.

After summing over $t = \{0, \cdots, T-1\}$ and applying the overall consensus error bound (Lemma C.14), we identify four additional terms that need to be addressed:

  (i) Local estimation error regarding $h$: $\frac{1}{N}\sum_{n=1}^{N} \mathbb{E}[\|g_{h,t}^{(n,k)} - h_t^{(n,k)}\|^2]$;

  (ii) Auxiliary term involving $z$: $\mathbb{E}[\|\bar{z}_{t+1} - \bar{z}_t\|^2]$;

  (iii) Primal term involving $x$: $\mathbb{E}[\|\bar{p}_t\|^2]$;

  (iv) Dual term involving $y$: $\mathbb{E}[\|\bar{q}_t\|^2]$.

By taking the summation over $t = \{0, \cdots, T-1\}$, applying Lemma C.14 and define

$$\mathcal{M} := M_{p,c} + M_{q,c} + \Big( \tau^2(\gamma_x^2 + \gamma_y^2)\eta^2 + 1 \Big)(2\gamma_x\eta + 5\gamma_y\eta)C_K^2 \,, \tag{36}$$

we obtain

$$\frac{\mathcal{P}_T - \mathcal{P}_0}{T} \leq -\frac{\gamma_x\eta}{4}\frac{1}{T}\sum_{t=0}^{T-1} \mathbb{E}[\|\nabla_x F(G(\bar{x}_t), \bar{y}_t; \bar{z}_t)\|^2] - \frac{\gamma_y\eta}{4}\frac{1}{T}\sum_{t=0}^{T-1} \mathbb{E}[\|\nabla_y F(G(\bar{x}_t), \bar{y}_t; \bar{z}_t)\|^2]$$

$$+ \Big( 2C_{x_{yz}^2} + 24\gamma_z\eta(C_{x_z}^2 + C_{x_{yz}^2}) + 216\tau^2\omega\mathcal{M} - \frac{1}{3\gamma_z\eta} \Big)\omega\frac{1}{T}\sum_{t=0}^{T-1} \mathbb{E}[\|\bar{z}_{t+1} - \bar{z}_t\|^2]$$

$$+ \Big( M_p + 6\tau E_p\mathcal{M} \Big)\frac{1}{T}\sum_{t=0}^{T-1} \mathbb{E}[\|\bar{p}_t\|^2] + \Big( M_q + 6\tau E_q\mathcal{M} \Big)\frac{1}{T}\sum_{t=0}^{T-1} \mathbb{E}[\|\bar{q}_t\|^2]$$

$$+ \Big( M_{g,l} + 6\tau E_g\mathcal{M} \Big)\frac{1}{T}\sum_{t=0}^{T-1}\frac{1}{N}\sum_{n=1}^{N} \mathbb{E}[\|g_{h,t}^{(n,k)} - h_t^{(n,k)}\|^2] + M_0 + 6\tau E\mathcal{M} \,.$$

(i). To eliminate the term $\frac{1}{N}\sum_{n=1}^{N} \mathbb{E}[\|g_{h,t}^{(n,k)} - h_t^{(n,k)}\|^2]$, from the definition of $M_{g,l}$, with $E_g \leq \sum_{k=1}^{K} 4\alpha^2\eta^4\tau(8C_K^2 + 1)\sum_{j=k}^{K}(2C_g^2)^{j-k}$, we enforce

$$4\alpha^2\eta^4 C_K^2\frac{c_a}{N}\sum_{j=k}^{K}(2C_g^2)^{j-k} + 4\alpha^2\eta^4 L_f^2\frac{c_b}{N}(2C_g^2)^{K-k}$$

$$+ 24\alpha^2\eta^4\tau^2(8C_K^2 + 1)\sum_{j=k}^{K}(2C_g^2)^{j-k}\mathcal{M} + 2\sum_{j=k}^{K}\nu_j(2C_g^2)^{j-k} - \alpha\eta^2\nu_k \leq 0 \,. \tag{37}$$

From the definition of $\mathcal{M}$ in Eq. (36), we further require

$$4\alpha^2\eta^4 C_K^2\frac{c_a}{N}\sum_{j=k}^{K}(2C_g^2)^{j-k} + 4\alpha^2\eta^4 L_f^2\frac{c_b}{N}(2C_g^2)^{K-k}$$

$$+ 24\alpha^2\eta^4\tau^2(8C_K^2+1)\sum_{j=k}^{K}(2C_g^2)^{j-k}\left(\gamma_x^2\eta^2\left(4C_K^2\sum_{k'=0}^{K}(2C_g^2)^{k'}\frac{c_a}{N} + 4C_K^2(2C_g^2)^K\frac{c_b}{N} + 2\sum_{k'=1}^{K}(2C_g^2)^{k'}\frac{\theta_{k'}}{N}\right)\right.$$

$$\left.+ \gamma_y^2\eta^2\left(4C_K^2\frac{c_a}{N} + 4L_f^2\frac{c_b}{N}\right) + \left(\tau^2(\gamma_x^2+\gamma_y^2)\eta^2+1\right)(2\gamma_x\eta+5\gamma_y\eta)C_K^2\right) - \frac{1}{2}\alpha\eta^2\nu_k \le 0 \,.$$

Moreover, under the conditions Eq. (26), we obtain

$$\nu_k = \frac{\gamma_x}{\eta N}\left(\frac{32}{\beta_x} + \frac{64\psi_{\gamma_y}}{\beta_y} + \frac{\sum_{k'=1}^{K}K\hat{\theta}_{k'} + 5(1+\psi_{\gamma_y})N}{\alpha}\right)C_K^2\sum_{j=k}^{K}(2C_g^2)^{j-k} \triangleq \frac{\gamma_x}{\eta N}\hat{\nu}_k \,. \tag{38}$$

Finally, by solving

$$24\alpha^2\eta^4\tau^2(8C_K^2+1)\sum_{j=k}^{K}(2C_g^2)^{j-k}2\gamma_x^2\eta^2\sum_{k'=1}^{K}(2C_g^2)^{k'}\nu_{k'} + 2\alpha^2\eta^4\sum_{j=k}^{K}\nu_j(2C_g^2)^{j-k} - \frac{1}{2}\alpha\eta^2\nu_k \le 0 \,,$$

we obtain

$$\eta \le \hat{\nu}_k^{1/2}\cdot\left(\sum_{j=k}^{K}\alpha(\sum_{k'=1}^{K}\hat{\nu}_{k'}+4\hat{\nu}_j)(2C_g^2)^{j-k}\right)^{-1/2} \,.$$

(ii). For the term $\mathbb{E}[\|\bar{z}_{t+1}-\bar{z}_t\|^2]$, we set

$$\left(2C_{x_{yz}^2} + 24\gamma_z\eta(C_{x_z}^2 + C_{x_{yz}^2}^2) + 216\tau^2\omega\mathcal{M} - \frac{1}{3\gamma_z\eta}\right)\omega \le -\frac{\omega}{4\gamma_z\eta} \,.$$

In particular, we impose

$$2C_{x_{yz}^2} \le \frac{1}{36\gamma_z\eta} \,, \quad 24\gamma_z\eta(C_{x_z}^2 + C_{x_{yz}^2}^2) \le \frac{1}{36\gamma_z\eta} \,, \quad 216\tau^2\omega\mathcal{M} \le \frac{1}{36\gamma_z\eta} \,.$$

For the first two inequalities, from $C_{x_z} = C_{x_{yz}^2}$ in Lemma C.1, we obtain: $\gamma_x \le \frac{1}{72\psi_{\gamma_z}C_{x_z}}$.

To handle the last inequality, under the conditions Eq. (26), we deduce $(\gamma_x^2+\gamma_y^2)\eta^2 \le \frac{200}{400^2\tau^2C_{K,\omega}^2}$. Moreover, from the definition of $\mathcal{M}$ in Eq. (36), we require

$$\frac{216\tau^2\omega\times200}{400^2\tau^2C_{K,\omega}^2}\sum_{k=1}^{K}(2C_g^2)^k\nu_k \le \frac{1}{216\gamma_z\eta} \,, \tag{39}$$

and from the definition of $\nu_k$ in Eq. (38), we further enforce:

$$\frac{216\tau^2\omega\times200}{400^2\tau^2C_{K,\omega}^2}\sum_{k=1}^{K}(2C_g^2)^k\frac{\gamma_x}{\eta N}\frac{32}{\beta_x}C_K^2\sum_{j=k}^{K}(2C_g^2)^{j-k} \le \frac{1}{216\times4\gamma_z\eta} \,,$$

$$\frac{216\tau^2\omega\times200}{400^2\tau^2C_{K,\omega}^2}\sum_{k=1}^{K}(2C_g^2)^k\frac{\gamma_x}{\eta N}\frac{64\psi_{\gamma_y}}{\beta_y}C_K^2\sum_{j=k}^{K}(2C_g^2)^{j-k} \le \frac{1}{216\times4\gamma_z\eta} \,,$$

$$\frac{216\tau^2\omega\times200}{400^2\tau^2C_{K,\omega}^2}(\sum_{k'=1}^{K}K\hat{\theta}_{k'} + 5(1+\psi_{\gamma_y})N)\sum_{k=1}^{K}(2C_g^2)^k\frac{\gamma_x}{\alpha\eta N}C_K^2\sum_{j=k}^{K}(2C_g^2)^{j-k} \le \frac{1}{216\times2\gamma_z\eta} \,.$$

Then, by solving the above inequalities and defining $V_{2C_g} = \sum_{k=1}^{K}(2C_g^2)^k\sum_{j=k}^{K}(2C_g^2)^{j-k}$, we obtain

$$\gamma_x \le \min\left\{\frac{\sqrt{\beta_x N}}{128\sqrt{\psi_{\gamma_z}\omega V_{2C_g}}}, \frac{\sqrt{\beta_y N}}{128\sqrt{2\psi_{\gamma_y}\psi_{\gamma_z}\omega V_{2C_g}}}, \frac{\sqrt{\alpha N}}{16\sqrt{\psi_{\gamma_z}\omega V_{2C_g}}}\cdot(\sum_{k'=1}^{K}K\hat{\theta}_{k'} + 5(1+\psi_{\gamma_y})N)^{-1/2}\right\} \,.$$

Finally, the remaining terms in $\mathcal{M}$ can be controlled in the same manner as in Eq. (39), which guarantees that the above bound is satisfied and further yields the following condition:

$$\gamma_x \leq \min\left\{\frac{5(1+\psi_{\gamma_y})^{-1/2}}{27\tau\eta\sqrt{10\psi_{\gamma_z}\omega}}, \frac{(1+\psi_{\gamma_y})^{-1/2}}{216\tau\eta C_K\sqrt{5\psi_{\gamma_z}\omega}}\right\}.$$

(iii) Similarly, for the term $\mathbb{E}[\|\bar{p}_t\|^2]$, from the definition of $M_p$, with $E_p \leq 4\tau\gamma_x^2\eta^2\left((8C_K^2+1)\sum_{k=0}^{K}(2C_g^2)^k + 6\omega^2\right)$, we enforce

$$\gamma_x^2\eta^2\left(\gamma_y\eta\ell^2 + 4C_K^2\sum_{k=0}^{K}(2C_g^2)^k\frac{c_a}{N} + 4L_f^2(2C_g^2)^K\frac{c_b}{N} + 2\sum_{k=1}^{K}(2C_g^2)^k(\frac{\theta_k}{N}+\nu_k)\right.$$
$$\left.+ 24\tau^2\left((8C_K^2+1)\sum_{k=0}^{K}(2C_g^2)^k + 6\omega^2\right)\mathcal{M} - \frac{1}{4\gamma_x\eta}\right) \leq 0. \tag{40}$$

Specifically, we further impose

$$\gamma_x^2\eta^2 4C_K^2\sum_{k=0}^{K}(2C_g^2)^k\frac{c_a}{N} \leq \frac{\gamma_x\eta}{32}, \quad \gamma_x^2\eta^2 4L_f^2(2C_g^2)^K\frac{c_b}{N} \leq \frac{\gamma_x\eta}{32},$$

$$\gamma_x^2\eta^2 2\sum_{k=1}^{K}(2C_g^2)^k\frac{\theta_k}{N} \leq \frac{\gamma_x\eta}{32}, \quad \gamma_x^2\eta^2 2\sum_{k=1}^{K}(2C_g^2)^k\nu_k \leq \frac{\gamma_x\eta}{32},$$

$$\gamma_x^2\eta^2\gamma_y\eta\ell^2 \leq \frac{\gamma_x\eta}{32}, \quad \gamma_x^2\eta^2 24\tau^2\left((8C_K^2+1)\sum_{k=0}^{K}(2C_g^2)^k + 6\omega^2\right)\mathcal{M} \leq \frac{\gamma_x\eta}{32}.$$

By solving the above inequalities, we obtain

$$\gamma_x \leq \min\left\{\frac{\sqrt{\beta_x N}}{96C_K\sqrt{V_{2C_g}}}, \frac{\sqrt{\beta_y N}}{96C_K\sqrt{2\psi_{\gamma_y}V_{2C_g}}}, \frac{\sqrt{\alpha N}}{12C_K\sqrt{V_{2C_g}}}\cdot(\sum_{k'=1}^{K}K\hat{\theta}_{k'} + 5(1+\psi_{\gamma_y})N)^{-1/2},\right.$$
$$\left.\frac{1}{4\ell\sqrt{2\psi_{\gamma_y}}}, \frac{5(1+\psi_{\gamma_y})^{-1/2}}{24\sqrt{10}\tau\eta C_K}, \frac{(1+\psi_{\gamma_y})^{-1/2}}{192\sqrt{5}\tau\eta C_K C_{K,\omega}}\right\}.$$

(iv) Subsequently, for the term $\mathbb{E}[\|\bar{q}_t\|^2]$, from the definition of $M_q$, with $E_q \leq 32\tau\gamma_y\eta^2 C_K^2$, we enforce

$$\gamma_y^2\eta^2\left(\frac{\omega+\ell}{2} + L_d + 4C_K^2\frac{c_a}{N} + 4L_f^2\frac{c_b}{N} + 192\tau^2 C_K^2\mathcal{M} - \frac{1}{8\gamma_y\eta}\right) \leq 0. \tag{41}$$

Specifically, we further impose

$$\gamma_y^2\eta^2(\frac{(\omega+\ell)}{2} + L_d) \leq \frac{\gamma_y\eta}{32}, \quad \gamma_y^2\eta^2 4C_K^2\frac{c_a}{N} \leq \frac{\gamma_y\eta}{32},$$
$$\gamma_y^2\eta^2 4L_f^2\frac{c_b}{N} \leq \frac{\gamma_y\eta}{32}, \quad \gamma_y^2\eta^2 192\tau^2 C_K^2\mathcal{M} \leq \frac{\gamma_y\eta}{32}.$$

By solving the above inequalities, we obtain

$$\gamma_x \leq \min\left\{\frac{\sqrt{\beta_x N}}{96C_K\sqrt{\psi_{\gamma_y}V_{2C_g}}}, \frac{\sqrt{\beta_y N}}{96C_K\psi_{\gamma_y}\sqrt{2V_{2C_g}}}, \frac{\sqrt{\alpha N}}{12C_K\sqrt{\psi_{\gamma_y}V_{2C_g}}}\cdot(\sum_{k'=1}^{K}K\hat{\theta}_{k'} + 5(1+\psi_{\gamma_y})N)^{-1/2},\right.$$
$$\left.\frac{(2L_d+\omega+\ell)^{-1}}{16\psi_{\gamma_y}}, \frac{5(1+\psi_{\gamma_y})^{-1/2}}{24\tau\eta C_K\sqrt{10\psi_{\gamma_y}}}, \frac{(1+\psi_{\gamma_y})^{-1/2}}{192\tau\eta C_K^2\sqrt{5\psi_{\gamma_y}}}\right\}.$$

## C.3.3. STEP III. PROOF COMPLETION.

In summary, the value in potential function Eq. (30) are given by

$$c_a = \frac{3\gamma_x}{\beta_x \eta} \,, \quad c_b = \frac{6\gamma_y}{\beta_y \eta} \,, \quad \theta_k = \frac{K\gamma_x}{\alpha\eta}\hat{\theta}_k \,, \quad \nu_k = \frac{\gamma_x}{\eta N}\hat{\nu}_k \,,$$

$$\hat{\theta}_k = 4A_k + 10\psi_{\gamma_y} L_f^2 C_g^{2(K-k)} \,,$$

$$\hat{\nu}_k = \left( \frac{32}{\beta_x} + \frac{64\psi_{\gamma_y}}{\beta_y} + \frac{\sum_{k'=1}^{K} K\hat{\theta}_{k'} + 5(1+\psi_{\gamma_y})N}{\alpha} \right) C_K^2 \sum_{j=k}^{K}(2C_g^2)^{j-k} \,. \tag{42}$$

Then, by denoting

$$V_{2C_g} = \sum_{k=1}^{K}(2C_g^2)^k \sum_{j=k}^{K}(2C_g^2)^{j-k} \,, \quad C_{K,\omega}^2 = (C_K^2+1)\sum_{k=0}^{K}(2C_g^2)^k + \omega^2$$

$$\psi_\omega = (1+C_K^2)(\psi_{\gamma_z}\omega+1)(1+\psi_{\gamma_y}) \,, \tag{43}$$

the hyperparameter setting can be summarized as

$$\gamma_x \le \min\left\{ \frac{\ell^2}{6\omega(\omega+\ell)^2} \,, \frac{64\ell}{(\omega-\ell)^2\sqrt{C_{x_{yz}^1}^2+1}} \,, \frac{1}{72\psi_{\gamma_z}C_{x_z}} \,, \frac{1}{4\ell\sqrt{2\psi_\omega}} \,, \frac{\sqrt{\beta_x N}}{128\sqrt{\psi_\omega V_{2C_g}}} \,, \frac{\sqrt{\beta_y N}}{128\sqrt{2\psi_\omega V_{2C_g}}} \,, \right.$$

$$\frac{\sqrt{\alpha N}}{16\sqrt{\psi_\omega V_{2C_g}}} \cdot \left(\sum_{k'=1}^{K} K\hat{\theta}_{k'} + 5(1+\psi_{\gamma_y})N\right)^{-1/2} \,, \frac{1}{216\tau\eta\sqrt{5\psi_\omega}} \,, \frac{(2L_d+\omega+\ell)^{-1}}{16\psi_{\gamma_y}} \,, \left. \frac{1}{192\tau\eta\sqrt{5\psi_\omega}C_{K,\omega}} \right\} \,,$$

$$\gamma_y = \gamma_x \underbrace{\frac{(\omega-\ell)^2}{64\ell^2}}_{\psi_{\gamma_y}=O(1)} \,, \quad \gamma_z = \gamma_x \underbrace{\frac{(\omega-\ell)^3\mu}{1024\times96\omega\ell^2}}_{\psi_{\gamma_z}=O(1/\kappa)} \,,$$

$$\eta \le \min\left\{ \frac{1}{\sqrt{\beta_x}}, \frac{1}{\sqrt{\beta_y}}, \frac{1}{\sqrt{\alpha}}, \frac{1}{\gamma_z}, \frac{1}{2\gamma_x(\omega+\ell)} \,, \frac{1}{400\tau C_{K,\omega}} \,, \frac{1}{2}\hat{\theta}_k^{1/2} \cdot \left(\sum_{j=k}^{K}\alpha\hat{\theta}_j(2C_g^2)^{j-k}\right)^{-1/2} \,, \right.$$

$$\left. \hat{\nu}_k^{1/2} \cdot \left(\sum_{j=k}^{K}\alpha(\sum_{k'=1}^{K}\hat{\nu}_{k'}+4\hat{\nu}_j)(2C_g^2)^{j-k}\right)^{-1/2} \right\} \,. \tag{44}$$

Therefore, we obtain

$$\frac{1}{T}\sum_{t=0}^{T-1}\left( \mathbb{E}[\|\nabla_x f(G(\bar{x}_t), \bar{y}_t)\|^2] + \kappa\mathbb{E}[\|\nabla_y f(G(\bar{x}_t), \bar{y}_t)\|^2] \right)$$

$$\le \frac{1}{T}\sum_{t=0}^{T-1}\left( 2\mathbb{E}[\|\nabla_x F(G(\bar{x}_t), \bar{y}_t; \bar{z}_t)\|^2] + 2\kappa\mathbb{E}[\|\nabla_y F(G(\bar{x}_t), \bar{y}_t; \bar{z}_t)\|^2] + 2\omega^2\mathbb{E}[\|\bar{x}_t - \bar{z}_t\|^2] \right)$$

$$\le \max\left\{ \frac{8}{\gamma_x\eta} \,, \frac{8\kappa}{\gamma_x\eta\psi_{\gamma_y}} \,, \frac{8\omega}{\gamma_x\eta\psi_{\gamma_z}} \right\} \left( \frac{\mathcal{P}_0 - \mathcal{P}_T}{T} + M_0 + 6\tau E\mathcal{M} \right) \,. \tag{45}$$

Furthermore, at the initial iteration, the batch size is set to $S_0$, then we have

$$\mathcal{P}_0 = \Psi_0 + \frac{3\gamma_x}{\beta_x\eta}\mathbb{E}[\|\bar{p}_0 - \frac{1}{N}\sum_{n=1}^{N}\nabla_x F_{h,0}^{(n)}\|^2] + \frac{6\gamma_y}{\beta_y\eta}\mathbb{E}[\|\bar{q}_0 - \frac{1}{N}\sum_{n=1}^{N}\nabla_y F_{h,0}^{(n)}\|^2]$$

$$+ \sum_{k=1}^{K}\frac{K\gamma_x}{\alpha\eta}\hat{\theta}_k\mathbb{E}[\|\frac{1}{N}\sum_{n=1}^{N}g_{h,0}^{(n,k)} - \frac{1}{N}\sum_{n=1}^{N}h_0^{(n,k)}\|^2] + \sum_{k=1}^{K}\frac{\gamma_x}{\eta N}\hat{\nu}_k\frac{1}{N}\sum_{n=1}^{N}\mathbb{E}[\|g_{h,0}^{(n,k)} - h_0^{(n,k)}\|^2]$$

$$\le \Psi_0 + \frac{3\gamma_x}{\beta_x\eta}\frac{C_K^2\sigma^2}{NS_0} + \frac{6\gamma_y}{\beta_y\eta}\frac{\sigma^2}{NS_0} + \sum_{k=1}^{K}\frac{K\gamma_x}{\alpha\eta}\hat{\theta}_k\frac{\sigma^2}{NS_0} + \sum_{k=1}^{K}\frac{\gamma_x}{\eta N}\hat{\nu}_k\frac{\sigma^2}{S_0} \,.$$

Setting $O(\beta) := O(\beta_x) = O(\beta_y) = O(\alpha)$, and $\omega = O(\ell)$. Then, taking the $O$ notation on the first term, we obtain

$$\max\{\frac{8}{\gamma_x\eta}, \frac{8\kappa}{\gamma_x\eta\psi_{\gamma_y}}, \frac{8\omega}{\gamma_x\eta\psi_{\gamma_z}}\}\frac{\mathcal{P}_0}{T} := O\left(\frac{\kappa\Psi_0}{\gamma_x\eta T}\right) + O\left(\frac{\kappa K}{\beta\eta^2 N S_0 T}\right) .$$

Similarly, taking the $O$ notation on the second and third terms, where $E$, $M_0$, and $\mathcal{M}$ are defined in Eq. (29), (31), (36), we obtain

$$\max\{\frac{8}{\gamma_x\eta}, \frac{8\kappa}{\gamma_x\eta\psi_{\gamma_y}}, \frac{8\omega}{\gamma_x\eta\psi_{\gamma_z}}\}M_0 := O\left(\frac{\kappa\beta\eta^2 K}{N}\right) ,$$

$$\max\{\frac{8}{\gamma_x\eta}, \frac{8\kappa}{\gamma_x\eta\psi_{\gamma_y}}, \frac{8\omega}{\gamma_x\eta\psi_{\gamma_z}}\}6\tau EM := O\left(\frac{\kappa K\tau^2\eta^4\beta\gamma_x^2}{N}\right) + O\left(\kappa\tau^4\eta^6\beta^2\gamma_x^2\right) + O\left(\kappa\tau^2\eta^4\beta^2\right) .$$

Finally, combining the results of above inequalities, we derive

$$\frac{1}{T}\sum_{t=0}^{T-1}\left(\mathbb{E}[\|\nabla_x f(G(\bar{x}_t), \bar{y}_t)\|^2] + \kappa\mathbb{E}[\|\nabla_y f(G(\bar{x}_t), \bar{y}_t)\|^2]\right)$$

$$\leq O\left(\frac{\kappa\Psi_0}{\gamma_x\eta T}\right) + O\left(\frac{\kappa K}{\beta\eta^2 N S_0 T}\right) + O\left(\frac{\kappa\beta\eta^2 K}{N}\right) + O\left(\frac{\kappa K\tau^2\eta^4\beta\gamma_x^2}{N}\right) + O\left(\kappa\tau^4\eta^6\beta^2\gamma_x^2\right) + O\left(\kappa\tau^2\eta^4\beta^2\right) . \quad (46)$$

This completes the proof of Theorem 3.6. $\qquad\square$

## D. Appendix: Stagewise-LSMC-GDA

---
**Algorithm 2** Stagewise-LSMC-GDA
---
**Require:** $\beta_x > 0, \beta_y > 0, \alpha > 0, \eta_{x,r} > 0, \eta_{y,r} > 0$.

1: **for** stage $r = 0, \cdots, R-1$, each device $n$ **do**

2: $\quad x_{r,0}^{(n)} = \tilde{x}_r^{(n)}, \quad y_{r,0}^{(n)} = \tilde{y}_r^{(n)}, \quad h_{r,0}^{(n,k)} = \tilde{h}_r^{(n,k)}$ for $k \in \{0, \cdots, K-1\}, \quad p_{r,0}^{(n)} = \tilde{p}_r^{(n)}, \quad q_{r,0}^{(n)} = \tilde{q}_r^{(n)}$.

3: $\quad$ **for** $t = 0, \cdots, T_r - 1$, **do**

4: $\quad\quad$ Update model parameters:

$\quad\quad\quad x_{r,t+1}^{(n)} = x_{r,t}^{(n)} - \eta_{x,r}p_{r,t}^{(n)}, \quad y_{r,t+1}^{(n)} = y_{r,t}^{(n)} + \eta_{y,r}q_{r,t}^{(n)}$ .

5: $\quad\quad$ Estimate the inner-level function: $h_{r,t+1}^{(0)} = x_{r,t+1}$,

$\quad\quad\quad h_{r,t+1}^{(n,k)} = (1 - \alpha\eta_{x,r}^2)(h_{r,t}^{(n,k)} - g^{(n,k)}(h_{r,t}^{(n,k-1)}; \xi_{r,t+1}^{(n,k)})) + g^{(n,k)}(h_{r,t+1}^{(n,k-1)}; \xi_{r,t+1}^{(n,k)})$, for $k \in \{1, \cdots, K\}$

6: $\quad\quad$ Update stochastic compositional gradients:

$\quad\quad\quad \nabla_x f_{h,r,t+1;t+1}^{(n)} = \nabla g^{(n,1)}(h_{r,t+1}^{(n,0)}; \xi_{r,t+1}^{(n,1)})\cdots\nabla g^{(n,K-1)}(h_{r,t+1}^{(n,K-2)}; \xi_{r,t+1}^{(n,K-1)})\nabla g^{(n,K)}(h_{r,t+1}^{(n,K-1)}; \xi_{r,t+1}^{(n,K)}) \times$

$\quad\quad\quad \nabla_1 f^{(n)}(h_{r,t+1}^{(n,K)}, y_{r,t+1}^{(n)}; \zeta_{r,t+1}^{(n)})$ ,

$\quad\quad\quad \nabla_y f_{h,r,t+1;t+1}^{(n)} = \nabla_2 f^{(n)}(h_{r,t+1}^{(n,K)}, y_{r,t+1}^{(n)}; \zeta_{r,t+1}^{(n)})$ ,

$\quad\quad\quad p_{r,t+1}^{(n)} = (1 - \beta_x\eta_{x,r}^2)(p_{r,t}^{(n)} - \nabla_x f_{h,r,t;t+1}^{(n)}) + \nabla_x f_{h,r,t+1;t+1}^{(n)}$,

$\quad\quad\quad q_{r,t+1}^{(n)} = (1 - \beta_y\eta_{y,r}^2)(q_{r,t}^{(n)} - \nabla_y f_{h,r,t;t+1}^{(n)}) + \nabla_y f_{h,r,t+1;t+1}^{(n)}$,

7: $\quad\quad$ **if** $\mod(r+1, \tau) == 0$ **then**

8: $\quad\quad\quad x_{r,t+1}^{(n)} = \frac{1}{N}\sum_{n'=1}^N x_{r,t+1}^{(n')}, \quad y_{r,t+1}^{(n)} = \frac{1}{N}\sum_{n'=1}^N y_{r,t+1}^{(n')}$,

$\quad\quad\quad p_{r,t+1}^{(n)} = \frac{1}{N}\sum_{n'=1}^N p_{r,t+1}^{(n')}, \quad q_{r,t+1}^{(n)} = \frac{1}{N}\sum_{n'=1}^N q_{r,t+1}^{(n')}$,

9: $\quad\quad\quad$ **for** $k = 1, \cdots, K$ **do**

10: $\quad\quad\quad\quad h_{r,t+1}^{(n,k)} = \frac{1}{N}\sum_{n'=1}^N h_{r,t+1}^{(n',k)}$,

11: $\quad\quad\quad$ **end for**

12: $\quad\quad$ **end if**

13: $\quad$ **end for**

14: $\quad$ Randomly select $(\tilde{x}_{r+1}^{(n)}, \tilde{y}_{r+1}^{(n)}, \tilde{h}_{r+1}^{(n,k)}, \tilde{p}_{r+1}^{(n)}, \tilde{q}_{r+1}^{(n)})$ from $\{(x_{r,t}^{(n)}, y_{r,t}^{(n)}, h_{r,t}^{(n,k)}, p_{r,t}^{(n)}, q_{r,t}^{(n)})\}_{r=0}^{R_t-1}$.

15: **end for**
---

### D.1. Proof of the Theorem 3.8

**Proof Sketch.** To prove Theorem 3.8, we proceed in three steps:

- First, we introduce a potential function together with the necessary definitions.

- Next, we establish two key lemmas based on this potential function, which play a central role in the subsequent analysis.

- Finally, by relating these two lemmas and applying an inductive argument, we complete the proof of Theorem 3.8.

**Unique Challenges.** Establishing convergence rates for PL-PL compositional minimax problems is significantly more challenging than in prior work. In particular, compared to (Shen et al., 2024), our objective involves a compositional structure that introduces bias into the stochastic gradient estimators; therefore, their analysis cannot be directly applied. Moreover, (Zhang et al., 2024) focuses only on the federated NC-SC setting, and it remains unclear how their method can be extended to the PL-PL case. In contrast to (Zhang & Gao, 2025b), which considers only the single-machine setting, our federated setting introduces additional difficulties in the convergence analysis due to communication and consensus errors.

To address these challenges, we develop an induction-based framework to establish the theoretical analysis. At a high level, we follow the framework of (Zhang & Gao, 2025b) to establish the convergence rate of Algorithm 2. However, we highlight several challenges that are unique to the federated setting:

- We introduce both global estimation errors, $\sum_{k=1}^{K} \theta_k \sigma_{r,t+1}^{h_\theta,k}$, and local estimation error, $\sum_{k=1}^{K} \nu_k \sigma_{r,t+1}^{h_\nu,k}$, which must be simultaneously controlled in the analysis, as shown in Eq. (47).

- We additionally need to handle consensus errors induced by federated learning, as characterized in Lemma D.4. In particular, determining appropriate values of $\nu_k$ requires carefully applying this lemma together with the constructions in Eq. (59)-(61), Eq. (66), and Eq. (70).

**Potential function.** Defining the potential function as:

$$\mathcal{P}_{r,t+1} = \mathcal{V}_{r,t+1} + \frac{8}{\beta_x \eta_{x,r}} \sigma_{r,t+1}^x + \frac{2}{\beta_y \eta_{y,r}} \sigma_{r,t+1}^y + \sum_{k=1}^{K} \theta_k \sigma_{r,t+1}^{h_\theta,k} + \sum_{k=1}^{K} \nu_k \sigma_{r,t+1}^{h_\nu,k} \,, \tag{47}$$

where

$$\mathcal{V}_{r,t+1} = \mathbb{E}[\mathcal{L}(x_{r,t+1})] - \mathbb{E}[\mathcal{L}(x_*)] + \frac{c_0 \eta_{x,r}}{\eta_{y,r}} (\mathbb{E}[\mathcal{L}(x_{r,t+1})] - \mathbb{E}[f(G(x_{r,t+1}), y_{r,t+1})]) \,,$$

$$\sigma_{r,t+1}^x = \mathbb{E}[\|\frac{1}{N}\sum_{n=1}^{N} \nabla_x f_{h,r,t+1}^{(n)} - \bar{p}_{r,t+1}\|^2] \,, \quad \sigma_{r,t+1}^y = \mathbb{E}[\|\frac{1}{N}\sum_{n=1}^{N} \nabla_y f_{h,r,t+1}^{(n)} - \bar{q}_{r,t+1}\|^2] \,,$$

$$\sigma_{r,t+1}^{h_\theta,k} = \mathbb{E}[\|\frac{1}{N}\sum_{n=1}^{N} g_{h,r,t+1}^{(n,k)} - \frac{1}{N}\sum_{n=1}^{N} h_{r,t+1}^{(n,k)}\|^2] \,, \quad \sigma_{r,t+1}^{h_\nu,k} = \frac{1}{N}\sum_{n=1}^{N} \mathbb{E}[\|g_{h,r,t+1}^{(n,k)} - h_{r,t+1}^{(n,k)}\|^2] \,. \tag{48}$$

**Key Lemmas.**

**Lemma D.1.** *Given Assumptions 3.1-3.5, by setting*

$$\beta_y = 1280 \frac{L_v^2}{N} \,, \ \beta_x = 51200 \frac{c_0 L_v^2}{N} \,, \ \alpha = 1280 \frac{c_0 L_v^2}{N} \,, \ c_0 = \frac{32 \ell^2}{\mu^2} \,,$$

*$L_v$ is defined in Eq. (63), such that $\eta_{x,r} = \frac{\eta_{y,r}}{10 c_0}$, we then obtain the following bound:*

$$\frac{1}{T_r} \sum_{r=0}^{T_r-1} \left( \mathbb{E}[\|\nabla \mathcal{L}(\bar{x}_{r,t})\|^2] + \frac{c_0 \eta_{x,r}}{\eta_{y,r}} \mathbb{E}[\|\nabla_y f(G(\bar{x}_{r,t}), \bar{y}_{r,t})\|^2] \right)$$

$$\leq \frac{40 c_0 \mathcal{V}_{r,0}}{\eta_{y,r} T_r} + \frac{80 c_0}{\beta_y \eta_{y,r}^2 T_r} \left( \sigma_{r,0}^x + \sigma_{r,0}^y + 5\sigma_{r,0}^{h_\theta} + 47\sigma_{r,0}^{h_\nu} \right) + 245 c_0 \beta_y \eta_{y,r}^2 L_v^2 \sigma^2 C_\tau \,,$$

*where*

$$C_\tau = \left( \frac{1}{N} + \frac{145}{N} \tau^2 \eta_{y,r}^2 + 20\tau^4 \eta_{y,r}^4 + 20\tau^2 \eta_{y,r}^2 \right) . \tag{49}$$

**Lemma D.2.** *Given Assumptions 3.1-3.5, the collective estimation error can be bounded as:*

$$\sigma_{r+1,0}^x + \sigma_{r+1,0}^y + 5\sigma_{r+1,0}^{h_\theta} + 47\sigma_{r+1,0}^{h_\nu}$$

$$\leq \frac{320c_0}{\beta_y \eta_{y,r}^2 T_r} \left( \sigma_{r,0}^x + \sigma_{r,0}^y + 5\sigma_{r,0}^{h_\theta} + 47\sigma_{r,0}^{h_\nu} \right) + \frac{45c_0 \mathcal{V}_{r,0}}{\eta_{y,r} T_r} + 480c_0 \beta_y \eta_{y,r}^2 L_v^2 \sigma^2 C_\tau .$$

The detailed proof of Lemma D.1 is provided in Section D.3, while that of Lemma D.2 appears in Section D.4.

**Inductive Argument for Theorem 3.8**

*Proof.* Under the two-sided PL condition, we can get

$$2\mu \left( \mathbb{E}[\mathcal{L}(\bar{x}_{r,t})] - \mathcal{L}(\bar{x}_*) + \frac{c_0 \eta_{x,r}}{\eta_{y,r}} (\mathbb{E}[\mathcal{L}(\bar{x}_{r,t})] - \mathbb{E}[f(G(\bar{x}_{r,t}), \bar{y}_{r,t})]) \right)$$

$$\leq \mathbb{E}[\|\nabla \mathcal{L}(\bar{x}_{r,t})\|^2] + \frac{c_0 \eta_{x,r}}{\eta_{y,r}} \mathbb{E}[\|\nabla_y f(G(\bar{x}_{r,t}), \bar{y}_{r,t})\|^2] .$$

Taking into account the random sampling in each outer iteration, we further derive

$$\mathcal{V}_{r+1,0} = \frac{1}{T_r} \sum_{t=0}^{T_r-1} \left( \mathbb{E}[\mathcal{L}(\bar{x}_{r,t})] - \mathcal{L}(\bar{x}_*) + \frac{c_0 \eta_{x,r}}{\eta_{y,r}} (\mathbb{E}[\mathcal{L}(\bar{x}_{r,t})] - \mathbb{E}[f(g(\bar{x}_{r,t}), \bar{y}_{r,t})]) \right)$$

$$\leq \frac{1}{2\mu} \frac{1}{T_r} \sum_{t=0}^{T_r-1} \left( \mathbb{E}[\|\nabla \mathcal{L}(\bar{x}_{r,t})\|^2] + \frac{c_0 \eta_{x,r}}{\eta_{y,r}} \mathbb{E}[\|\nabla_y f(g(\bar{x}_{r,t}), \bar{y}_{r,t})\|^2] \right)$$

$$\leq \frac{1}{2\mu} \left( \frac{40c_0 \mathcal{V}_{r,0}}{\eta_{y,r} T_r} + \frac{80c_0}{\beta_y \eta_{y,r}^2 T_r} (\sigma_{r,0}^x + \sigma_{r,0}^y + 5\sigma_{r,0}^{h_\theta} + 47\sigma_{r,0}^{h_\nu}) + 245c_0 \beta_y \eta_{y,r}^2 L_v^2 \sigma^2 C_\tau \right.$$

$$\leq \frac{1}{\mu} \left( \frac{45c_0 \mathcal{V}_{r,0}}{\eta_{y,r} T_r} + \frac{320c_0}{\beta_y \eta_{y,r}^2 T_r} (\sigma_{r,0}^x + \sigma_{r,0}^y + 5\sigma_{r,0}^{h_\theta} + 47\sigma_{r,0}^{h_\nu}) + 480c_0 \beta_y \eta_{y,r}^2 L_v^2 \sigma^2 C_\tau \right) . \tag{50}$$

At $t = 0$, we have $\sigma_{0,0}^x + \sigma_{0,0}^y + 5\sigma_{0,0}^{h_\theta} + 47\sigma_{0,0}^{h_\nu} = 54L_v^2 \sigma^2$. Then, by Eq. (50) and Lemma D.2, we obtain

$$\sigma_{1,0}^x + \sigma_{1,0}^y + 5\sigma_{1,0}^{h_\theta} + 47\sigma_{1,0}^{h_\nu} \leq \frac{17280c_0 L_v^2}{\beta_y \eta_{y,0}^2 T_r} \sigma^2 + \frac{45c_0 \mathcal{V}_{0,0}}{\eta_{y,0} T_r} + 480c_0 \beta_y \eta_{y,0}^2 L_v^2 \sigma^2 C_\tau ,$$

$$\mathcal{V}_{1,0} \leq \frac{1}{\mu} \left( \frac{17280c_0 L_v^2}{\beta_y \eta_{y,0}^2 R_0} \sigma^2 + \frac{45c_0 \mathcal{V}_{0,0}}{\eta_{y,0} R_0} + 480c_0 \beta_y \eta_{y,0}^2 L_v^2 \sigma^2 C_\tau \right) .$$

By setting $\eta_{y,0} = \frac{N}{40\tau L_v}$ and $R_0 = \max\{300\tau^2, \frac{25\mathcal{V}_{0,0}}{NL_v \sigma^2}\}$, together with $\beta_y = 1280\frac{L_v^2}{N}$, we obtain

$$\sigma_{1,0}^x + \sigma_{1,0}^y + 27\sigma_{1,0}^{h_\theta} + 52\sigma_{1,0}^{h_\nu} \leq \frac{17280 \times 5c_0 \tau^2 L_v^2}{4R_0} \sigma^2 + \frac{45 \times 40c_0 L_v \mathcal{V}_{0,0}}{R_0} + 384c_0 L_v^2 \sigma^2 C_\tau \leq 600c_0 L_v^2 \sigma^2 C_\tau ,$$

$$\mathcal{V}_{1,0} \leq \frac{600c_0 L_v^2 \sigma^2 C_\tau}{\mu} .$$

Therefore, we define $\epsilon_1 \triangleq 600c_0 L_v^2 \sigma^2 C_\tau / \mu$ so that

$$\sigma_{1,0}^x + \sigma_{1,0}^y + 5\sigma_{1,0}^{h_\theta} + 47\sigma_{1,0}^{h_\nu} \leq \mu\epsilon_1 , \quad \mathcal{V}_{1,0} \leq \epsilon_1 .$$

This provide an important connection between $\mathcal{V}_{r,0}$ and $\sigma_{r,0}^x + \sigma_{r,0}^y + 5\sigma_{r,0}^{h_\theta} + 47\sigma_{r,0}^{h_\nu}$. Building on this finding, we apply an inductive argument to establish the desired result.

Suppose $\sigma_{r,0}^x + \sigma_{r,0}^y + 5\sigma_{r,0}^{h_\theta} + 47\sigma_{r,0}^{h_\nu} \le \mu\epsilon_r$ and $\mathcal{V}_{r,0} \le \epsilon_r$, we will prove that $\sigma_{r+1,0}^x + \sigma_{r+1,0}^y + 5\sigma_{r+1,0}^{h_\theta} + 47\sigma_{r+1,0}^{h_\nu} \le \mu\epsilon_r/2$ and $\mathcal{V}_{r+1,0} \le \epsilon_r/2$. To begin, we have

$$\sigma_{r+1,0}^x + \sigma_{r+1,0}^y + 5\sigma_{r+1,0}^{h_\theta} + 47\sigma_{r+1,0}^{h_\nu} \le \frac{320c_0 L_v^2}{\beta_y \eta_{y,r}^2 T_r}\mu\epsilon_r + \frac{45c_0}{\eta_{y,r} T_r}\epsilon_r + 480c_0 \beta_y \eta_{y,r}^2 L_v^2 \sigma^2 C_\tau .$$

To ensure $\sigma_{r+1,0}^x + \sigma_{r+1,0}^y + 5\sigma_{r+1,0}^{h_\theta} + 47\sigma_{r+1,0}^{h_\nu} \le \mu\epsilon_r/2$, we enforce each component to be bounded by $\epsilon_r/6$. In particular, by setting

$$\frac{320c_0 L_v^2}{\beta_y \eta_{y,r}^2 T_r}\mu\epsilon_r \le \frac{\mu\epsilon_r}{6} , \quad \frac{45c_0}{\eta_{y,r} T_r}\epsilon_r \le \frac{\mu\epsilon_r}{6} , \quad 480c_0 \beta_y \eta_{y,r}^2 L_v^2 \sigma^2 C_\tau \le \frac{\mu\epsilon_r}{6} ,$$

with $\tau = O(\frac{1}{N}\sqrt{\frac{c_0}{\mu\epsilon_r}})$, we obtain

$$\eta_{y,r} = \frac{\sqrt{\mu\epsilon_r}N}{3840\sqrt{c_0}L_v^2\sigma} , \quad T_r \ge \max\{\frac{270 \times 3840c_0 L_v^2 \sqrt{c_0}\sigma}{\mu\sqrt{\mu\epsilon_r}N} , \frac{5760 \times 3840c_0^2 L_v^4 \sigma^2}{\mu\epsilon_r N}\} .$$

Therefore, under the above condition, we derive

$$\sigma_{r+1,0}^x + \sigma_{r+1,0}^y + 5\sigma_{r+1,0}^{h_\theta} + 47\sigma_{r+1,0}^{h_\nu} \le \frac{\mu\epsilon_r}{6} + \frac{\mu\epsilon_r}{6} + \frac{\mu\epsilon_r}{6} \le \frac{\mu\epsilon_r}{2} ,$$

and

$$\mathcal{V}_{r+1,0} \le \frac{1}{\mu}\left(\frac{320c_0 L_v^2}{\beta_y \eta_{y,r}^2 T_r}\mu\epsilon_r + \frac{45c_0^2}{\eta_{y,r} T_r}\epsilon_r + 480c_0 \beta_y \eta_{y,r} L_v^2 \sigma^2 C_\tau\right) \le \frac{\epsilon_r}{2} .$$

Since $\epsilon_r = \frac{\epsilon_1}{2^{r-1}} = \frac{600c_0 L_v^2 \sigma^2 C_\tau}{2^{r-1}\mu} = O\left(\frac{c_0 L_v^2 \sigma^2}{2^{r-1}\mu}\right)$, we further deduce

$$T_r = O\left(\frac{c_0^2 L_v^4 \sigma^2}{\mu\epsilon_r N}\right) = O\left(\frac{c_0^2 L_v^4 \sigma^2}{\mu N} \times \frac{2^{r-1}\mu}{c_0 L_v^2 \sigma^2}\right) = O\left(\frac{c_0}{N} \times 2^{r-1}\right)$$

$$T_r = O\left(\frac{c_0 L_v^2 \sqrt{c_0}\sigma}{\mu\sqrt{\mu\epsilon_r}N}\right) = O\left(\frac{c_0 L_v^2 \sqrt{c_0}\sigma}{\mu\sqrt{\mu}N} \times \frac{\sqrt{2^{r-1}\mu}}{\sqrt{c_0}L_v\sigma}\right) = O\left(\frac{c_0}{\mu N} \times \sqrt{2^{r-1}}\right) .$$

Accordingly, we may set $T_r = O(\frac{c_0}{\mu N} \times 2^{r-1})$. Finally, to achieve $\mathcal{V}_{r,0} \le \epsilon$, one need $\frac{\epsilon_1}{2^{R-1}} = \epsilon$, such that $R = \log\frac{2\epsilon_1}{\epsilon}$. Thus, the total number of iterations is

$$O(T_0 + \sum_{r=1}^R T_r) = O\left(\max\{300\tau^2, \frac{25\mathcal{V}_{0,0}}{NL_v\sigma^2}\} + \sum_{r=1}^R \frac{c_0}{\mu N} \times 2^{r-1}\right) = \left(\frac{c_0\epsilon_1}{\mu\epsilon N}\right) = O\left(\frac{1}{\mu^6\epsilon N}\right) . \tag{51}$$

Additionally, we obtain

$$\tau = O\left(\frac{1}{N}\sqrt{\frac{c_0}{\mu\epsilon_r}}\right) = O\left(\sqrt{\frac{c_0}{\mu}} \times \sqrt{\frac{2^{r-1}\mu}{c_0 L_v^2 \sigma^2}}\right) = O\left(\frac{\sqrt{2^{r-1}}}{NL_v}\right) ,$$

$$\eta_{y,r} = O\left(\frac{\sqrt{\mu\epsilon_r}N}{\sqrt{c_0}L_v^2\sigma}\right) = O\left(\frac{\sqrt{\mu}N}{\sqrt{c_0}L_v^2\sigma} \times \sqrt{\frac{c_0 L_v^2 \sigma^2}{2^{r-1}\mu}}\right) = O(\frac{N}{\sqrt{2^{r-1}}L_v}) ,$$

and it follows that $\eta_{x,r} = O(\frac{\mu^2 N}{\sqrt{2^{r-1}}L_v})$. Moreover, it is easy to know that $T_r/\tau = O(2^{(r-1)/2})$. Then, similar to Eq. (51), we can obtain that the communication complexity is in the order of $O(1/\epsilon^{1/2})$.

Thus, the proof of Theorem 3.8 is completed. $\qquad\square$

### D.2. Useful Lemmas

To prove the two key lemmas above, we first present several useful lemmas.

**Lemma D.3.** *(Zhang & Gao, 2025b) Given Assumptions 3.1-3.5 and $\eta_{x,r} \leq \min\{\frac{1}{2L_{\mathcal{L}}}, \frac{1}{16\ell}\}$, $\eta_{y,r} \leq \frac{1}{2\ell}$, we derive:*

$$\mathbb{E}[\mathcal{L}(\bar{x}_{r,t+1})] - \mathbb{E}[\mathcal{L}(\bar{x}_{r,t})] \leq -\frac{\eta_{x,r}}{2}\mathbb{E}[\|\nabla\mathcal{L}(\bar{x}_{r,t})\|^2] - \frac{\eta_{x,r}}{4}\mathbb{E}[\|\bar{p}_{r,t}\|^2]$$
$$+ \eta_{x,r}\mathbb{E}[\|\nabla\mathcal{L}(\bar{x}_{r,t}) - \nabla_x f(G(\bar{x}_{r,t}),\bar{y}_{r,t})\|^2] + \eta_{x,r}\mathbb{E}[\|\nabla_x f(G(\bar{x}_{r,t}),\bar{y}_{r,t}) - \bar{p}_{r,t}\|^2],$$

*and*

$$\mathbb{E}[\mathcal{L}(\bar{x}_{r,t+1}) - f(G(\bar{x}_{r,t+1}),\bar{y}_{r,t+1})] - \mathbb{E}[\mathcal{L}(\bar{x}_{r,t}) - f(G(\bar{x}_{r,t}),\bar{y}_{r,t})]$$
$$\leq 4\eta_{x,r}\mathbb{E}[\|\nabla\mathcal{L}(\bar{x}_{r,t}) - \nabla_x f(G(\bar{x}_{r,t}),\bar{y}_{r,t})\|^2] + \frac{5\eta_{x,r}}{2}\mathbb{E}[\|\nabla\mathcal{L}(\bar{x}_{r,t})\|^2] - \frac{\eta_{y,r}}{4}\mathbb{E}[\|\nabla_y f(G(\bar{x}_{r,t}),\bar{y}_{r,t})\|^2]$$
$$+ \frac{3\eta_{x,r}}{2}\mathbb{E}[\|\nabla_x f(G(\bar{x}_{r,t}),\bar{y}_{r,t}) - \bar{p}_{r,t}\|^2] + \eta_{y,r}\mathbb{E}[\|\nabla_y f(G(\bar{x}_{r,t}),\bar{y}_{r,t}) - \bar{q}_{r,t}\|^2]$$
$$- \frac{\eta_{x,r}}{8}\mathbb{E}[\|\bar{p}_{r,t}\|^2] - \frac{\eta_{y,r}}{4}\mathbb{E}[\|\bar{q}_{r,t}\|^2].$$

*Proof.* This lemma follows directly from Lemma C.3 and C.5 in (Zhang & Gao, 2025b). Since our algorithm operates in a federated learning setting, we do not decompose the terms $\mathbb{E}[\|\nabla_x f(G(\bar{x}_{r,t}),\bar{y}_{r,t}) - \bar{p}_{r,t}\|^2]$ and $\mathbb{E}[\|\nabla_y f(G(\bar{x}_{r,t}),\bar{y}_{r,t}) - \bar{q}_{r,t}\|^2]$ for now. □

**Lemma D.4.** *Given Assumptions 3.1-3.5, we derive **overall consensus error**: By setting*

$$\eta_{x,r} \leq \frac{1}{400\tau L_v}, \quad \eta_{y,r} \leq \frac{1}{400\tau L_v}, \quad \{\alpha, \beta_x, \beta_y\} = \frac{L_v^2}{N}, \tag{52}$$

*we have*

$$\frac{1}{T_r}\sum_{t=0}^{T_r-1}\frac{1}{N}\sum_{n=1}^{N}\left(\mathbb{E}[\|p_{r,t}^{(n)} - \bar{p}_{r,t}\|^2] + \mathbb{E}[\|q_{r,t}^{(n)} - \bar{q}_{r,t}\|^2] + \sum_{k=1}^{K}\mathbb{E}[\|h_{r,t}^{(n,k)} - \bar{h}_{r,t}^{(k)}\|^2]\right)$$
$$\leq 6\tau\tilde{E}_p\frac{1}{T_r}\sum_{t=0}^{T_r-1}\mathbb{E}[\|\bar{p}_{r,t}\|^2] + 6\tau\tilde{E}_q\frac{1}{T_r}\sum_{t=0}^{T_r-1}\mathbb{E}[\|\bar{q}_{r,t}\|^2] + 6\tau\tilde{E}_g\frac{1}{T_r}\sum_{t=0}^{T_r-1}\frac{1}{N}\sum_{n=1}^{N}\mathbb{E}[\|g_{h,r,t}^{(n,k)} - h_{r,t}^{(n,k)}\|^2] + 6\tau\tilde{E}.$$

*where the coefficient of each term is denoted as*

$$\tilde{E}_p = 8\tau\eta_{x,r}^2 C_K^2 \sum_{k=0}^{K}(2C_g^2)^k + 8\tau\eta_{x,r}^2 L_f^2(2C_g^2)^K + 4\tau\eta_{x,r}^2 \sum_{k=1}^{K}(2C_g^2)^k,$$
$$\tilde{E}_q = 8\tau\eta_{y,r}^2 C_K^2 + 8\tau\eta_{y,r}^2 L_f^2,$$
$$\tilde{E}_g = \sum_{k=1}^{K}\alpha^2\eta_{x,r}^4\left(8\tau C_K^2\sum_{j=k}^{K}(2C_g^2)^{j-k} + 4\tau\sum_{j=k}^{K}(2C_g^2)^{j-k} + 8\tau L_f^2(2C_g^2)^{K-k}\right),$$
$$\tilde{E} = 4\tau\alpha^2\eta_{x,r}^4(4C_K^2+1)\sum_{k=1}^{K}\sum_{j=1}^{k}(2C_g^2)^{j-1}\sigma^2 + 8\tau\beta_x^2\eta_{x,r}^4 C_K^2\sigma^2 + 8\tau\beta_y^2\eta_{y,r}^4\sigma^2 + 8\tau\alpha^2\eta_{x,r}^4\sigma^2 K. \tag{53}$$

*Proof.* The proof of this lemma follows the same arguments as Lemma C.14, with the only difference being that it does not involve a smoothed variable. For simplicity, the details are omitted. □

### D.3. Proof of Lemma D.1

*Proof.* By applying Lemma C.7, C.8, C.9, C.10, and D.3 to the potential function in Eq. (47), we obtain

$$\mathcal{P}_{r,t+1} - \mathcal{P}_{r,t} \leq -\frac{\eta_{x,r}}{4}\mathbb{E}[\|\nabla\mathcal{L}(\bar{x}_{r,t})\|^2] - \frac{c_0\eta_{x,r}}{4}\mathbb{E}[\|\nabla_y f(G(\bar{x}_{r,t}),\bar{y}_{r,t})\|^2]$$

$$+ (\eta_{x,r} + \frac{4c_0\eta_{x,r}^2}{\eta_{y,r}})\mathbb{E}[\|\nabla\mathcal{L}(\bar{x}_{r,t}) - \nabla_x f(G(\bar{x}_{r,t}), \bar{y}_{r,t})\|^2] + (\eta_{x,r} + \frac{3c_0\eta_{x,r}^2}{2\eta_{y,r}})\mathbb{E}[\|\nabla_x f(G(\bar{x}_{r,t}), \bar{y}_{r,t}) - \bar{p}_{r,t}\|^2]$$

$$+ c_0\eta_{x,r}\mathbb{E}[\|\nabla_y f(G(\bar{x}_{r,t}), \bar{y}_{r,t}) - \bar{q}_{r,t}\|^2] - 8\eta_{x,r}\mathbb{E}[\|\bar{p}_{r,t} - \frac{1}{N}\sum_{n=1}^{N}\nabla_x f_{h,r,t}^{(n)}\|^2] - 2\eta_{y,r}\mathbb{E}[\|\bar{q}_{r,t} - \frac{1}{N}\sum_{n=1}^{N}\nabla_y f_{h,r,t}^{(n)}\|^2]$$

$$+ (\tilde{M}_{p,c} - \frac{\eta_{x,r}}{4} - \frac{c_0\eta_{x,r}^2}{8\eta_{y,r}})\mathbb{E}[\|\bar{p}_{r,t}\|^2] + (\tilde{M}_{q,c} - \frac{c_0\eta_{x,r}}{4})\mathbb{E}[\|\bar{q}_{r,t}\|^2] + \tilde{M}_{g,l}\frac{1}{N}\sum_{n=1}^{N}\mathbb{E}[\|g_{h,r,t}^{(n,k)} - h_{r,t}^{(n,k)}\|^2]$$

$$+ \sum_{k=1}^{K}\Big(2\alpha^2\eta_{x,r}^4\sum_{j=k+1}^{K}\theta_j(2C_g^2)^{j-k} - \alpha\eta_{x,r}^2\theta_k\Big)\mathbb{E}[\|\frac{1}{N}\sum_{n=1}^{N}g_{h,r,t}^{(n,k)} - \frac{1}{N}\sum_{n=1}^{N}h_{r,t}^{(n,k)}\|^2]$$

$$+ \tilde{M}_{p,c}\frac{1}{N}\sum_{n=1}^{N}\mathbb{E}[\|\bar{p}_{r,t} - p_{r,t}^{(n)}\|^2] + \tilde{M}_{q,c}\frac{1}{N}\sum_{n=1}^{N}\mathbb{E}[\|\bar{q}_{r,t} - q_{r,t}^{(n)}\|^2] + \tilde{M}_0 , \tag{54}$$

where, for readability, we use the following notation for the coefficients of each term and the constant term:

$$\tilde{M}_{p,c} = \frac{32\eta_{x,r}}{\beta_x N}C_K^2\sum_{k=0}^{K}(2C_g^2)^k + \frac{8\eta_{x,r}^2}{\beta_y\eta_{y,r}N}L_f^2(2C_g^2)^K + 2\eta_{x,r}^2\sum_{k=1}^{K}(\frac{\theta_k}{N} + \nu_k)(2C_g^2)^k ,$$

$$\tilde{M}_{q,c} = \frac{32\eta_{y,r}^2}{\beta_x\eta_{x,r}N}C_K^2 + \frac{8\eta_{y,r}}{\beta_y N}L_f^2 ,$$

$$\tilde{M}_{g,l} = \alpha^2\eta_{x,r}^4\sum_{k=1}^{K}\Big(\frac{32C_K^2}{\beta_x\eta_{x,r}N}\sum_{j=k}^{K}(2C_g^2)^{j-k} + \frac{8L_f^2}{\beta_y\eta_{y,r}N}(2C_g^2)^{K-k} + \sum_{j=k+1}^{K}2\nu_j(2C_g^2)^{j-k} - \frac{\nu_k}{\alpha\eta_{x,r}^2}\Big) ,$$

$$\tilde{M}_0 = \frac{32\alpha^2\eta_{x,r}^4}{\beta_x\eta_{x,r}}C_K^2\sum_{k=1}^{K}\sum_{j=1}^{k}(2C_g^2)^{j-1}\frac{\sigma^2}{N} + 16\beta_x\eta_{x,r}^3 C_K^2\frac{\sigma^2}{N} + \frac{8\alpha^2\eta_{x,r}^4}{\beta_y\eta_{y,r}}L_f^2\sum_{k=1}^{K}(2C_g^2)^{k-1}\frac{\sigma^2}{N} + 4\beta_y\eta_{y,r}^3\frac{\sigma^2}{N}$$

$$+ 2\alpha^2\eta_{x,r}^4\sum_{k=1}^{K}\theta_k\sum_{j=0}^{k-1}(2C_g^2)^j\frac{\sigma^2}{N} + 2\alpha^2\eta_{x,r}^4\sigma^2\sum_{k=1}^{K}\nu_k\sum_{j=0}^{k-1}(2C_g^2)^j . \tag{55}$$

Building on Eq. (33) and (34), we derive the expansions of both $\mathbb{E}[\|\nabla_x f(G(\bar{x}_{r,t}), \bar{y}_{r,t}) - \bar{p}_{r,t}\|^2]$ and $\mathbb{E}[\|\nabla_y f(G(\bar{x}_{r,t}), \bar{y}_{r,t}) - \bar{q}_{r,t}\|^2]$, which yield:

$$\mathbb{E}[\|\nabla_x f(G(\bar{x}_{r,t}), \bar{y}_{r,t}) - \bar{p}_{r,t}\|^2]$$

$$\leq 4C_K^2\frac{1}{N}\sum_{n=1}^{N}\mathbb{E}[\|\bar{x}_{r,t} - x_{r,t}^{(n)}\|^2] + 4C_K^2\frac{1}{N}\sum_{n=1}^{N}\mathbb{E}[\|\bar{y}_{r,t} - y_{r,t}^{(n)}\|^2] + 4C_K^2\frac{1}{N}\sum_{n=1}^{N}\sum_{k=1}^{K}\mathbb{E}[\|\bar{h}_{r,t}^{(k)} - h_{r,t}^{(n,k)}\|^2]$$

$$+ 4K\sum_{k=1}^{K}A_k\mathbb{E}[\|\frac{1}{N}\sum_{n=1}^{N}g_{h,r,t}^{(n,k)} - \frac{1}{N}\sum_{n=1}^{N}h_{r,t}^{(n,k)}\|^2] + 4\mathbb{E}[\|\frac{1}{N}\sum_{n=1}^{N}\nabla_x f_{h,r,t}^{(n)} - \bar{p}_{r,t}\|^2] ,$$

$$\mathbb{E}[\|\nabla_y f(G(\bar{x}_{r,t}), \bar{y}_{r,t}) - \bar{q}_{r,t}\|^2]$$

$$\leq 4L_f^2 C_G^2\frac{1}{N}\sum_{n=1}^{N}\mathbb{E}[\|\bar{x}_{r,t} - x_{r,t}^{(n)}\|^2] + 4L_f^2\frac{1}{N}\sum_{n=1}^{N}\mathbb{E}[\|\bar{y}_{r,t} - y_{r,t}^{(n)}\|^2] + 4L_f^2\frac{1}{N}\sum_{n=1}^{N}\mathbb{E}[\|\bar{h}_{r,t}^{(K)} - h_{r,t}^{(n,K)}\|^2]$$

$$+ 4L_f^2 K\sum_{k=1}^{K}C_g^{2(K-k)}\mathbb{E}[\|\frac{1}{N}\sum_{n=1}^{N}g_{h,r,t}^{(n,k)} - \frac{1}{N}\sum_{n=1}^{N}h_{r,t}^{(n,k)}\|^2] + 4\mathbb{E}[\|\frac{1}{N}\sum_{n=1}^{N}\nabla_y f_{h,r,t}^{(n)} - \bar{q}_{r,t}\|^2] . \tag{56}$$

Consequently, with $\eta_{x,r} = \frac{\eta_{y,r}}{10c_0}$, the coefficients of $\mathbb{E}[\|\frac{1}{N}\sum_{n=1}^{N}\nabla_x f_{h,r,t}^{(n)} - \bar{p}_{r,t}\|^2]$ and $\mathbb{E}[\|\frac{1}{N}\sum_{n=1}^{N}\nabla_y f_{h,r,t}^{(n)} - \bar{q}_{r,t}\|^2]$ can be expressed as:

$$4(\eta_{x,r} + \frac{3c_0\eta_{x,r}^2}{2\eta_{y,r}}) - 8\eta_{x,r} = -\frac{17}{5}\eta_{x,r} , \quad 4c_0\eta_{x,r} - 2\eta_{y,r} = -\frac{8}{5}\eta_{y,r} . \tag{57}$$

For the term $\mathbb{E}[\|\frac{1}{N}\sum_{n=1}^{N} g_{h,r,t}^{(n,k)} - \frac{1}{N}\sum_{n=1}^{N} h_{r,t}^{(n,k)}\|^2]$, we following similar steps of Eq. (35), by setting

$$\theta_k = \frac{Kc_0}{\alpha\eta_{x,r}}(16A_k + 10L_f^2 C_g^{2(K-k)}) \triangleq \frac{Kc_0\hat{\theta}_k}{\alpha\eta_{x,r}} \ ,$$

$$\eta_{x,r} \le \frac{1}{2}\hat{\theta}_k^{1/2} \cdot \Big(\sum_{j=k}^{K} \alpha\hat{\theta}_j(2C_g^2)^{j-k}\Big)^{-1/2} \ ,$$

we arrive at

$$4KA_k(\eta_{x,r} + \frac{3c_0\eta_{x,r}^2}{2\eta_{y,r}}) + 4L_f^2 KC_g^{2(K-k)}c_0\eta_{x,r} + 2\alpha^2\eta_{x,r}^4 \sum_{j=k+1}^{K} \theta_j(2C_g^2)^{j-k} - \alpha\eta_{x,r}^2\theta_k$$

$$\le \frac{46}{10}KA_k\eta_{x,r} + 4L_f^2 KC_g^{2(K-k)}c_0\eta_{x,r} - \frac{1}{2}\alpha\eta_{x,r}^2\frac{K}{\alpha\eta_{x,r}}(16A_k + 10c_0 L_f^2 C_g^{2(K-k)})$$

$$= -\frac{34}{10}\eta_{x,r}KA_k - c_0\eta_{x,r}L_f^2 KC_g^{2(K-k)} \ . \tag{58}$$

Moreover, setting $c_0 = \frac{32\ell^2}{\mu^2}$ yields $\Big(\eta_{x,r} + \frac{3c_0\eta_{x,r}^2}{\eta_{y,r}}\Big)\frac{\ell^2}{\mu^2} \le \frac{c_0\eta_{x,r}}{16}$, and $-\frac{\eta_{x,r}}{4} \ge -\frac{\eta_{y,r}}{8}$. Then, invoking Lemma A.7 in (Chen et al., 2022), we have $(\eta_{x,r} + \frac{3c_0\eta_{x,r}^2}{\eta_{y,r}})\mathbb{E}[\|\nabla\mathcal{L}(\bar{x}_{r,t}) - \nabla_x f(G(\bar{x}_{r,t}), \bar{y}_{r,t})\|^2] \le \frac{c_0\eta_{x,r}}{16}\mathbb{E}[\|\nabla_y f(G(\bar{x}_{r,t}), \bar{y}_{r,t})\|^2]$. It directly leads to

$$-\frac{c_0\eta_{x,r}}{4}\mathbb{E}[\|\nabla_y f(G(\bar{x}_{r,t}), \bar{y}_{r,t})\|^2] + (\eta_{x,r} + \frac{3c_0\eta_{x,r}^2}{\eta_{y,r}})\mathbb{E}[\|\nabla\mathcal{L}(\bar{x}_{r,t}) - \nabla_x f(G(\bar{x}_{r,t}), \bar{y}_{r,t})\|^2]$$

$$\le -\frac{c_0\eta_{x,r}}{4}\mathbb{E}[\|\nabla_y f(G(\bar{x}_{r,t}), \bar{y}_{r,t})\|^2] + \frac{c_0\eta_{x,r}}{16}\mathbb{E}[\|\nabla_y f(G(\bar{x}_{r,t}), \bar{y}_{r,t})\|^2]$$

$$= -\Big(\frac{\eta_{y,r}}{8}\frac{c_0\eta_{x,r}}{\eta_{y,r}} + \frac{c_0\eta_{x,r}}{16}\Big)\mathbb{E}[\|\nabla_y f(G(\bar{x}_{r,t}), \bar{y}_{r,t})\|^2] \le -\Big(\frac{\eta_{x,r}}{4}\frac{c_0\eta_{x,r}}{\eta_{y,r}} + \frac{c_0\eta_{x,r}}{16}\Big)\mathbb{E}[\|\nabla_y f(G(\bar{x}_{r,t}), \bar{y}_{r,t})\|^2] \ .$$

Then, by summing over $t = \{0, \cdots, T_r - 1\}$, combining Eq. (57) and (58), applying the overall consensus error Lemma D.4 to Eq. (54) and introducing

$$\tilde{\mathcal{M}} = \frac{32\eta_{x,r}}{\beta_x N}C_K^2 \sum_{k=0}^{K}(2C_g^2)^k + \frac{8\eta_{x,r}^2}{\beta_y\eta_{y,r}N}L_f^2(2C_g^2)^K + 2\eta_{x,r}^2 \sum_{k=1}^{K}(\frac{\theta_k}{N} + \nu_k)(2C_g^2)^k$$

$$+ \frac{32\eta_{y,r}^2}{\beta_x\eta_{x,r}N}C_K^2 + \frac{8\eta_{y,r}}{\beta_y N}L_f^2 + 4C_K^2\Big(\frac{23\eta_{x,r}}{5} + 4c_0\eta_{x,r}\Big)(\tau^2\eta_{x,r}^2 + \tau^2\eta_{y,r}^2 + 1) \ , \tag{59}$$

we obtain

$$\frac{\mathcal{P}_{r,T_r} - \mathcal{P}_{r,0}}{T} \le -\frac{\eta_{x,r}}{4}\frac{1}{T_r}\sum_{t=0}^{T_r-1}\mathbb{E}[\|\nabla\mathcal{L}(\bar{x}_{r,t})\|^2] - \Big(\frac{\eta_{x,r}}{4}\frac{c_0\eta_{x,r}}{\eta_{y,r}} + \frac{c_0\eta_{x,r}}{16}\Big)\frac{1}{T_r}\sum_{t=0}^{T_r-1}\mathbb{E}[\|\nabla_y f(G(\bar{x}_{r,t}), \bar{y}_{r,t})\|^2]$$

$$- \frac{17}{5}\eta_{x,r}\mathbb{E}[\|\bar{p}_{r,t} - \frac{1}{N}\sum_{n=1}^{N}\nabla_x f_{h,r,t}^{(n)}\|^2] - \frac{8}{5}\eta_{y,r}\mathbb{E}[\|\bar{q}_{r,t} - \frac{1}{N}\sum_{n=1}^{N}\nabla_y f_{h,r,t}^{(n)}\|^2]$$

$$- \sum_{k=1}^{K}\Big(\frac{34}{10}\eta_{x,r}KA_k + c_0\eta_{x,r}L_f^2 KC_g^{2(K-k)}\Big)\mathbb{E}[\|\frac{1}{N}\sum_{n=1}^{N}g_{h,r,t}^{(n,k)} - \frac{1}{N}\sum_{n=1}^{N}h_{r,t}^{(n,k)}\|^2]$$

$$+ (\tilde{M}_{p,c} + 6\tau\tilde{E}_p\tilde{\mathcal{M}} - \frac{\eta_{x,r}}{4} - \frac{c_0\eta_{x,r}^2}{8\eta_{y,r}})\mathbb{E}[\|\bar{p}_{r,t}\|^2] + (\tilde{M}_{q,c} + 6\tau\tilde{E}_q\tilde{\mathcal{M}} - \frac{c_0\eta_{x,r}}{4})\mathbb{E}[\|\bar{q}_{r,t}\|^2]$$

$$+ (\tilde{M}_{g,l} + 6\tau\tilde{E}_g\tilde{\mathcal{M}})\frac{1}{N}\sum_{n=1}^{N}\mathbb{E}[\|g_{h,r,t}^{(n,k)} - h_{r,t}^{(n,k)}\|^2] + \tilde{M}_0 + 6\tau\tilde{E}\tilde{\mathcal{M}} \ . \tag{60}$$

Following similar steps of Eq. (37), by setting

$$\nu_k = \frac{1}{\eta_{x,r}N}\left(\frac{65}{\beta_x} + \frac{17}{10c_0\beta_y} + \frac{\sum_{k'=1}^{K} K\hat{\theta}_{k'}c_0 + 5c_0N}{\alpha}\right)C_K^2 \sum_{j=k}^{K}(2C_g^2)^{j-k} \triangleq \frac{\hat{\nu}_k}{\eta_{x,r}N},$$

$$\eta_{x,r} \leq \hat{\nu}_k^{1/2} \cdot \left(\sum_{j=k}^{K}\alpha(\sum_{k'=1}^{K}\hat{\nu}_{k'} + 4\hat{\nu}_j)(2C_g^2)^{j-k}\right)^{-1/2}, \tag{61}$$

the coefficient of $\frac{1}{N}\sum_{n=1}^{N}\mathbb{E}[\|g_{h,r,t}^{(n,k)} - h_{r,t}^{(n,k)}\|^2]$ can be upper bounded by:

$$-4\tau^2 C_K^2 \alpha\eta_{x,r}^2 \sum_{j=k}^{K}(2C_g^2)^{j-k}. \tag{62}$$

Similarly, for the term $\mathbb{E}[\|\bar{p}_{r,t}\|^2]$, we enforce

$$\frac{32\eta_{x,r}}{\beta_x N}C_K^2\sum_{k=0}^{K}(2C_g^2)^k + \frac{8\eta_{x,r}^2}{\beta_y\eta_{y,r}N}L_f^2(2C_g^2)^K + 2\eta_{x,r}^2\sum_{k=1}^{K}(\frac{\theta_k}{N} + \nu_k)(2C_g^2)^k$$

$$+ 24\tau^2\eta_{x,r}^2(4C_K^2 + 1)\sum_{k=0}^{K}(2C_g^2)^k\tilde{\mathcal{M}} - \frac{\eta_{x,r}}{4} - \frac{c_0\eta_{x,r}^2}{8\eta_{y,r}} \leq 0,$$

and for the term $\mathbb{E}[\|\bar{q}_{r,t}\|^2]$, we enforce

$$\frac{32\eta_{y,r}^2}{\beta_x\eta_{x,r}N}C_K^2 + \frac{8\eta_{y,r}}{\beta_y N}L_f^2 + 96\tau^2\eta_{y,r}^2C_K^2\tilde{\mathcal{M}} - \frac{c_0\eta_{x,r}}{4} \leq 0.$$

Following similar steps of Eq. (40) and Eq. (41), we obtain

$$\beta_x \geq \max\{3600\frac{C_K^2}{N}V_{2C_g}, 25600\frac{c_0C_K^2}{N}\}, \quad \beta_y \geq \max\{100\frac{C_K^2}{c_0N}V_{2C_g}, 1280\frac{C_K^2}{N}\},$$

$$\alpha \geq 64V_{2C_g}\frac{\sum_{k=1}^{K}Kc_0\hat{\theta}_k + 5c_0N}{N}C_K^2, \quad \eta_{y,r} \leq \min\{\frac{1}{\tau C_K}, \frac{\left((C_K^2 + 1)\sum_{k=1}^{K}(2C_g^2)^k\right)^{-1/2}}{360\tau C_K}\},$$

where $V_{2C_g} = \sum_{k=1}^{K}(2C_g^2)^k\sum_{j=k}^{K}(2C_g^2)^{j-k}$.

Then, we set

$$L_v^2 = \max\left\{1, C_K^2(C_K^2 + 1)(V_{2C_g} + 1)\left(V_{2C_g} + V_{2C_g}\sum_{k=1}^{K}K\hat{\theta}_k + V_{2C_g}N + 1\right)\right\}$$

$$\beta_y = 1280\frac{L_v^2}{N}, \beta_x = 51200\frac{c_0L_v^2}{N}, \alpha = 1280\frac{c_0L_v^2}{N}, \tag{63}$$

It indicates $\beta_x = 40c_0\beta_y, \alpha = c_0\beta_y$.

As a result, we reformulate Eq. (60) and derive

$$\frac{1}{T_r}\sum_{r=0}^{T_r-1}\left(\mathbb{E}[\|\nabla\mathcal{L}(\bar{x}_{r,t})\|^2] + \frac{c_0\eta_{x,r}}{\eta_{y,r}}\mathbb{E}[\|\nabla_y f(G(\bar{x}_{r,t}), \bar{y}_{r,t})\|^2]\right) \leq \frac{4(\mathcal{P}_{r,0} - \mathcal{P}_{r,T_r})}{\eta_{x,r}T_r} + \frac{4}{\eta_{x,r}}(\tilde{M}_0 + 6\tau\tilde{E}\tilde{\mathcal{M}}). \tag{64}$$

In particular, from the definition of $\tilde{E}$, $\tilde{M}_0$, and $\tilde{\mathcal{M}}$ in Eq. (53), (55), (59), with $\beta_x = 40c_0\beta_y, \alpha = c_0\beta_y, \eta_{x,r} = \frac{\eta_{y,r}}{10c_0}$, and $C_\tau$ in Eq. (49), we obtain

$$\frac{4}{\eta_{x,r}}(\tilde{M}_0 + 6\tau\tilde{E}\tilde{\mathcal{M}}) \leq 245c_0\beta_y\eta_{y,r}^2L_v^2\sigma^2C_\tau. \tag{65}$$

Moreover, from the definition of $\mathcal{P}_{r,0}$ in Eq. (47), we obtain

$$
\begin{aligned}
\frac{4(\mathcal{P}_{r,0} - \mathcal{P}_{r,T_r})}{\eta_{x,r} T_r} \leq{} & \frac{4\mathcal{V}_{r,0}}{\eta_{x,r} T_r} + \frac{32\sigma_{r,0}^x}{\beta_x \eta_{x,r}^2 T_r} + \frac{8\sigma_{r,0}^y}{\beta_y \eta_{x,r} \eta_{y,r} T_r} + \sum_{k=1}^{K} \frac{4K c_0 \hat{\theta}_k \sigma_{r,0}^{h_\theta,k}}{\alpha \eta_{x,r}^2 T_r} \\
& + \sum_{k=1}^{K} \frac{4\sigma_{r,0}^{h_\nu,k}}{\eta_{x,r} T_r} \Big( \frac{65}{\beta_x \eta_{x,r} N} + \frac{17}{\beta_y \eta_{y,r} N} + \frac{\sum_{k'=1}^{K} \hat{\theta}_{k'} c_0 + 5c_0 N}{\alpha \eta_{x,r} N} \Big) C_K^2 \sum_{j=k+1}^{K} (2C_g^2)^{j-k} \\
\leq{} & \frac{40 c_0 \mathcal{V}_{r,0}}{\eta_{y,r} T_r} + \frac{80 c_0 \sigma_{r,0}^x}{\beta_y \eta_{y,r}^2 T_r} + \frac{80 c_0 \sigma_{r,0}^y}{\beta_y \eta_{y,r}^2 T_r} + \frac{400 c_0 \sigma_{r,0}^{h_\theta}}{\beta_y \eta_{y,r}^2 T_r} + \frac{3730 c_0 \sigma_{r,0}^{h_\nu}}{\beta_y \eta_{y,r}^2 T_r} \\
\leq{} & \frac{40 c_0 \mathcal{V}_{r,0}}{\eta_{y,r} T_r} + \frac{80 c_0}{\beta_y \eta_{y,r}^2 T_r} \Big( \sigma_{r,0}^x + \sigma_{r,0}^y + 5\sigma_{r,0}^{h_\theta} + 47\sigma_{r,0}^{h_\nu} \Big) ,
\end{aligned}
$$

where $\sigma_{r,0}^{h_\theta} = \sum_{k=1}^{K} \hat{\theta}_k \sigma_{r,0}^{h_\theta,k}$, $\sigma_{r,0}^{h_\nu} = \frac{1}{N} \sum_{k=1}^{K} \nu_k^\sigma \sigma_{r,0}^{h_\nu,k}$, $\nu_k^\sigma = C_K^2 (1 + K \sum_{k=1}^{K} \hat{\theta}_k) \sum_{j=k+1}^{K} (2C_g^2)^{j-k}$.

Therefore, we obtain

$$
\begin{aligned}
& \frac{1}{T_r} \sum_{r=0}^{T_r-1} \Big( \mathbb{E}[\|\nabla \mathcal{L}(\bar{x}_{r,t})\|^2] + \frac{c_0 \eta_{x,r}}{\eta_{y,r}} \mathbb{E}[\|\nabla_y f(G(\bar{x}_{r,t}), \bar{y}_{r,t})\|^2] \Big) \\
& \leq \frac{40 c_0 \mathcal{V}_{r,0}}{\eta_{y,r} T_r} + \frac{80 c_0}{\beta_y \eta_{y,r}^2 T_r} \Big( \sigma_{r,0}^x + \sigma_{r,0}^y + 5\sigma_{r,0}^{h_\theta} + 47\sigma_{r,0}^{h_\nu} \Big) + 245 c_0 \beta_y \eta_{y,r}^2 L_v^2 \sigma^2 C_\tau .
\end{aligned}
$$

The proof is complete. $\qquad\square$

### D.4. Proof of Lemma D.2

*Proof.* From Lemma C.10, we obtain

$$
\begin{aligned}
& \frac{1}{T_r} \sum_{t=0}^{T_r-1} \sum_{k=1}^{K} \nu_k^\sigma \frac{1}{N} \sum_{n=1}^{N} \mathbb{E}[\|h_{r,t+1}^{(n,k)} - g_{h,r,t+1}^{(n,k)}\|^2] \\
& \leq \frac{1}{T_r} \sum_{t=0}^{T_r-1} \Bigg( \sum_{k=1}^{K} (1 - \alpha \eta_{x,r}^2) \nu_k^\sigma + 2\alpha^2 \eta_{x,r}^4 \sum_{k=1}^{K} \Big( \sum_{j=k+1}^{K} \nu_j^\sigma (2C_g^2)^{j-k} \Big) \Bigg) \frac{1}{N} \sum_{n=1}^{N} \mathbb{E}[\|h_{r,t}^{(n,k)} - g_{h,r,t}^{(n,k)}\|^2] \\
& + 2\eta_{x,r}^2 \frac{1}{T_r} \sum_{t=0}^{T_r-1} \sum_{k=1}^{K} \nu_k^\sigma (2C_g^2)^k \frac{1}{N} \sum_{n=1}^{N} \mathbb{E}[\|\bar{p}_{r,t} - p_{r,t}^{(n)}\|^2] + 2\eta_{x,r}^2 \frac{1}{T_r} \sum_{t=0}^{T_r-1} \sum_{k=1}^{K} \nu_k^\sigma (2C_g^2)^k \mathbb{E}[\|\bar{p}_{r,t}\|^2] \\
& + 2\alpha^2 \eta_{x,r}^4 \sigma^2 \sum_{k=1}^{K} \nu_k^\sigma \sum_{j=0}^{k-1} (2C_g^2)^j .
\end{aligned}
$$

Applying Lemma D.4 and with definition of $L_v^2$, we further derive

$$
\begin{aligned}
& \frac{1}{T_r} \sum_{t=0}^{T_r-1} \sum_{k=1}^{K} \nu_k^\sigma \frac{1}{N} \sum_{n=1}^{N} \mathbb{E}[\|h_{r,t+1}^{(n,k)} - g_{h,r,t+1}^{(n,k)}\|^2] \\
& \overset{\text{Lemma D.4}}{\leq} \frac{1}{T_r} \sum_{t=0}^{T_r-1} \Bigg( \sum_{k=1}^{K} (1 - \alpha \eta_{x,r}^2) \nu_k^\sigma + 2\alpha^2 \eta_{x,r}^4 \sum_{k=1}^{K} \Big( \sum_{j=k+1}^{K} \nu_j^\sigma (2C_g^2)^{j-k} \Big) \Bigg) \frac{1}{N} \sum_{n=1}^{N} \mathbb{E}[\|h_{r,t}^{(n,k)} - g_{h,r,t}^{(n,k)}\|^2] \\
& + 2\eta_{x,r}^2 \sum_{k=1}^{K} \nu_k^\sigma (2C_g^2)^k \Bigg( 6\tau \tilde{E}_p \frac{1}{T_r} \sum_{t=0}^{T_r-1} \mathbb{E}[\|\bar{p}_{r,t}\|^2] + 6\tau \tilde{E}_q \frac{1}{T_r} \sum_{t=0}^{T_r-1} \mathbb{E}[\|\bar{q}_{r,t}\|^2] \\
& + 6\tau \tilde{E}_g \frac{1}{T_r} \sum_{t=0}^{T_r-1} \frac{1}{N} \sum_{n=1}^{N} \mathbb{E}[\|h_{r,t}^{(n,k)} - g_{h,r,t}^{(n,k)})\|^2] + 6\tau \tilde{E} \Bigg) + 2\eta_{x,r}^2 L_v^2 \frac{1}{T_r} \sum_{t=0}^{T_r-1} \sum_{k=1}^{K} \mathbb{E}[\|\bar{p}_{r,t}\|^2] + 2\alpha^2 \eta_{x,r}^4 L_v^2 \sigma^2
\end{aligned}
$$

$$\leq (1 - \frac{1}{2}\alpha\eta_{x,r}^2)\frac{1}{T_r}\sum_{t=0}^{T_r-1}\sum_{k=1}^{K}\nu_k^{\sigma}\frac{1}{N}\sum_{n=1}^{N}\mathbb{E}[\|h_{r,t}^{(n,k)} - g_{h,r,t}^{(n,k)}\|^2]$$

$$+ 2\eta_{x,r}^2 L_v^2\left(\frac{24}{625}\frac{1}{T_r}\sum_{t=0}^{T_r-1}\mathbb{E}[\|\bar{p}_{r,t}\|^2] + \frac{24}{625}\frac{1}{T_r}\sum_{t=0}^{T_r-1}\mathbb{E}[\|\bar{q}_{r,t}\|^2]\right) + 2\eta_{x,r}^2 L_v^2\frac{1}{T_r}\sum_{t=0}^{T_r-1}\mathbb{E}[\|\bar{p}_{r,t}\|^2]$$

$$+ 12\eta_{x,r}^2\sum_{k=1}^{K}\nu_k^{\sigma}(2C_g^2)^k\tau\tilde{E} + 2\alpha^2\eta_{x,r}^4 L_v^2\sigma^2 \,, \tag{66}$$

where the last step follows from Eq. (61) and $\tilde{E}_p \leq \frac{2}{25\tau}, \tilde{E}_q \leq \frac{2}{25\tau}$.

By reformulating this expression and accounting for the random sampling in each outer iteration, we arrive at,

$$\sigma_{r+1,0}^{h_\nu} = \frac{1}{N}\sum_{k=1}^{K}\nu_k^{\sigma}\frac{1}{N}\sum_{n=1}^{N}\mathbb{E}[\|h_{r+1,0}^{(n,k)} - g_{h,r+1,0}^{(n,k)}\|^2] = \frac{1}{T_r}\sum_{t=0}^{T_r-1}\frac{1}{N}\sum_{k=1}^{K}\nu_k^{\sigma}\frac{1}{N}\sum_{n=1}^{N}\mathbb{E}[\|h_{r,t}^{(n,k)} - g_{h,r,t}^{(n,k)}\|^2]$$

$$\leq \frac{2}{\alpha\eta_{x,r}^2 T_r}\frac{1}{N}\sum_{k=1}^{K}\nu_k^{\sigma}\frac{1}{N}\sum_{n=1}^{N}\mathbb{E}[\|h_{r,0}^{(n,k)} - g_{h,r,0}^{(n,k)}\|^2] + \frac{4L_v^2}{\alpha N}\left(\frac{24}{625}\frac{1}{T_r}\sum_{t=0}^{T_r-1}\mathbb{E}[\|\bar{p}_{r,t}\|^2] + \frac{24}{625}\frac{1}{T_r}\sum_{t=0}^{T_r-1}\mathbb{E}[\|\bar{q}_{r,t}\|^2]\right)$$

$$+ \frac{4L_v^2}{\alpha N}\frac{1}{T_r}\sum_{t=0}^{T_r-1}\mathbb{E}[\|\bar{p}_{r,t}\|^2] + \frac{12}{\alpha N}\sum_{k=1}^{K}\nu_k^{\sigma}(2C_g^2)^k\tau\tilde{E} + 4\alpha\eta_{x,r}^2 L_v^2\sigma^2$$

$$\leq \frac{200c_0\sigma_{r,0}^{h_\nu}}{\beta_y\eta_{y,r}^2 T_r} + \frac{5L_v^2}{\beta_y N}\left(\frac{1}{T_r}\sum_{t=0}^{T_r-1}\mathbb{E}[\|\bar{p}_{r,t}\|^2] + \frac{1}{10}\frac{1}{T_r}\sum_{t=0}^{T_r-1}\mathbb{E}[\|\bar{q}_{r,t}\|^2]\right) + \frac{446}{N}c_0\tau^2\beta_y\eta_{y,r}^4 L_v^2\sigma^2 + \frac{c_0\beta_y\eta_{y,r}^2}{25N}L_v^2\sigma^2 \,,$$

where the last step holds due to $\beta_x = 40c_0\beta_y, \alpha = c_0\beta_y$, and $\eta_{x,r} = \frac{\eta_{y,r}}{10c_0}$.

Following similar steps, from Lemma C.7, Lemma C.8, and Lemma C.9, we derive

$$\sigma_{r+1,0}^{x} = \mathbb{E}[\|\bar{p}_{r+1,0} - \frac{1}{N}\sum_{n=1}^{N}\nabla_x f_{h,r+1,0}^{(n)}\|^2] \leq \frac{5}{2\beta_y\eta_{y,r}^2 T_r}\left(\sigma_{r,0}^{x} + 4\sigma_{r,0}^{h_\nu}\right) + \frac{590}{N}c_0\tau^2\beta_y\eta_{y,r}^4 L_v^2\sigma^2$$

$$+ \frac{101L_v^2}{\beta_y N}\left(\frac{1}{T_r}\sum_{t=0}^{T_r-1}\mathbb{E}[\|\bar{p}_{r,t}\|^2] + \frac{1}{10}\frac{1}{T_r}\sum_{t=0}^{T_r-1}\mathbb{E}[\|\bar{q}_{r,t}\|^2]\right) + \frac{1}{N}c_0\beta_y\eta_{y,r}^2 L_v^2\sigma^2 \,,$$

$$\sigma_{r+1,0}^{y} = \mathbb{E}[\|\bar{q}_{r+1,0} - \frac{1}{N}\sum_{n=1}^{N}\nabla_y f_{h,r+1,0}^{(n)}\|^2] \leq \frac{1}{\beta_y\eta_{y,r}^2 T_r}(\sigma_{r,0}^{y} + 4\sigma_{r,0}^{h_\nu}) + \frac{240}{N}c_0\tau^2\beta_y\eta_{y,r}^4 L_v^2\sigma^2$$

$$+ \frac{42L_v^2}{\beta_y N}\left(\frac{1}{T_r}\sum_{t=0}^{T_r-1}\mathbb{E}[\|\bar{p}_{r,t}\|^2] + \frac{1}{10}\frac{1}{T_r}\sum_{t=0}^{T_r-1}\mathbb{E}[\|\bar{q}_{r,t}\|^2]\right) + \frac{21}{10N}c_0\beta_y\eta_{y,r}^2 L_v^2\sigma^2 \,,$$

$$\sigma_{r+1,0}^{h_\theta} = \sum_{k=1}^{K}\hat{\theta}_k\mathbb{E}[\|\frac{1}{N}\sum_{n=1}^{N}h_{r+1,0}^{(n,k)} - \frac{1}{N}\sum_{n=1}^{N}g_{h,r+1,0}^{(n,k)}\|^2] \leq \frac{200}{\beta_y\eta_{y,r}^2 T_r}\left(\sigma_{r,0}^{h_\theta} + \sigma_{r,0}^{h_\nu}\right) + \frac{892}{N}c_0\tau^2\beta_y\eta_{y,r}^4 L_v^2\sigma^2$$

$$+ \frac{10L_v^2}{\beta_y N}\left(\frac{1}{T_r}\sum_{t=0}^{T_r-1}\mathbb{E}[\|\bar{p}_{r,t}\|^2] + \frac{1}{10}\frac{1}{T_r}\sum_{t=0}^{T_r-1}\mathbb{E}[\|\bar{q}_{r,t}\|^2]\right) + \frac{2}{25N}c_0\beta_y\eta_{y,r}^2 L_v^2\sigma^2 \,.$$

Next, we combine the four resulting inequalities with $\beta_y = 1280\frac{L_v^2}{N}$, which yields:

$$\sigma_{r+1,0}^{x} + \sigma_{r+1,0}^{y} + 5\sigma_{r+1,0}^{h_\theta} + 47\sigma_{r+1,0}^{h_\nu}$$

$$\leq \frac{5\sigma_{r,0}^{x}}{2\beta_y\eta_{y,r}^2 T_r} + \frac{\sigma_{r,0}^{y}}{\beta_y\eta_{y,r}^2 T_r} + \frac{1000\sigma_{r,0}^{h_\theta}}{\beta_y\eta_{y,r}^2 T_r} + \frac{10420\sigma_{r,0}^{h_\nu}}{\beta_y\eta_{y,r}^2 T_r} + \frac{26300}{N}c_0\tau^2\beta_y\eta_{y,r}^4 L_v^2\sigma^2$$

$$+ \frac{450L_v^2}{\beta_y N}\left(\frac{1}{T_r}\sum_{t=0}^{T_r-1}\mathbb{E}[\|\bar{p}_{r,t}\|^2] + \frac{1}{10}\frac{1}{T_r}\sum_{t=0}^{T_r-1}\mathbb{E}[\|\bar{q}_{r,t}\|^2]\right) + \frac{6}{N}c_0\beta_y\eta_{y,r}^2 L_v^2\sigma^2$$

$$\leq \frac{5\sigma_{r,0}^x}{2\beta_y\eta_{y,r}^2 T_r} + \frac{\sigma_{r,0}^y}{\beta_y\eta_{y,r}^2 T_r} + \frac{1000\sigma_{r,0}^{h_\theta}}{\beta_y\eta_{y,r}^2 T_r} + \frac{10420\sigma_{r,0}^{h_\nu}}{\beta_y\eta_{y,r}^2 T_r}$$

$$+ \frac{1}{2}\left(\frac{1}{T_r}\sum_{t=0}^{T_r-1}\mathbb{E}[\|\bar{p}_{r,t}\|^2] + \frac{1}{10}\frac{1}{T_r}\sum_{t=0}^{T_r-1}\mathbb{E}[\|\bar{q}_{r,t}\|^2]\right) + 182c_0\beta_y\eta_{y,r}^2 L_v^2\sigma^2\left(\frac{1}{N} + \frac{145}{N}\tau^2\eta_{y,r}^2\right). \quad (67)$$

For the term $\frac{1}{T_r}\sum_{t=0}^{T_r-1}\mathbb{E}[\|\bar{p}_{r,t}\|^2] + \frac{c_0\eta_{y,r}}{\eta_{x,r}}\frac{1}{T_r}\sum_{t=0}^{T_r-1}\mathbb{E}[\|\bar{q}_{r,t}\|^2]$, we obtain

$$\mathbb{E}[\|\bar{p}_{r,t}\|^2] + \frac{c_0\eta_{x,r}}{\eta_{y,r}}\mathbb{E}[\|\bar{q}_{r,t}\|^2]$$

$$\leq 2\mathbb{E}[\|\bar{p}_{r,t} - \nabla_x f(G(\bar{x}_{r,t}),\bar{y}_{r,t}) + \nabla_x f(G(\bar{x}_{r,t}),\bar{y}_{r,t}) - \nabla\mathcal{L}(\bar{x}_{r,t})\|^2] + 2\mathbb{E}[\|\nabla\mathcal{L}(\bar{x}_{r,t})\|^2]$$

$$+ \frac{c_0\eta_{x,r}}{\eta_{y,r}}\left(2\mathbb{E}[\|\bar{q}_{r,t} - \nabla_y f(G(\bar{x}_{r,t}),\bar{y}_{r,t})\|^2] + 2\mathbb{E}[\|\nabla_y f(G(\bar{x}_{r,t}),\bar{y}_{r,t})\|^2]\right)$$

$$\leq 4\mathbb{E}[\|\nabla\mathcal{L}(\bar{x}_{r,t}) - \nabla_x f(G(\bar{x}_{r,t}),\bar{y}_{r,t})\|^2] + 2\mathbb{E}[\|\nabla\mathcal{L}(\bar{x}_{r,t})\|^2] + \frac{2c_0\eta_{x,r}}{\eta_{y,r}}\mathbb{E}[\|\nabla_y f(G(\bar{x}_{r,t}),\bar{y}_{r,t})\|^2]$$

$$+ 4\mathbb{E}[\|\bar{p}_{r,t} - \nabla_x f(G(\bar{x}_{r,t}),\bar{y}_{r,t})\|^2] + \frac{2c_0\eta_{x,r}}{\eta_{y,r}}\mathbb{E}[\|\bar{q}_{r,t} - \nabla_y f(G(\bar{x}_{r,t}),\bar{y}_{r,t})\|^2]$$

$$\leq \frac{8}{\eta_{x,r}}\left(\frac{\eta_{x,r}}{4}\mathbb{E}[\|\nabla\mathcal{L}(\bar{x}_{r,t})\|^2] + \frac{\eta_{x,r}}{4}\frac{c_0\eta_{x,r}}{\eta_{y,r}}\mathbb{E}[\|\nabla_y f(G(\bar{x}_{r,t}),\bar{y}_{r,t})\|^2]\right) + 4\frac{L_f^2}{\mu^2}\mathbb{E}[\|\nabla_y f(G(\bar{x}_{r,t}),\bar{y}_{r,t})\|^2]$$

$$+ 4\mathbb{E}[\|\bar{p}_{r,t} - \nabla_x f(G(\bar{x}_{r,t}),\bar{y}_{r,t})\|^2] + \frac{2c_0\eta_{x,r}}{\eta_{y,r}}\mathbb{E}[\|\bar{q}_{r,t} - \nabla_y f(G(\bar{x}_{r,t}),\bar{y}_{r,t})\|^2].$$

Applying Eq. (56) with $\eta_{x,r} = \frac{\eta_{y,r}}{10c_0}$, we obtain

$$\mathbb{E}[\|\bar{p}_{r,t}\|^2] + \frac{c_0\eta_{x,r}}{\eta_{y,r}}\mathbb{E}[\|\bar{q}_{r,t}\|^2]$$

$$\leq \frac{8}{\eta_{x,r}}\left(\frac{\eta_{x,r}}{4}\mathbb{E}[\|\nabla\mathcal{L}(\bar{x}_{r,t})\|^2] + \frac{\eta_{x,r}}{4}\frac{c_0\eta_{x,r}}{\eta_{y,r}}\mathbb{E}[\|\nabla_y f(G(\bar{x}_{r,t}),\bar{y}_{r,t})\|^2]\right) + 4\frac{L_f^2}{\mu^2}\mathbb{E}[\|\nabla_y f(G(\bar{x}_{r,t}),\bar{y}_{r,t})\|^2]$$

$$+ \sum_{k=1}^{K}(16KA_k + \frac{4}{5}L_f^2 K\sum_{k=1}^{K}C_g^{2(K-k)})\mathbb{E}[\|\frac{1}{N}\sum_{n=1}^{N}g_{h,r,t}^{(n,k)}) - \frac{1}{N}\sum_{n=1}^{N}h_{r,t}^{(n,k)}\|^2] + 16\mathbb{E}[\|\frac{1}{N}\sum_{n=1}^{N}\nabla_x f_{h,r,t}^{(n)} - \bar{p}_{r,t}\|^2]$$

$$+ \frac{4}{5}\mathbb{E}[\|\frac{1}{N}\sum_{n=1}^{N}\nabla_y f_{h,r,t}^{(n)} - \bar{q}_{r,t}\|^2] + 17C_K^2\frac{1}{N}\sum_{n=1}^{N}\mathbb{E}[\|\bar{x}_{r,t} - x_{r,t}^{(n)}\|^2] + 17C_K^2\frac{1}{N}\sum_{n=1}^{N}\mathbb{E}[\|\bar{y}_{r,t} - y_{r,t}^{(n)}\|^2]. \quad (68)$$

In addition, from Eq. (60), Eq. (62), and Eq. (65), we obtain

$$\frac{8}{\eta_{x,r}}\left(\frac{\eta_{x,r}}{4}\mathbb{E}[\|\nabla\mathcal{L}(\bar{x}_{r,t})\|^2] + \frac{\eta_{x,r}}{4}\frac{c_0\eta_{x,r}}{\eta_{y,r}}\mathbb{E}[\|\nabla_y f(G(\bar{x}_{r,t}),\bar{y}_{r,t})\|^2]\right) \leq \frac{8(\mathcal{P}_{r,t} - \mathcal{P}_{r,t+1})}{\eta_{x,r}} - \frac{c_0}{2}\mathbb{E}[\|\nabla_y f(G(\bar{x}_{r,t}),\bar{y}_{r,t})\|^2]$$

$$- 8\frac{17}{5}\mathbb{E}[\|\frac{1}{N}\sum_{n=1}^{N}\nabla_x f_{h,r,t}^{(n)} - \bar{p}_{r,t}\|^2] - 8\frac{8}{5}10c_0\mathbb{E}[\|\frac{1}{N}\sum_{n=1}^{N}\nabla_y f_{h,r,t}^{(n)} - \bar{q}_{r,t}\|^2]$$

$$- 8\sum_{k=1}^{K}\left(\frac{34}{10}KA_k + c_0 L_f^2 KC_g^{2(K-k)}\right)\mathbb{E}[\|\frac{1}{N}\sum_{n=1}^{N}h_{r,t}^{(n,k)} - \frac{1}{N}\sum_{n=1}^{N}g_{h,r,t}^{(n,k)}\|^2]$$

$$- 32\tau^2 C_K^2\alpha\eta_{x,r}^2\sum_{k=1}^{K}\sum_{j=k+1}^{K}(2C_g^2)^{j-k}\frac{1}{N}\sum_{n=1}^{N}\mathbb{E}[\|h_{r,t}^{(n,k)} - g_{h,r,t}^{(n,k)}\|^2] + 490c_0\beta_y\eta_{y,r}^2 L_v^2\sigma^2 C_\tau. \quad (69)$$

Combining Eq. (68) and Eq. (69), with $c_0 = \frac{32L_f^2}{\mu^2}$, we further deduce

$$\mathbb{E}[\|\bar{p}_{r,t}\|^2] + \frac{c_0\eta_{x,r}}{\eta_{y,r}}\mathbb{E}[\|\bar{q}_{r,t}\|^2]$$

$$\leq \frac{8(\mathcal{P}_{r,t} - \mathcal{P}_{r,t+1})}{\eta_{x,r}} + 490 c_0 \beta_y \eta_{y,r}^2 L_v^2 \sigma^2 C_\tau - 32\tau^2 C_K^2 \alpha \eta_{x,r}^2 \sum_{k=1}^{K} \sum_{j=k+1}^{K} (2C_g^2)^{j-k} \frac{1}{N} \sum_{n=1}^{N} \mathbb{E}[\|h_{r,t}^{(n,k)} - g_{h,r,t}^{(n,k)}\|^2]$$

$$+ 17 C_K^2 \frac{1}{N} \sum_{n=1}^{N} \mathbb{E}[\|\bar{x}_{r,t} - x_{r,t}^{(n)}\|^2] + 17 C_K^2 \frac{1}{N} \sum_{n=1}^{N} \mathbb{E}[\|\bar{y}_{r,t} - y_{r,t}^{(n)}\|^2] .$$

By summing over $t = \{0, \cdots, T_r - 1\}$ and applying the overall consensus error Lemma D.4, we derive

$$\frac{1}{T_r} \sum_{t=0}^{T_r-1} \left( \mathbb{E}[\|\bar{p}_{r,t}\|^2] + \frac{c_0 \eta_{x,r}}{\eta_{y,r}} \mathbb{E}[\|\bar{q}_{r,t}\|^2] \right)$$

$$\leq \frac{8(\mathcal{P}_{r,0} - \mathcal{P}_{r,t_t})}{\eta_{x,r} T_r} + 490 c_0 \beta_y \eta_{y,r}^2 L_v^2 \sigma^2 C_\tau - 32\tau^2 C_K^2 \alpha \eta_{x,r}^2 \sum_{k=1}^{K} \sum_{j=k+1}^{K} (2C_g^2)^{j-k} \frac{1}{T_r} \sum_{t=0}^{T_r-1} \frac{1}{N} \sum_{n=1}^{N} \mathbb{E}[\|h_{r,t}^{(n,k)} - g_{h,r,t}^{(n,k)}\|^2]$$

$$+ 17 C_K^2 \tau^2 \eta_{x,r}^2 \frac{1}{T_r} \sum_{t=0}^{T_r-1} \frac{1}{N} \sum_{n=1}^{N} \mathbb{E}[\|\bar{p}_{r,t} - p_{r,t}^{(n)}\|^2] + 17 C_K^2 \tau^2 \eta_{y,r}^2 \frac{1}{T_r} \sum_{t=0}^{T_r-1} \frac{1}{N} \sum_{n=1}^{N} \mathbb{E}[\|\bar{q}_{r,t} - q_{r,t}^{(n)}\|^2]$$

$$\leq \frac{8(\mathcal{P}_{r,0} - \mathcal{P}_{r,t_t})}{\eta_{x,r} T_r} + 490 c_0 \beta_y \eta_{y,r}^2 L_v^2 \sigma^2 C_\tau - 32\tau^2 C_K^2 \alpha \eta_{x,r}^2 \sum_{k=1}^{K} \sum_{j=k+1}^{K} (2C_g^2)^{j-k} \frac{1}{T_r} \sum_{t=0}^{T_r-1} \frac{1}{N} \sum_{n=1}^{N} \mathbb{E}[\|h_{r,t}^{(n,k)} - g_{h,r,t}^{(n,k)}\|^2]$$

$$+ \frac{34}{360^2} \left( 6\tau \tilde{E}_g \frac{1}{T_r} \sum_{t=0}^{T_r-1} \frac{1}{N} \sum_{n=1}^{N} \mathbb{E}[\|h_{r,t}^{(n,k)} - g_{h,r,t}^{(n,k)}\|^2] + \frac{24}{625} \frac{1}{T_r} \sum_{t=0}^{T_r-1} \mathbb{E}[\|\bar{p}_{r,t}\|^2] + \frac{24}{625} \frac{1}{T_r} \sum_{t=0}^{T_r-1} \mathbb{E}[\|\bar{q}_{r,t}\|^2] + 6\tau \tilde{E} \right)$$

$$\leq \frac{8(\mathcal{P}_{r,0} - \mathcal{P}_{r,t_t})}{\eta_{x,r} T_r} + 491 c_0 \beta_y \eta_{y,r}^2 L_v^2 \sigma^2 C_\tau + \frac{1}{9} \left( \frac{1}{T_r} \sum_{t=0}^{T_r-1} \mathbb{E}[\|\bar{p}_{r,t}\|^2] + \frac{1}{10} \frac{1}{T_r} \sum_{t=0}^{T_r-1} \mathbb{E}[\|\bar{q}_{r,t}\|^2] \right) . \tag{70}$$

Reformulating this expression then gives

$$\frac{1}{T_r} \sum_{t=0}^{T_r-1} \left( \mathbb{E}[\|\bar{p}_{r,t}\|^2] + \frac{c_0 \eta_{x,r}}{\eta_{y,r}} \mathbb{E}[\|\bar{q}_{r,t}\|^2] \right) \leq \frac{9(\mathcal{P}_{r,0} - \mathcal{P}_{r,t_t})}{\eta_{x,r} T_r} + 560 c_0 \beta_y \eta_{y,r}^2 L_v^2 \sigma^2 C_\tau$$

$$\leq \frac{90 c_0 \mathcal{V}_{r,0}}{\eta_{y,r} T_r} + \frac{180 c_0}{\beta_y \eta_{y,r}^2 T_r} \left( \sigma_{r,0}^x + \sigma_{r,0}^y + 5\sigma_{r,0}^{h_\theta} + 47\sigma_{r,0}^{h_\nu} \right) + 560 c_0 \beta_y \eta_{y,r}^2 L_v^2 \sigma^2 C_\tau .$$

Finally, substituting this inequality into Eq. (67) yields

$$\sigma_{r+1,0}^x + \sigma_{r+1,0}^y + 5\sigma_{r+1,0}^{h_\theta} + 47\sigma_{r+1,0}^{h_\nu}$$

$$\leq \frac{5\sigma_{r,0}^x}{2\beta_y \eta_{y,r}^2 T_r} + \frac{\sigma_{r,0}^y}{\beta_y \eta_{y,r}^2 T_r} + \frac{1000 \sigma_{r,0}^{h_\theta}}{\beta_y \eta_{y,r}^2 T_r} + \frac{10420 \sigma_{r,0}^{h_\nu}}{\beta_y \eta_{y,r}^2 T_r}$$

$$+ \frac{1}{2} \left( \frac{1}{T_r} \sum_{t=0}^{T_r-1} \mathbb{E}[\|\bar{p}_{r,t}\|^2] + \frac{1}{10} \frac{1}{T_r} \sum_{t=0}^{T_r-1} \mathbb{E}[\|\bar{q}_{r,t}\|^2] \right) + 182 c_0 \beta_y \eta_{y,r}^2 L_v^2 \sigma^2 \left( \frac{1}{N} + \frac{145}{N} \tau^2 \eta_{y,r}^2 \right)$$

$$\leq \frac{5\sigma_{r,0}^x}{2\beta_y \eta_{y,r}^2 T_r} + \frac{\sigma_{r,0}^y}{\beta_y \eta_{y,r}^2 T_r} + \frac{1000 \sigma_{r,0}^{h_\theta}}{\beta_y \eta_{y,r}^2 T_r} + \frac{10420 \sigma_{r,0}^{h_\nu}}{\beta_y \eta_{y,r}^2 T_r} + \frac{1}{2} \left( \frac{90 c_0 \mathcal{V}_{r,0}}{\eta_{y,r} T_r} + \frac{180 c_0}{\beta_y \eta_{y,r}^2 T_r} \left( \sigma_{r,0}^x + \sigma_{r,0}^y + 5\sigma_{r,0}^{h_\theta} + 47\sigma_{r,0}^{h_\nu} \right) \right.$$

$$\left. + 560 c_0 \beta_y \eta_{y,r}^2 L_v^2 \sigma^2 C_\tau \right) + 182 c_0 \beta_y \eta_{y,r}^2 L_v^2 \sigma^2 \left( \frac{1}{N} + \frac{145}{N} \tau^2 \eta_{y,r}^2 \right)$$

$$\leq \frac{320 c_0}{\beta_y \eta_{y,r}^2 T_r} \left( \sigma_{r,0}^x + \sigma_{r,0}^y + 5\sigma_{r,0}^{h_\theta} + 47\sigma_{r,0}^{h_\nu} \right) + \frac{45 c_0 \mathcal{V}_{r,0}}{\eta_{y,r} T_r} + 480 c_0 \beta_y \eta_{y,r}^2 L_v^2 \sigma^2 C_\tau .$$

$\square$

