# OpenReview forum: "LS$^{2}$MC-GDA: A Smoothed Algorithm for Federated Stochastic Multi-Level Compositional Minimax Optimization"
_ICML.cc/2026/Conference — ICML 2026 regular_

### Official Review · Reviewer_m7J7 · 2026-03-12

**Soundness:** 2
**Presentation:** 2
**Significance:** 3
**Originality:** 3
**Overall Recommendation:** 4
**Confidence:** 2

**Summary:**

The authors consider federated stochastic multi-level NC-PL saddle point problems. They propose a novel scheme and derive a bound on its communication complexity of $O(\kappa/\varepsilon^2)$. Improvement over previous best $O(\kappa^3/\varepsilon^3)$ is achieved via utilization of smoothing technique. A modification of the method is also proposed for the PL-PL scenario. Experimental section considers AUC maximization on the CATvsDOG dataset, as well as binary interpretations of CIFAR-10, CIFAR-100, and STL-10.

**Compliance With Llm Reviewing Policy:**

Affirmed.

**Final Justification:**

The authors have addressed my concerns regarding reported results and provided comprehensive explanations of the experimental part of their work. Given that the proposed algorithm is a combination of well-established techniques, I consider 4 to be the appropriate score.

**Key Questions For Authors:**

1) Could the authors clarify how the bound in Theorem 3.6 becomes independent of the level $K$ after substituting the proposed hyperparameters?

2) Could the authors explain the unusual stochastic behavior observed in the experiments? If this is standard for saddle point problems, could they provide a discussion of this phenomenon with appropriate citations?

I am willing to reconsider my evaluation if all questions are addressed.

**Limitations:**

Yes

**Strengths And Weaknesses:**

Strengths:

1) Algorithmic design combines several techniques (variance reduction, smoothing, and local steps) within a single algorithm and provides a rigorous analysis with convergence guarantees under the mild NC-PL assumption.

2) The proposed LSSMC-GDA enjoys significant dominance over previous known communication bound $O(\kappa^3/\varepsilon^3)$

3) Experiments cover a wide range of problems and baselines. The corresponding section includes a description of the algorithm’s hyperparameter tuning, which may be useful for its implementation by other researchers.

Weaknesses:

1) Despite the merits such as the description of technical challenges and proof sketches, the paper is difficult to read. For example, the text does not specify which communication complexity metric is considered (rounds, communications, etc); the algorithm refers to numerous equations in different places of the text (a complete listing in the appendix would have been useful to immediately understand the main ideas); the experimental section does not discuss the observed results.

2) My main concern pertains to Theorem 3.6 and its corresponding Corollary 3.7. First, I would like to point out that parameters $\gamma_y$ and $\gamma_z$ are absent from the result of Theorem 3.6, which may indicate possible typos. Furthermore, the authors claim that their bound is level-independent. However, this does not seem to be the case, since the constant $K$ corresponding to the level is not included in the chosen hyperparameter values.

3) Additionally to the previous issue, I am concerned that authors handle stochasticity by using a batch of size $O(\kappa^{1/2}/\varepsilon)$. In my view, this issue could potentially be addressed. For example, in [1], the authors develop a stochastic analysis for centralized NC-SC problems without relying on batching.

4) In the numerical section, I am concerned about the convergence behavior of the methods. The large stochastic fluctuations at the beginning of training, which abruptly subside around epoch 40, may indicate potential data manipulation. Authors do not discuss this phenomenon in the text.

---

[1] Guo, Zhishuai, et al. "Unified convergence analysis for adaptive optimization with moving average estimator." arXiv preprint arXiv:2104.14840 (2021).

---

> ### Author Rebuttal · Authors · 2026-03-27
>
> We thank the reviewer for the helpful comments and questions, and we address them as follows.
>
> ----
>
> **W1:** Communication complexity refers to the number of communication rounds, which is the standard metric in federated learning; we will clarify this explicitly in the revision. For the algorithm, we will include a complete version in the appendix for better readability. For the experimental section, we discussed that our method outperforms the baselines, and analyzed the effects of key hyperparameters(e.g., imratio, smoothing $\gamma$, and inner-level $K$). We will further expand the discussion to better interpret the results.
>
> **W2 and Q1:** In Theorem 3.6, the parameters $\gamma_y$ and $\gamma_z$  depend on $\gamma_x$ from Eq.(43). We omitted the full expression of Eq.(43) in the main text for brevity. We will add these relation in the revision to avoid ambiguity.
>
> By “level-dependent”, we means that **the order of the convergence rate does not depend on $K$**, although $K$ may appear in the constants. For example, as discussed in the related work, prior methods such as Yang et al.(2019) achieve a rate of $O(\epsilon^{-(7 + K)/ 2})$, which depends exponentially on $K$, while subsequent works obtain level-independent rates such as $O(\epsilon^{-4})$ and $O(\epsilon^{-3})$. Our result also falls into this category:  $K$ does not affect the order of rate. We will clarify this distinction more explicitly.
>
> * Yang et al.(2019): Multilevel stochastic gradient methods for nested composition optimization.
>
> **W3:** As stated in Corollary 3.7, the batch size $S_0$ is only used at the first step ($t=0$). When $t>0$, our algorithm operates with constant(i.e., $O(1)$) batch size, similar to [1].
>
> **W4 and Q2:** As described in the experimental setup, “the learning rate $\gamma\eta$ is decayed by a factor of 10 at the 10 at the 50-th and 75-th epochs for all methods to avoid overfitting”. This strategy is standard in deep auc maximization(e.g., Yuan et al. (2021), Zhang et al. (2023), Liu et al. (2024)).
>
> The observed fluctuations in early epochs are typical for stochastic minimax training. After learning rate decay, the training becomes more stable, which explains the smoother behavior after around epoch 50. We will add a brief discussion of this phenomenon in the revision.
> * Yuan et al. (2021): Compositional training for end-to-end deep auc maximization.
> * Zhang et al. (2023): Federated compositional deep auc maximization.
> * Liu et al. (2024): Faster stochastic variance reduction methods for compositional minimax optimization.

---

> > ### Author Rebuttal · Reviewer_m7J7 · 2026-04-03
> >
> > Thank you to the authors for their efforts in preparing the response. I have no further questions regarding this work, and I am raising my score to 4.

---

> > > ### Author Response · Authors · 2026-04-03
> > >
> > > We appreciate that our clarification has adequately addressed your concerns, and we sincerely thank you for the increase in score.
> > >
> > > ---
> > >
> > > **Update after the reviewer posted the final justification**
> > >
> > > We would like to respectfully emphasize that our paper is not a direct combination of well-established technique. Rather, we address unique challenges in federated learning. Specifically, we show how to bound the consensus error to establish communication complexity in the presence of a smoothing term for both nonconvex-PL and PL-PL problems. More importantly, we address the unique challenges arising from the smoothing technique and the new problem structures.
> > >
> > > - *New challenges in bounding consensus errors due to the smoothing technique.* When bounding the consensus error for the gradient estimator $p$ in Lemma C.11, we must handle the update of an auxiliary variable, i.e., $\|z^{(n)}\_{t+1} - z^{(n)}\_{t}\|^2$ (Lines 1054–1055). This aspect is absent in existing work. For example, [2] (Zhang, Xinwen, et al.) does not involve such an auxiliary variable, making our setting more challenging. In [a], the auxiliary variable is updated on the central server rather than on local workers, which simplifies their analysis compared to ours.
> > >
> > > - *New challenges in establishing communication complexity for PL-PL problems.* Establishing communication complexity for PL-PL compositional minimax problems is significantly more challenging than in prior work. In particular, compared to [a], our loss function has a compositional structure, which introduces bias in the stochastic gradients; hence, their approach cannot be directly applied. To address this issue, we develop an induction-based approach to establish communication complexity. Compared to [2], which focuses only on the nonconvex–strongly concave setting, it remains unclear how their method can be extended to the PL-PL case. Compared to [1], which considers only the single-machine setting, our distributed setting leads to a more challenging convergence analysis.
> > >
> > > In summary, our problem setup is substantially different from existing works, and our paper successfully addresses the unique challenges arising in this new setting. Moreover, our results are of independent interest and can be extended to other optimization settings, such as traditional minimax problems [a] and constrained minimization problems [b]. Therefore, our work makes important contributions to the study of smoothing techniques in federated learning and has broad implications for federated optimization.
> > >
> > > [1] Zhang, Xinwen, and Hongchang Gao. "On the Convergence of Stochastic Smoothed Multi-Level Compositional Gradient Descent Ascent." The Thirty-ninth Annual Conference on Neural Information Processing Systems.
> > >
> > > [2] Zhang, Xinwen, et al. "A federated stochastic multi-level compositional minimax algorithm for deep AUC maximization." Forty-first International Conference on Machine Learning. 2024.
> > >
> > > [a] Shen, W., Huang, M., Zhang, J., and Shen, C. Stochastic smoothed gradient descent ascent for federated minimax optimization. In International Conference on Artificial Intelligence and Statistics, pp. 3988–3996. PMLR, 2024.
> > >
> > > [b] Huang R, Zhang J, Alacaoglu A. Stochastic Smoothed Primal-Dual Algorithms for Nonconvex Optimization with Linear Inequality Constraints[J]. arXiv preprint arXiv:2504.07607, 2025.

---

### Official Review · Reviewer_Mys3 · 2026-03-12

**Soundness:** 2
**Presentation:** 3
**Significance:** 3
**Originality:** 3
**Overall Recommendation:** 4
**Confidence:** 4

**Summary:**

This article proposes a novel algorithm for federated stochastic compositional minimax optimization. It solves a multi-level compositional problem with faster convergence rates in terms of sample complexity and communication complexity.  Numerical experiments show the effectiveness of the approach compared to the state-of-the art.

**Compliance With Llm Reviewing Policy:**

Affirmed.

**Final Justification:**

I have raised my score to 4 since they answered all my questions. The readability of the statement of Lemma C.11 could still be improved to clarify the points discussed during the rebuttal.

**Key Questions For Authors:**

- Eq 9 seems to have some notation issue as it does not make sense to rewrite a variable mathematically. My point is that in you proof on the consensus error in lemma C.11, how do you take into account of the effect of the communication period? I do not see how the choice of \tau impacts the convergence analysis in the lemma.

- The proof seems to be incomplete in the sense that the definition of s_t on line 1018 is not defined in the text before. I suspect that this might a typo, but it could be a flaw as well.

- Are the results in Fig 2 about the last iterate performance? The theory is about the ergodic iterate, so this point should be clarified.

**Strengths And Weaknesses:**

The article proposes an interesting algorithm to improve convergence rates based on smoothing. It overcomes slowdown issues related to a condition number. The article is mostly well written, and lengthy.

Some technical points could still be discussed as I find that the proof might contain some flaw. From a high-level, I do not understand why the algorithm could converge since it is mentioned in Section 3.2 that there is an inherent bias in the gradients of eq 3 if using stochastic gradients. Even though you could reduce the variance in the stochastic gradients, it is not so clear the algorithm still optimizes the original function.

---

> ### Author Rebuttal · Authors · 2026-03-27
>
> We thank the reviewer for the helpful comments and questions, and we address them as follows.
>
> -----
>
> **W1:** The biased stochastic estimators discussed in Section 3.2 do not prevent convergence; rather, they represent the core technical challenge in federated multi-level compositional minimax optimization. The key idea is to control and bound the bias accumulated across levels. Our analysis explicitly handles this by jointly controlling the gradient estimation error and consensus error via a recursive potential function. For example, in Appendix C.2.1, we establish the recursive upper bound for the gradient estimation error.  With these upper bounds for gradient estimation errors, we establish that the iterates converge to an $(\epsilon, \epsilon / \sqrt{\kappa})$-stationarity point of the original objective.
>
> **Q1:** Eq.(9) is not intended to rewrite a variable, but to describe the periodic communication step: once every $\tau$ iterations, each local variable is replaced by the average across workers, which is standard in federated learning.
>
> The effect of the communication period $\tau$ is incorporated throughout the consensus analysis. In particular, $\tau$ appears in the bounds in Lemma C.11, is further handled in Lemma C.14, and ultimately enters the final convergence bound in Theorem 3.6, as reflected in the last three terms of Eq.(10). Therefore, the dependence on $\tau$ is fully accounted for in the analysis.
>
> **Q2:** The symbol $s_t$ (introduced in line 1070) denotes the index of the latest communication round before iteration $t$. We will clarify this in the revised version.
>
> **Q3:** Figure 2 reports the test AUC over training epochs, rather than the ergodic iterate. The figures show the performance of iterates along training (per epoch), which is standard in empirical evaluation.

---

> > ### Author Rebuttal · Reviewer_Mys3 · 2026-04-02
> >
> > I thank the author for their clarification. I still do not understand one point about your **consensus error** analysis, that is when t % tau = 0, each local variable is replaced by the global average across all workers, i would expect to have zero consensus error in your analysis of Lemma C.11. I do not understand this upper bound, even though it involves the s_t that you have clarified. Why do you consider s_t times tau in the sum of hat t in Lemma C.11?
> >
> > Even though your Figure 2 is standard in empirical evaluation, i think there is a gap between the theory and practical evaluation, which should still be mentioned.

---

> > > ### Author Response · Authors · 2026-04-03
> > >
> > > We appreciate the follow-up questions that provide an opportunity for further clarification.
> > >
> > > ---
> > >
> > > **Follow-up 1:** We agree that for the consensus error, when $t$ % $\tau = 0$, the error becomes zero. This implies that at iteration $\hat{t} = s_t \times \tau$, we have $[\\| p_{\hat{t}}^{(n)} - p_{\hat{t}}\\|^2] = 0$. The notation that counts from $\hat{t} = s_t \times \tau$ to $t$ is primarily for convenience, indicating that the consensus error accumulates starting from the most recent communication round prior to iteration $t$.
> > >
> > > Alternatively, one may write the starting point as $\hat{t}  = s_t \times \tau + 1$.  These two formulations are identical in effect, since the consensus error at $s_t \times \tau$ is zero, and hence including or excluding this iteration does not change the accumulated error.
> > >
> > > **Follow-up 2:** For Figure 2, we will explicitly clarify that the original figure reports the performance of the last iterate. In addition, we included new experimental results on the catvsdog dataset with communication period is $\tau = 8$, and imbalance ratio 0.02, where the test AUC is evaluated using the ergodic iterate. (Please see [Figure](https://anonymous.4open.science/r/Anonymous43-F530/catvsdog_ergodic.pdf)). It can still be observed that our proposed method outperforms the baseline methods.
> > >
> > > ---
> > >
> > > We hope that our clarification has adequately addressed your concerns.

---

### Official Review · Reviewer_uSfQ · 2026-03-13

**Soundness:** 3
**Presentation:** 3
**Significance:** 2
**Originality:** 2
**Overall Recommendation:** 4
**Confidence:** 3

**Summary:**

This paper addresses the problem of multi-level composite minimax optimization in a federated learning setting.  The main contribution is closing the gap in the centralized and federated learning convergence rates in terms of the condition number $\kappa$ and the stationarity gap $\epsilon$. In order to do so, the authors propose to perform stochastic variance reduction (STORM) updates on the forward and backward passes through the compositional gradient structure.

**Compliance With Llm Reviewing Policy:**

Affirmed.

**Final Justification:**

The authors have satisfactorily addressed my concerns regarding the paper's contribution, in the rebuttal.  The paper has sufficient originality and I believe my earlier score of 4 is appropriate.

**Key Questions For Authors:**

- In the start of section 3.3, the authors mention that the slow convergence can be fixed with STORM-like updates that reduce variance in the gradients.  However, when referring to [2], I notice that they also perform STORM-like updates to the gradients (momentum terms).  Specifically, Eqs. (6)-(7) in the manuscript are the same as Eqs. (10)-(11) in [2].  Is there another reason for the speed-up in convergence in the manuscript?  It would be very clarifying if the authors can specifically point out where the proof exactly differs from [2].

- In the experimental plots for AUC vs epochs (Fig. 1), the LocalSMCGDAM results are much lower than the plots in depicted in [2].  From the Fig. 2 in [2] it looks like the proposed approach should be tied with LocalSMCGDAM.  Can the authors clarify why there is a discrepancy here?

**Limitations:**

As stated earlier, clarifying comments on how the proof diverges from the reference [2] above would be very helpful.  Also, the experimental results of the baseline LocalSMCGDAM should be cross-checked with the original paper.

**Strengths And Weaknesses:**

#### Strengths
The convergence analysis seems thorough and overall the work appears sound and clear.

#### Weaknesses
In the light of previous works, the contribution seems somewhat incremental.  Specifically, previously in [1] the authors have already introduced a version of the smoothing function for centralized learning.  In [2] the authors already formulate the federated composite minimax problem and provide very similar STORM-like updates, but they do not do smoothing. The current manuscript seems to simply put these ideas together, and modify the convergence proof accordingly.

#### References
[1] Zhang, Xinwen, and Hongchang Gao. "On the Convergence of Stochastic Smoothed Multi-Level Compositional Gradient Descent Ascent." The Thirty-ninth Annual Conference on Neural Information Processing Systems.
[2] Zhang, Xinwen, et al. "A federated stochastic multi-level compositional minimax algorithm for deep AUC maximization." Forty-first International Conference on Machine Learning. 2024.

---

> ### Author Rebuttal · Authors · 2026-03-27
>
> We thank the reviewer for the helpful comments and questions, and we address them as follows.
>
> ----
>
> **W1 and Q1:** First, regarding [2], their method applies a STORM-like update only to the inner-level function in Eq.(9), while the updates in Eqs.(10)-(11) rely on **moving-average momentum**. In contrast, our method applies STORM-style variance reduction to both the inner-level estimators and the momentum terms, which is a key distinction. This difference is also reflected in the convergence rates: [2] achieve $O(1/N\epsilon^4)$, while our method improves this to $O(1/N\epsilon^3)$.
>
> Second, the improvement is not solely due to adopting STORM, but arises from a systematic redesign of both the algorithm and the analysis, incorporating smoothing techniques under the federated multi-level compositional minimax problems. In Section 3.3, we provide a detailed comparison highlight the **key differences from [2] in Technical Challenges**, including:
> 1) gradient errors on momentums,
> 2) global estimation error across inner levels,
> 3) consensus errors,
> 4) smoothed objective, and
> 5) different assumptions.
>
> These differences lead to a nontrivial changes in both the algorithm design and the convergence proof, resulting in improved dependence on both $\epsilon$ and $\kappa$, as summarized in Table 1. In addition, this type of analysis is **unique to the federated setting and does not arise in the single-machine setting** considered in [1].
>
> **Q2:** The discrepancy is due to different experimental settings. In [2], the imbalance ratio is 0.05, while we consider a more challenging setting with ratio 0.02, which leads to more severe class imbalance and consequently lower AUC scores. Therefore, the difference compared to [2] is expected. Notably, under this setting, our method still achieves better performance, demonstrating its robustness.

---

> > ### Author Rebuttal · Reviewer_uSfQ · 2026-04-04
> >
> > I thank the authors for their rebuttal.  My questions have been addressed somewhat satisfactorily, but I still feel the contribution is a little incremental in the light of previous works.  One way to potentially interpret this work is that it is a federated version of the existing centralized multi-level compositional minmax algorithm using both variance reduction and smoothing in [1] (Zhang & Gao, 2025), so this feels like a direct extension of the proofs of that work.  However, I have already given a score of 4 previously, so I will maintain it.

---

> > > ### Author Response · Authors · 2026-04-04
> > >
> > > We appreciate that our clarification has adequately addressed your concerns, and we sincerely thank you for the positive score.
> > >
> > > In addition, we would like to emphasize that our paper is not a direct extension of existing work [1] (Zhang & Gao, 2025). Rather, we address unique challenges in federated learning. Specifically, we show how to bound the consensus error to establish communication complexity in the presence of a smoothing term for both nonconvex-PL and PL-PL problems. More importantly, we address the unique challenges arising from the smoothing technique and the new problem structures.
> > >
> > > - *New challenges in bounding consensus errors due to the smoothing technique.* When bounding the consensus error for the gradient estimator $p$ in Lemma C.11, we must handle the update of an auxiliary variable, i.e., $\|z^{(n)}\_{t+1} - z^{(n)}\_{t}\|^2$ (Lines 1054–1055). This aspect is absent in existing work. For example, [2] (Zhang, Xinwen, et al.) does not involve such an auxiliary variable, making our setting more challenging. In [a], the auxiliary variable is updated on the central server rather than on local workers, which simplifies their analysis compared to ours.
> > >
> > > - *New challenges in establishing communication complexity for PL-PL problems.* Establishing communication complexity for PL-PL compositional minimax problems is significantly more challenging than in prior work. In particular, compared to [a], our loss function has a compositional structure, which introduces bias in the stochastic gradients; hence, their approach cannot be directly applied. To address this issue, we develop an induction-based approach to establish communication complexity. Compared to [2], which focuses only on the nonconvex–strongly concave setting, it remains unclear how their method can be extended to the PL-PL case. Compared to [1], which considers only the single-machine setting, our distributed setting leads to a more challenging convergence analysis.
> > >
> > > In summary, our problem setup is substantially different from existing works, and our paper successfully addresses the unique challenges arising in this new setting. Moreover, our results are of independent interest and can be extended to other optimization settings, such as traditional minimax problems [a] and constrained minimization problems [b]. Therefore, our work makes important contributions to the study of smoothing techniques in federated learning and has broad implications for federated optimization.
> > >
> > > [a] Shen, W., Huang, M., Zhang, J., and Shen, C. Stochastic smoothed gradient descent ascent for federated minimax optimization. In International Conference on Artificial Intelligence and Statistics, pp. 3988–3996. PMLR, 2024.
> > >
> > > [b] Huang R, Zhang J, Alacaoglu A. Stochastic Smoothed Primal-Dual Algorithms for Nonconvex Optimization with Linear Inequality Constraints[J]. arXiv preprint arXiv:2504.07607, 2025.

---

### Official Review · Reviewer_kRPT · 2026-03-22

**Soundness:** 3
**Presentation:** 2
**Significance:** 2
**Originality:** 2
**Overall Recommendation:** 4
**Confidence:** 4

**Summary:**

The paper proposes the Local Stochastic Smoothed Multi-level Compositional Gradient Descent Ascent $(LS^2MC-GDA)$ method for multi-level compositional optimization, integrating smoothing and variance-reduction techniques in a federated setting. The algorithm aims to address the slowdown in convergence in terms of condition number and convergence error that arises when incorporating techniques to mitigate bias in the global K-level function and its gradient. Under NC-PL conditions, they derive a sample complexity of $Ο(\kappa^{3/2}/N\epsilon^3)$ and the communication complexity of $Ο(\kappa/\epsilon^2)$.

**Compliance With Llm Reviewing Policy:**

Affirmed.

**Key Questions For Authors:**

- Is the stagewise variant needed for theoretical reasons only, or is it practically relevant?
- Corollary 3.7 does not follow directly from Theorem 3.6 for the given parameter choices, as the terms are not $Ο(\epsilon^2)$ for it to obtain an $(\epsilon,\epsilon/\sqrt{κ})$- stationary point. Can you clarify this?
- Since a smoothed optimisation problem is solved, which becomes strongly convex in $x$ due to regularisation, how does this solution relate to the original problem, which is not convex in $x$?

**Limitations:**

The paper assumes that functions $g^ {(n,k)} (.) $ and $ f^ {(n)} (.) $ are Lipschitz continuous, which is a strong condition. This assumption limits the scope of theoretical analysis. I think this assumption also implicitly allows tackling statistical heterogeneity across client objectives, which would otherwise pose an additional challenge for the analysis and needs an explicit characterization.

**Strengths And Weaknesses:**

**Strengths**
- The paper provides convergence rates while tackling bias in gradient computation using methods like STORM and controlling consensus estimation errors across local and global models for multi-level compositional minimax federated optimization.
- For a fair comparison, they also translate the $(\epsilon,\epsilon/\sqrt{κ})$- stationary point to an $ϵ$- stationary point for federated multi-level compositional optimization.
- The derived complexity improves dependence on $\kappa$ and error $\epsilon$ compared to the complexities obtained by existing algorithms.

**Weaknesses**
- The bound is slower than that of Shen et al., even though they consider Lipschitz continuity. What is the reason for this?
- There is insufficient introductory material on federated multi-compositional objective, and for an uninitiated reader, the presentation makes it difficult to understand the nuances of the author's approach, as both the main text and the appendix rely heavily on existing works.
- A few notations are not clear, such as the argument-wise gradient in Equation (6), the definition of $G^{n'}$ after Equation (1).
- Algorithms are incomplete due to missing variable initializations, making them hard to comprehend.
- The paper does not discuss the computational and memory overhead of auxiliary variables introduced in the algorithm, such as $h_t^{n,k}$, $p_t$, $z_t$, $q_t$, etc.
- The role of the stage-wise algorithm is not emphasized.

---

> ### Author Rebuttal · Authors · 2026-03-27
>
> We thank the reviewer for the helpful comments and questions, and we address them as follows.
>
> -----
>
> **W1:** The comparison involves two different results in our paper. Our main result under the NC–PL condition achieves improved sample complexity and the same communication complexity compared to Shen et al. In particular, our sample complexity $O(\frac{\kappa^{3/2}}{N\epsilon^3})$ improves over $O(\frac{\kappa^2}{N\epsilon^4})$ in Shen et al, while the communication complexity remains the same at $O(\frac{\kappa}{\epsilon^2})$.
>
> For the PL–PL setting, our method also improves the sample complexity, achieving $(\frac{1}{N\epsilon})$, compared to $O(\frac{1}{\log(\epsilon)N\epsilon^2})$ in Shen et al. The slightly slower communication complexity in this setting arises from the additional multi-level compositional structure, which introduces extra estimation and consensus errors that must be controlled. This distinction has been discussed in Table 1 and Remark 3.9.
>
> **W2-W4:** We will improve the presentation by providing more intuitive descriptions of the federated multi-level compositional minimax objective, clarifying the notations in Eq. (1) and Eq. (6), and including explicit algorithm initializations for completeness.
>
> **W5:** From the theoretical perspective, introducing auxiliary variables does not worsen the sample or communication complexity; instead, they are essential for achieving the improved rates. From the practical perspective, the additional overhead is moderate and structured: $p$ and $z$ are of the same order as $x$, $q$ is the same order as $y$, and the additional $h$ scale linearly with the number of compositional levels $K$. This type of overhead is standard in multi-level compositional methods.
>
> **W6:** The stage-wise algorithm serves as a theoretical bridge that converts the smoothing methods $(\epsilon, \epsilon / \sqrt{\kappa})$-stationarity guarantee into the standard $\epsilon$-stationarity. This translation is commonly used in smoothing approaches(e.g., Yang et al. (2022); Shen et al) and has been discussed in the introduction and Section 3.4.
>
> * Yang et al.(2022): Faster single-loop algorithms for minimax optimization without strong concavity.
>
> **Q1:** The stage-wise variant is not only for theoretical purposes. It is also applicable in settings where the objective satisfies a two-sided PL condition, which arises in several practical problems, such as linear quadratic regulator (LQR), robust phase retrieval, robust control, as discussed in Yang et al.(2020).
>
> * Yang et al.(2020): Global convergence and variance-reduced optimization for a class of nonconvex-nonconcave minimax problems.
>
> **Q2:** Corollary 3.7 is obtained by directly instantiating the hyperparameter choice in Theorem 3.6. In particular, from Eq.(43), we set $\beta = O(\frac{1}{N})$ and $\gamma_x = O(1)$. Then, considering the third term in Eq.(10), $O(\frac{\kappa\beta\eta^2K}{N})$, we choose $\eta = O(\frac{N\epsilon}{\kappa^{1/2}})$ to ensure it is $O(\epsilon^2)$. For the last term $O(\tau^2\eta^4\beta^2)$, we set $\tau = O(\frac{\kappa^{1/2}}{N\epsilon})$ so that it is also $O(\epsilon^2)$. Similarly, for the first term $O(\frac{\kappa\Phi_{0}}{\gamma_x\eta T})$, we choose $T = O(\frac{\kappa^{3/2}}{N\epsilon^3})$ to ensure it is $O(\epsilon^2)$. The same procedure applies to other terms and the choice of $S_0$. Hence, Corollary 3.7 follows directly.
>
> **Q3:** In Theorem 3.6, the convergence rate is established with respect to **the gradient of the original objective** $f(G(x), y)$, rather than the smoothed function $F(G(x), y; z)$. Therefore, the obtained solution directly corresponds to the original problem under the nonconvex–PL condition. The smoothing is only used as an analysis tool and does not alter the objective.

---

> > ### Author Rebuttal · Reviewer_kRPT · 2026-04-04
> >
> > Thank you for your rebuttal. I will keep my positive score unchanged. The presentation of your submitted draft must be improved if at all this paper is accepted to proceedings.

---

> > > ### Author Response · Authors · 2026-04-04
> > >
> > > We appreciate that our clarification has adequately addressed your concerns, and we sincerely thank you for the positive score.

---

### Decision · Program_Chairs · 2026-04-30

**Decision:**

Accept (regular)

**Comment:**

This paper proposes a novel algorithm, $LS^{2}MC-GDA$, for federated stochastic multi-level compositional minimax optimization. By cleverly combining smoothing techniques with variance reduction (STORM), the method successfully overcomes the unique theoretical challenges of controlling consensus errors within a federated setting. Under the nonconvex-PL condition, the algorithm achieves a sample complexity of $O(\kappa^{3/2}/N\epsilon^{3})$ and a communication complexity of $O(\kappa/\epsilon^{2})$, significantly improving upon existing theoretical results while achieving a linear speedup with respect to the number of workers $N$.
During the rebuttal phase, the authors adequately addressed the reviewers' questions regarding theoretical derivation details, demonstrating that their theoretical contributions are substantial and non-trivial.

Weaknesses:
Although technically solid, the work conceptually relies on integrating existing variance reduction and smoothing techniques within a federated scenario, which some reviewers felt was somewhat incremental. Furthermore, the paper's presentation requires refinement; some notation definitions are unclear, and the algorithmic pseudocode lacks variable initialization instructions, making the manuscript somewhat difficult to read.

Conclusion:
After a thorough discussion, the review committee unanimously recognized the paper's theoretical derivations and contributions under a complex federated optimization setting. With all reviewers issuing positive scores, this submission is uniformly recommended for acceptance.